# Beyond Average Value Function in Precision Medicine: Maximum Probability-Driven Reinforcement Learning for Survival Analysis

**Jianqi Feng**[*]
Shandong University
202412070@mail.sdu.edu.cn

**Chengchun Shi**
LSE
c.shi7@lse.ac.uk

**Zhenke Wu**
University of Michigan
zhenkewu@umich.edu

**Xiaodong Yan**[†]
Xi'an Jiaotong University
yanxiaodong@xjtu.edu.cn

**Wei Zhao**[†]
Shandong University
wzhao92@sdu.edu.cn

## Abstract

Constructing multistage optimal decisions for alternating recurrent event data is critically important in medical and healthcare research. Current reinforcement learning (RL) algorithms have only been applied to time-to-event data, with the objective of maximizing expected survival time. However, alternating recurrent event data has a different structure, which motivates us to model the probability and frequency of event occurrences rather than a single terminal outcome. In this paper, we introduce an RL framework specifically designed for alternating recurrent event data. Our goal is to maximize the probability that the duration between consecutive events exceeds a clinically meaningful threshold. To achieve this, we identify a lower bound of this probability, which transforms the problem into maximizing a cumulative sum of log probabilities, thus enabling direct application of standard RL algorithms. We establish the theoretical properties of the resulting optimal policy and demonstrate through numerical experiments that our proposed algorithm yields a larger probability of that the time between events exceeds a critical threshold compared with existing state-of-the-art algorithms.

## 1 Introduction

**Motivation**. Precision medicine is a paradigm that aims to tailor treatments to individual patient characteristics. Recent studies have increasingly developed reinforcement learning (RL) algorithms in this context (see, e.g., 28; 33; 48; 52; 49; 53; 58; 35; 36; 31; 25; 50; 26; 4; 7; 56; 57). This paper focuses on RL for survival analysis. The existing literature has primarily studied time-to-event data, with the objective of maximizing expected survival time (see Section 1.1 for details). However, for patients with chronic diseases, data often involves alternating recurrent events, and the clinical goal shifts to maximizing the probability that the duration between events exceeds a critical threshold. In response, we propose an RL framework designed to maximize this probability. Below, we provide an example to illustrate the structure of alternating recurrent event data.

Our work is motivated by the ongoing Intern Health Study (IHS) which recruited first-year medical interns at US institutions to study factors that may impact their mental health and general well-being (29). The medical internship is an initial step towards trained and practicing physicians. They

---

[*]All authors contributed equally and are listed in alphabetical order by surname.
[†]Corresponding Author

39th Conference on Neural Information Processing Systems (NeurIPS 2025).

often face difficult decisions, challenging shift work schedules, lack of time for exercise, and sleep disruptions, resulting in higher rates of depression (34). Wearable and smartphone devices provide daily ecological momentary assessments about physical activities, sleep, and mood scores over multiple months since the start of the internship. Digital health interventions sent via prompts on the study app serve as a low-touch and easily accessible approach to delivering data insights, tips, and other personalized content for mitigating mental health issues especially when access to therapists may not always be timely (19; 43). It has been an ongoing cohort that facilitates the study of genetic and lifestyle risk factors for mental and behavioral health outcomes. In this paper, we focus on the management of recurrent low-mood episodes, which requires a shift toward dynamic, personalized interventions. For example, we may define a low-mood episode by a significant drop (e.g., $20\%$) from an individual's baseline mood before internship. Moving beyond prediction, RL offers a powerful computational framework to learn optimal, adaptive intervention policies. The objective is to train an agent to select sequences of actions that either preemptively steer an individual away from an impending low-mood state or, should an episode occur, expedite their return to baseline, thereby minimizing the episode's duration and severity.

**Challenges**. Applying RL to alternating recurrent event data raises two challenges:

1. Alternating recurrent event data is substantially more complex than the time-to-event data, the latter being the focus of the current literature (14; 47). Unlike time-to-event data, alternating recurrent events can switch back and forth multiple times during the follow-up period, which substantially increases the modeling complexity.

2. Most existing RL algorithms consider maximizing the expectation. However, in survival analysis, the primary interest often lies in the probability that the duration between recurrent events exceeds a specified threshold, a perspective that has received limited attention in the existing literature.

**Contributions**. To address these challenges, we propose a novel maximum-probability-driven RL algorithm tailored to alternating recurrent event data. Our primary methodological contribution is the derivation of a lower bound for the maximum-probability objective, which reformulates the original objective into a sum of cumulative log probabilities. This enables the application of standard RL algorithms designed to maximize cumulative reward in the context of recurrent event data. We further provide theoretical guarantees and conduct extensive numerical experiments to demonstrate the effectiveness of the proposed algorithm.

## 1.1 Related Work

Our paper is closely related to two strands of research in survival analysis: one focusing on the use of RL for survival analysis, and the other handling recurrent event data.

**RL for survival analysis**. RL has recently received considerable attention in the survival analysis literature with censored outcomes (11). Specifically, Goldberg & Kosorok (14) and Liu et al. (23) construct a Q-function suitable for time-to-event censored data by inverse probability weighting. To address the issue of unequal numbers of individuals at each stage, they construct an auxiliary problem and provide an unbiased analysis of the optimal policy. Zhao et al. (54) offers a doubly robust estimation method based on this work. Liu et al. (22) uses the estimated Q-function values to fill in censored values, but no theoretical analysis is provided. Lee et al. (17) uses deep learning to learn the distribution of survival time for analysis. However, these methods are applied to time-to-event data and focus on maximizing expected value of survival time.

**Recurrent event data**. Given the recurrent nature of chronic diseases, the more recent trend of the existing literature has moved beyond the time-to-event data to handle recurrent event data. Among those available, Lee et al. (18) and Wang et al. (44) propose to use a counting process for modeling recurrent events and estimate the gap time between recurrent events. Xia et al. (45) and Loe et al. (24) propose to segment recurrent events within follow-up windows and estimate the probability of the duration of recurrent events considering all stages. However, these papers do not study policy optimization.

## 2 Data Structure and Optimization Objectives

In this section, we introduce the data structure of alternating recurrent events and our considered optimization objective. As mentioned earlier, unlike most existing studies in the literature, we focus on maximizing the probability that the duration time exceeds a pre-specified threshold.

**Data structure.** We consider a multistage decision-making process based on a censored recurrent event structure. Let $G_k$ $(k = 1, 2, \dots)$ be the duration of the $k$-th follow-up stage. For each stage, let $\boldsymbol{X}_k \in \mathcal{X}$ and $A_k \in \mathcal{A}$ respectively be the covariates collected and the treatment received at the beginning of the $k$-th stage, where $\mathcal{X} \subset \mathbb{R}^p$ is the set of states and $\mathcal{A}$ is a compactness set of available treatment options. The observed values of covariates $\boldsymbol{X}_k$ and treatment $A_k$ are denoted as $\boldsymbol{x}_k$ and $a_k$, respectively. Suppose that each subject's follow-up period begins at time 0. We consider subjects who can experience two types of alternating events that recur in multiple follow-up stages (45). We denote $L_i$ as the $i$-th occurrence time from high mood to low mood (first-type event ) and $H_i$ as the $i$-th occurrence time from low mood to high mood (second-type event). Suppose that $0 \le L_1 < H_1 < L_2 < H_2 < \cdots$ and $L_0 = H_0 = 0$.

In this work, we are interested in the following recurrent-event duration $T_k$ for stage $k$, which represents the duration from the occurrence of a second-type event to the next first-type event. The formula for $T_k$ is given by

$$\min\left\{ \sum_{i=J_{k-1}+1}^{J_k-1} (L_{i+1} - H_i) + \max\left( L_{J_{k-1}+1} - \sum_{i=1}^{k-1} G_i, 0 \right) + \max\left( \sum_{i=1}^{k} G_i - H_{J_k}, 0 \right), G_k \right\},$$

where $J_k = \arg\max_j (H_j \le \sum_{i=1}^{k} G_i)$ denotes the time of the last second-type recurrent event occurring at stage $k$. We assume that $J_0 = 0$ and $\sum_{i=a}^{b} \cdot = 0$ for $b < a$. Since $L_j$ and $H_j$ are random variables, we assume here a reward sample space $\mathcal{R}$ such that $T_k \in \mathcal{R}$. Let $C \in \mathbb{R}^+$ represent the censoring time measured from the beginning of the study until a censoring event occurs, such as the end of the study or the loss of follow-up, then the observed duration time at stage $k$ is given by $Y_k = \min\{T_k, C - \sum_{i=1}^{k-1} T_i\}$ and the corresponding censoring indicator at stage $k$ is denoted by $\Delta_k = \mathbf{1}(\sum_{i=1}^{k} T_i \le C)$. We assume that the censoring time is independent of covariates, treatments, and duration times for simplicity. Note that the censoring event can occur at any stage. Once the censoring event occurs at stage $k$, the treatments, covariates, and duration times after stage $k$ are not observed. Let the number of stages of treatment received by a subject be $\bar{K} \in \mathbb{N}^+$, where $\bar{K} = \inf\{k : \Delta_k = 0\}$. With notations above, the observed individual trajectories for one subject can be represented as (subject index omitted): $\{\boldsymbol{X}_1, A_1, \Delta_1, Y_1, \dots, \boldsymbol{X}_{\bar{K}}, A_{\bar{K}}, \Delta_{\bar{K}}, Y_{\bar{K}}\}$.

In Figure 1, we illustrate an individual trajectory from the motivating Intern Health Study. In this study, a treatment policy needs to be developed to determine whether to send weekly text messages to medical interns to maintain high mood levels for a longer duration. As illustrated, the shifts between high mood and low mood occur repeatedly. In this example, $G_k$ is consistently 7 (days) for any $k$, and $T_k$ of primary interest represents the duration of the high-mood period within each week. For this specific individual, a censoring event occurs in stage 3, resulting in $\bar{K} = 3$ and $Y_3$ being observed instead of the true $T_3$.

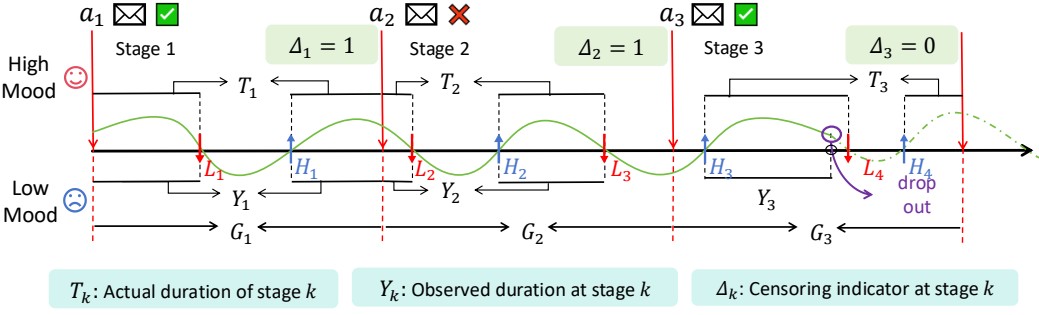

Figure 1: An example for the individual trajectory in the Intern Health Study.

**Optimization objective**. In traditional RL, the strategy is often formulated with the objective of the value function $\mathbb{E}^{P_0, \pi, P_T}[\sum_{k=t}^{\bar{K}} T_k | \boldsymbol{X}_t]$, where $P_0$ is the transition kernel that maps state space $\mathcal{X}$ and action space $\mathcal{A}$ to a distribution over state space $\mathcal{X}$, $\pi$ is a policy that maps state space $\mathcal{X}$ to a probability distribution over the action space $\mathcal{A}$, and $P_T$ is the distribution on $\mathcal{R}$ under condition of $\mathcal{X}$ and $\mathcal{A}$. Moreover, in health studies, we place more emphasis on the probability that the duration time is greater than a threshold at each stage. Therefore, we propose the following objective:

$$\mathsf{P}^\pi(\boldsymbol{x}_t) = \mathbb{P}^{P_0, \pi}\left(T_t > \alpha_t G_t, \ldots, T_{\bar{K}} > \alpha_{\bar{K}} G_{\bar{K}} \,|\, \boldsymbol{X}_t = \boldsymbol{x}_t\right), \tag{1}$$

where $\alpha_k \in [0, 1]$, for $k = 1, \ldots, \bar{K}$. In the context of the Intern Health Study illustrated in Figure 1, setting $\alpha_k = 4/7$ for all $k \leq \bar{K}$ implies that we aim for interns to maintain a high mood for at least four days per week throughout the follow-up period. However, if we expect interns to have a better mood in the first few weeks, $\alpha_k$'s can be increased slightly during those weeks. See Appendix A.3 for further discussion of $\alpha_k$ selection.

To handle (1), we first impose two assumptions (Assumption 2.1, 2.2). These assumptions allow us to decompose (1) into a product of stage-specific survival functions, resulting in a simplified objective function (see Lemma 2.3).

**Assumption 2.1** (Markov assumption). For any $v \geq 0$ and $k > t$, we assume that

$$\mathbb{P}(T_k > v | \{T_s\}_{s=t}^{k-1}, \{\boldsymbol{X}_s\}_{s=t}^k, \{A_s\}_{s=t}^k) = \mathbb{P}(T_k > v | \boldsymbol{X}_k, A_k).$$

**Assumption 2.2** (Conditional independence assumption). For any $k > t$, we assume that $(\boldsymbol{X}_k, A_k)$ and $\{T_s\}_{s=t}^{k-1}$ are conditional independent given $\{(\boldsymbol{X}_s, A_s)\}_{s=t}^{k-1}$.

**Lemma 2.3.** *When Assumptions 2.1 and 2.2 hold, we have*

$$\mathbb{P}^{P_0, \pi}\left(T_t > \alpha_t G_t, \ldots, T_{\bar{K}} > \alpha_{\bar{K}} G_{\bar{K}} \,|\, \boldsymbol{X}_t\right) = \mathbb{E}^{P_0, \pi}\left[\prod_{k=t}^{\bar{K}} S_{T_k}(\alpha_k G_k | \boldsymbol{X}_k, A_k) \,\Big|\, \boldsymbol{X}_t\right], \tag{2}$$

*where $S_{T_k}(\cdot | \boldsymbol{X}_k, A_k)$ is the conditional survival function of the duration time $T_k$ given $\boldsymbol{X}_k$ and $A_k$.*

The proof of Lemma 2.3 is provided in Appendix B.1. However, directly optimizing $\mathsf{P}^\pi(\boldsymbol{x}_t)$ is challenging because evaluating the objective requires computing the product of probabilities from $k = t$ to $\bar{K}$, which can become extremely small when $\bar{K}$ is large, making the resulting optimization unstable. To address this, we observe that, by Jensen's inequality,

$$V_0^\pi(\boldsymbol{x}_t) := \mathbb{E}^{P_0, \pi}\left[\sum_{k=t}^{\bar{K}} \log S_{T_k} \,\Big|\, \boldsymbol{X}_t = \boldsymbol{x}_t\right] \leq \log \mathbb{E}^{P_0, \pi}\left[\prod_{k=t}^{\bar{K}} S_{T_k} \,\Big|\, \boldsymbol{X}_t = \boldsymbol{x}_t\right] = \log \mathsf{P}^\pi(\boldsymbol{x}_t), \tag{3}$$

where $S_{T_k}$ is a shorthand for $S_{T_k}(\alpha_k G_k | \boldsymbol{X}_k, A_k)$ and we denote the lower bound of $\log \mathsf{P}^\pi(\boldsymbol{x}_t)$ by $V_0^\pi(\boldsymbol{x}_t)$. (3) motivates us to maximize this lower bound of $\log \mathsf{P}^\pi$, which equals a sum of expected cumulative log probabilities and can thus be optimized using existing RL frameworks. A natural question arises: does this shift in the objective alter the optimal policy? Theorem 2.4 below answer this question, showing that the two objectives induce the same optimal policy when $P_0$ is a deterministic transition kernel; that is, for every $\boldsymbol{x}, a$, there exists a unique $\boldsymbol{x}'$ such that $P_0(\boldsymbol{x}'|\boldsymbol{x}, a) = 1$.

**Theorem 2.4.** *Suppose that for any $\boldsymbol{x}_t$, the supremum of $V^\pi(\boldsymbol{x}_t)$ is attainable, and that $P_0$ is a deterministic transition kernel. Then, $\Pi^v = \Pi^p$, where $\Pi^v = \{\pi^v | V_0^{\pi^v}(\boldsymbol{x}_t) = \max_\pi V_0^\pi(\boldsymbol{x}_t)\}$ and $\Pi^p = \{\pi^p | \mathsf{P}^{\pi^p}(\boldsymbol{x}_t) = \max_\pi \mathsf{P}^\pi(\boldsymbol{x}_t)\}$.*

*Remark* 2.5. Theorem 2.4 applies to both discrete and continuous action spaces. In the continuous case, a deterministic policy can be represented as a degenerate distribution $\pi^\mu(a | \boldsymbol{x}) = \delta(a - \mu(\boldsymbol{x}))$, where $\delta(\cdot)$ is the Dirac delta function and $\mu$ is the deterministic mapping from states to actions.

The proof of Theorem 2.4 is provided in Appendix B.3. The value function in (3) remains very difficult to optimize, particularly in the long horizon setting with a large $\bar{K}$, due to the absence of a discount factor. To address this, we further derive a lower bound for the value function in (3) using the following discounted value function and introduce its associated Q-function:

$$V^\pi(\boldsymbol{x}_t) = \mathbb{E}^{P_0, \pi}\left[\sum_{k=t}^{\bar{K}} \gamma^{k-t} \log S_{T_k} \,\Big|\, \boldsymbol{x}_t\right]; Q^\pi(\boldsymbol{x}_t, a_t) = \mathbb{E}^{P_0, \pi}\left[\sum_{k=t}^{\bar{K}} \gamma^{k-t} \log S_{T_k} \,\Big|\, \boldsymbol{x}_t, a_t\right], \tag{4}$$

for some the discount factor $\gamma < 1$. Our optimization procedure for this discounted value function will be detailed in Section 3.

# 3 Estimation Procedure

In this section, we provide a detailed description of the proposed policy optimization procedure. When censoring occurs, i.e., when $\Delta_k = 0$, we treat the process as terminating in the RL framework (a "Done" state), and then return the reward to optimize policy. Consequently, the data collected at each stage are subject to censoring. Unlike general RL, the reward here needs to be estimated. So we first estimate the reward, then plug in the estimated reward to optimize the policy.

We first present the estimation methods and asymptotic properties of the survival function estimator $\hat{S}_{T_k}$, obtained using either the Cox proportional hazards model (10) or the Aalen's additive hazards model (1). These models are chosen for their widespread use and relative simplicity, although more complex alternatives exist (e.g., transformation models, varying-coefficient models).

**Definition**. Let $\mathcal{D}_k = \{(\boldsymbol{X}_{i,k}, A_{i,k}, Y_{i,k}, \Delta_{i,k})\}_{1 \leq i \leq N_k}$ denote the data collected at stage $k$, representing $N_k$ independent observations. Denote the counting process at stage $k$ as $U_{ik}(t) = I(Y_{i,k} \leq t, \Delta_{i,k} = 1)$, and the corresponding at-risk process as $V_{ik}(t) = I(Y_{i,k} \geq t)$. Since $S_{T_k}(t|\boldsymbol{X}_k, A_k) = \exp\{-\Lambda_{T_k}(t|\boldsymbol{X}_k, A_k)\}$, where $\Lambda_{T_k}(t|\boldsymbol{X}_k, A_k)$ is the cumulative hazard function, estimating $S_{T_k}$ is equivalent to estimating $\Lambda_{T_k}(t|\boldsymbol{X}_k, A_k)$. We define the covariates vector $\boldsymbol{Z}_k := (\boldsymbol{X}_k^\top, A_k, A_k \boldsymbol{X}_k^\top)^\top$ and $\boldsymbol{Z}_{ik} := (\boldsymbol{X}_{i,k}^\top, A_{i,k}, A_{i,k}\boldsymbol{X}_{i,k}^\top)^\top$ for subjects. Subsequently, we provide the estimation of $\Lambda_{T_k}(t|\boldsymbol{X}_k, A_k)$ in the following two models.

**Cox model**. We assume a stage-specific Cox proportional hazards model as $\Lambda_{T_k}(t|\boldsymbol{X}_k, A_k) = \Lambda_{0,k}(t) \exp\left\{\boldsymbol{\eta}_k^\top \boldsymbol{Z}_k\right\}$, $k = 1, \ldots, \bar{K}$, where $\Lambda_{0,k}(t)$'s are the stage-specific unknown cumulative baseline hazard functions and $\boldsymbol{\eta}_k$'s are the stage-specific unknown $(2p+1)$-dimension unknown parameters. Estimates of $\boldsymbol{\eta}_k$ and $\Lambda_{0,k}(t)$ can be obtained by maximizing the partial likelihood (6; 2). Specifically,

$$\hat{\boldsymbol{\eta}}_k = \text{argmax}_{\boldsymbol{\eta}} \prod_{i=1}^{N_k} \left( \frac{e^{\boldsymbol{\eta}^\top \boldsymbol{Z}_{ik}}}{\sum_{j=1}^{N_k} V_{jk}(Y_{i,k}) e^{\boldsymbol{\eta}^\top \boldsymbol{Z}_{jk}}} \right)^{\Delta_{i,k} \mathbf{1}\{Y_{i,k} \leq t\}} , \text{ and}$$

$$\hat{\Lambda}_{0,k}(t) = \int_0^t \frac{\sum_{i=1}^{N_k} dU_{ik}(s)}{\sum_{i=1}^{N_k} V_{ik}(s) e^{\hat{\boldsymbol{\eta}}_k^\top \boldsymbol{Z}_{ik}}}. \tag{5}$$

Consequently, estimators for $\Lambda_{T_k}(t|\boldsymbol{X}_k, A_k)$ are given by

$$\hat{\Lambda}_{T_k}(t|\boldsymbol{X}_k, A_k) = \hat{\Lambda}_{0,k}(t) \exp\left\{\hat{\boldsymbol{\eta}}_k^\top \boldsymbol{Z}_k\right\}, \ k = 1, \ldots, \bar{K}. \tag{6}$$

**Additive hazard model**. We assume a stage-specific Aalen's additive hazards model as $\Lambda_{T_k}(t \mid \boldsymbol{Z}_k) = \Lambda_{0,k}(t) + \boldsymbol{\beta}_k^\top \boldsymbol{Z}_k t$, $k = 1, \ldots, \bar{K}$, where $\Lambda_{0,k}(t)$'s are the cumulative baseline hazard functions, and $\boldsymbol{\beta}_k$'s are the vectors of regression coefficients. Following the arguments in (21), the estimator for $\boldsymbol{\beta}_k$ is given by

$$\hat{\boldsymbol{\beta}}_k = \frac{\sum_{j=1}^{N_k} \int_0^\infty V_{jk}(s) \left[\boldsymbol{Z}_{jk} - \bar{\boldsymbol{Z}}_k(s)\right]^{\otimes 2} ds}{\sum_{j=1}^{N_k} \int_0^\infty \left[\boldsymbol{Z}_{jk} - \bar{\boldsymbol{Z}}_k(s)\right] dU_{jk}(s)}, \text{ where } \bar{\boldsymbol{Z}}_k(s) = \frac{\sum_{j=1}^{N_k} V_{jk}(s)\boldsymbol{Z}_{jk}}{\sum_{j=1}^{N_k} V_{jk}(s)}, \tag{7}$$

with $\mathbf{a}^{\otimes 2} = \mathbf{a}\mathbf{a}^\top$. $\Lambda_{0,k}(t)$ is then estimated by

$$\hat{\Lambda}_{0,k}(\hat{\boldsymbol{\beta}}_k, t) = \int_0^t \sum_{j=1}^{N_k} \left\{dU_{jk}(s) - V_{jk}(s)\hat{\boldsymbol{\beta}}_k^\top \boldsymbol{Z}_{jk} ds\right\} / \sum_{j=1}^{N_k} V_{jk}(s).$$

Consequently, the estimator for $\Lambda_{T_k}(t \mid \boldsymbol{Z}_k)$ is given by $\hat{\Lambda}_{T_k}(t \mid \boldsymbol{Z}_k) = \hat{\Lambda}_{0,k}(\hat{\boldsymbol{\beta}}_k, t) + \hat{\boldsymbol{\beta}}_k^\top \boldsymbol{Z}_k t$.

We summarize the consistency and the asymptotic normality property of $\hat{\Lambda}_{T_k}(t \mid \boldsymbol{Z}_k)$ in Theorem 3.1. The detailed proof is provided in Appendix B.7.

**Theorem 3.1.** *For each $1 \leq k \leq \bar{K}$, suppose that there exists a constant $M_k$ such that $G_k \leq M_k$. Then, under assumptions B.2-B.5 for the Cox model, or assumptions B.6-B.9 for the additive hazards model, for any $t \in [0, G_k]$, as $N_k \to \infty$, we have*

(i) $\sup_{t \in [0, G_k]} |\hat{\Lambda}_{T_k}(t \mid \boldsymbol{Z}_k) - \Lambda_{T_k}(t \mid \boldsymbol{Z}_k)| \xrightarrow{p} 0$;

(ii) $\hat{\Lambda}_{T_k}(t \mid \boldsymbol{Z}_k) - \Lambda_{T_k}(t \mid \boldsymbol{Z}_k) = O_p(N_k^{-\frac{1}{2}})$;

(iii) $\sqrt{N_k}\left\{\hat{\Lambda}_{T_k}(t \mid \boldsymbol{Z}_k) - \Lambda_{T_k}(t \mid \boldsymbol{Z}_k)\right\} \xrightarrow{d} N(0, \sigma_k^2(t; \boldsymbol{Z}_k))$, *where the variance function* $\sigma_k^2(t; \boldsymbol{Z}_k)$ *is given in (A.6) for the Cox model and (A.7) for the additive hazards model.*

Theorem 3.1 characterizes the uniform convergence, convergence rate, and asymptotic normality of the estimator $\hat{\Lambda}_{T_k}(t \mid \boldsymbol{Z}_k)$, thus demonstrating its validity. Accordingly, the use of $\hat{S}_{T_k}(t|\boldsymbol{X}_k, A_k) = \exp\{-\hat{\Lambda}_{T_k}(t|\boldsymbol{X}_k, A_k)\}$ as an estimator for $S_{T_k}(t|\boldsymbol{X}_k, A_k)$ is also reasonable. Therefore, we replace $S_{T_k}$ in the Q-function in (4) with $\hat{S}_{T_k}$, resulting in a feasible Q-function as follows:

$$\hat{Q}^\pi(\boldsymbol{x}_t, a_t) = -\mathbb{E}^{P_0, \pi}\left[\sum_{k=t}^{\infty} \gamma^{k-t}\hat{\Lambda}_{T_k}\left(\alpha_k G_k|\boldsymbol{X}_k, A_k\right) \middle| \boldsymbol{X}_t = \boldsymbol{x}_t, A_t = a_t\right]. \tag{8}$$

Through the lower bound of $\log \mathsf{P}^\pi$, we transform the problem of maximizing the probability that the duration between consecutive events exceeds a meaningful threshold into maximizing a discounted sum of estimated cumulative hazards. This enables the direct application of standard RL algorithms to determine the optimal policy.

Specifically, we optimize $\hat{Q}^\pi$ in (8) using the soft-update Deep Q-Network (DQN) algorithm (20) for discrete action spaces and the Deep Deterministic Policy Gradient (DDPG) algorithm (20) for continuous action spaces. The detailed algorithms for obtaining the optimal policy are provided in Appendix A.6 due to space limitations.

## 4 Simulation Studies

In this section, we conduct numerical studies to assess the finite-sample performance of the proposed method and demonstrate its advantages. Specifically, our aim is to address the following research questions (RQs). **RQ1:** How does the proposed decision-making strategy compare with existing approaches in terms of improving the total recurrent-event duration? **RQ2:** Does our method exhibit stable and consistent performance across various distributions of $T_{i,k}$? We first outline the simulation setups and subsequently address the aforementioned questions through extensive numerical experiments.

### 4.1 Simulation Setups

**Setting 1. Linear transition and discrete actions.** For the initial stage ($k = 1$), we generate $\boldsymbol{X}_{i,1}$ from a standard normal distribution $\mathcal{N}(\boldsymbol{0}_2, \boldsymbol{I}_2)$. For each stage, the available actions are chosen from $\mathcal{A} = \{0, 1\}$. $\boldsymbol{X}_{i,k+1}$ evolves according to the following iterative formula,

$$\boldsymbol{X}_{i,k+1} = \begin{bmatrix} 3\left(2A_{i,k} - 1\right)/4 & 0 \\ 0 & 3\left(1 - 2A_{i,k}\right)/4 \end{bmatrix}\boldsymbol{X}_{i,k} + \boldsymbol{\epsilon}_{i,k}, \text{ for } k = 1, 2, \ldots, \bar{K} - 1,$$

where $\boldsymbol{\epsilon}_{i,k} \overset{i.i.d}{\sim} \mathcal{N}\left(\boldsymbol{0}_2, \boldsymbol{I}_2/4\right)$. Similar simulation setups can be found in Shi et al. (37) and Chen et al. (8). We let $\theta_{i,k} = \exp\{(2A_{i,k} - 1)\boldsymbol{X}_{i,k}^\top\boldsymbol{\nu}\}$, where $\boldsymbol{\nu} = (2, -1)^\top$.

**Setting 2. nonlinear transition and continuous actions.** For the initial stage ($k = 1$), we generate $\boldsymbol{X}_{i,1}$ from a standard normal distribution $\mathcal{N}(\boldsymbol{0}_{50}, \boldsymbol{I}_{50})$. For each stage, the available actions are chosen from $\mathcal{A} = [-1, 1]$. $\boldsymbol{X}_{i,k+1}$ evolves according to the following iterative formula,

$$\boldsymbol{X}_{i,k+1} = 3A_{i,k}\boldsymbol{X}_{i,k}/4 + \sin(\boldsymbol{X}_{i,k}) + \cos(\boldsymbol{X}_{i,k}) + \boldsymbol{\epsilon}_{i,k}, \text{ for } k = 1, 2, \ldots, \bar{K} - 1,$$

where $\boldsymbol{\epsilon}_{i,k} \overset{i.i.d}{\sim} \mathcal{N}(\boldsymbol{0}_{50}, \mathbf{I}_{50}/4)$. We let $\theta_{i,k} = \exp\{(-A_{i,k} \cdot \boldsymbol{X}_{i,k}^T\boldsymbol{\beta})\}$, where $\boldsymbol{\beta} \in \mathbb{R}^{50}$ is a vector whose first three elements are generated from the standard normal distribution, and whose remaining elements are 0.

In Setting 1 and Setting 2, we consider total stages $\bar{K} = 20$. $A_{i,k}$ is selected by maximizing the proposed Q-function in (8), and is chosen from $\mathcal{A}$ with a decreasing greedy probability. We set $G_k = 7$, and the duration $T_{i,k}$ is generated according to $T_{i,k} = \min\{B_{i,k}, G_k\}$, where $B_{i,k} \sim$ Gamma$(1/\theta_{i,k}, \theta_{i,k})$. We let $C_i$ follow the uniform distribution $\mathcal{U}(0, 7 \times 10)$, independent of

$T_{i,k}, \boldsymbol{X}_{i,k}$. We set the pre-specified threshold $\alpha_k = 1/14$. To simulate the real-world uncertainty, we consider transition kernels with stochastic perturbations. We optimize the objective function defined in (8) using the soft-update DQN algorithm for Setting 1 and the DDPG algorithm for Setting 2. The implementation code is available at `https://github.com/fjqfengjianqi/NIPS2025-RL-for-Survival`.

## 4.2 Methods and Metrics

**Methods**. Our investigation of the aforementioned RQs is based on the comparison of the following several methods:

- **Cox:** our proposed method that uses the Cox model (10) to estimate the survival function.
- **Aah:** our proposed method that uses the additive hazards model (1) to estimate the survival function.
- **Baseline:** the optimal treatment policy proposed by Liu et al. (23). Specifically, their optimal treatment policy is determined by optimizing the total recurrent-event duration $\sum_{k=1}^{\bar{K}} T_k$ via finite-stage dynamic treatment regimes. The corresponding objective function, derived from Liu et al. (23), is provided in Appendix B.8.

**Metrics**. We conduct $M = 50$ epochs with a sample size of $N$, each progressing from stage 1 to stage $\bar{K} = 20$, with different methods updating their policies as the experiments progressed. The experiments are repeated for $S = 10$ different seeds. For each seed $s = 1, \ldots, S$ and epoch $m = 1, \ldots, M$, we record $T_{i,k}^{(s,m)}$ and $\theta_{i,k}^{(s,m)}$ at each stage $k$. We then predict the values of $T_{i,k}$ and $\theta_{i,k}$ at epoch $m$ derived from simulations using the Cox, Aah, and Baseline methods, denoted as $\hat{T}_{i,k}^{(m)} = \sum_{s=1}^{S} T_{i,k}^{(s,m)}/S$ and $\hat{\theta}_{i,k}^{(m)} = \sum_{s=1}^{S} \theta_{i,k}^{(s,m)}/S$, respectively. In this paper, we employ the following metrics to evaluate the effectiveness and robustness of the proposed method:

- Average Recurrent-Event Duration (**ARED**): $\text{ARED}^{(m)} = \sum_{i=1}^{N} \sum_{k=1}^{\bar{K}} \hat{T}_{i,k}^{(m)}/N$.
- Average Estimated Variance (**AEV**): $\text{AEV}^{(m)} = \sum_{i=1}^{N} \sum_{k=1}^{\bar{K}} \hat{\theta}_{i,k}^{(m)}/N$.
- Average ARED over Last 40 Epochs (**AARED**): $\text{AARED} = \sum_{m=11}^{50} \text{ARED}^{(m)}/40$.
- Average AEV over Last 40 Epochs (**AAEV**): $\text{AAEV} = \sum_{m=11}^{50} \text{AEV}^{(m)}/40$.

## 4.3 Comparisons on the recurrent-event duration (RQ1)

Figure 2 illustrates the changes in ARED and $\log$ ARED with increasing epochs in Setting 1 and Setting 2 for varying sample sizes $N$. The results suggest that our proposed method exhibits faster convergence than the baseline model.

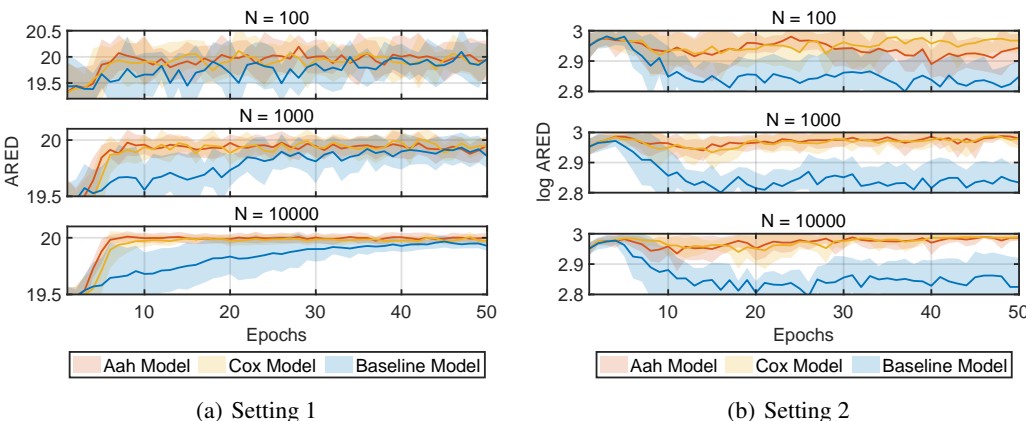

Figure 2: The average recurrent-event duration over epochs with 10 seeds.

Table 1: Mean (Standard Deviation) of AARED under different settings, models and sample sizes $N$

| Setting | $N$ | Aah | Cox | Baseline |
|---------|-----|-----|-----|----------|
| Setting 1 | 100 | **19.96 (0.006)** | 19.93 (0.007) | 19.80 (0.021) |
| | 1000 | **19.94 (<0.001)** | 19.94 (0.001) | 19.83 (0.008) |
| | 10000 | **20.00 (<0.001)** | 19.98 (<0.001) | 19.87 (0.006) |
| Setting 2 | 100 | 18.88 (0.125) | **19.19 (0.086)** | 17.17 (0.074) |
| | 1000 | **19.52 (0.047)** | 19.45 (0.057) | 17.06 (0.074) |
| | 10000 | **19.57 (0.055)** | 19.54 (0.072) | 17.16 (0.080) |

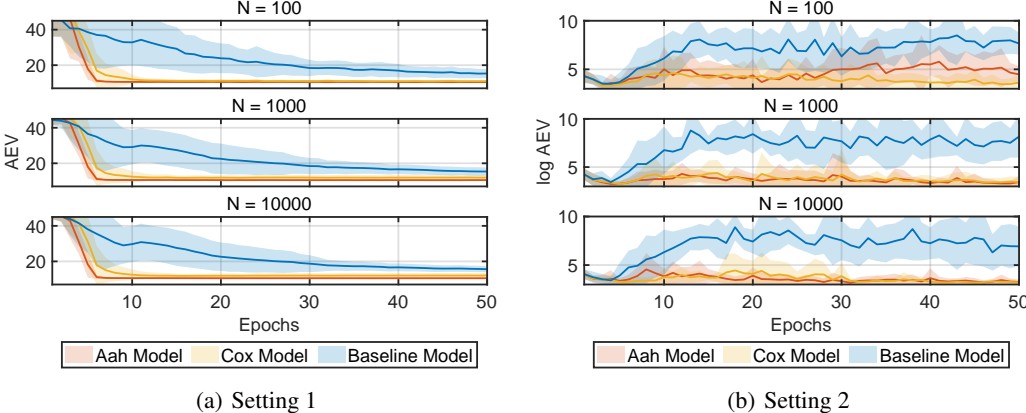

(a) Setting 1          (b) Setting 2

Figure 3: The average estimated variance over epochs with 10 seeds.

Table 1 presents a comparison of the AARED values for different models with varying sample sizes $N$. From Table 1, we can observe that as the sample size increases, the variance of AARED decreases. At the same time, our model consistently achieves a higher AARED compared to the baseline model. Under Setting 1, the maximum improvement is 0.8% ($N = 100$), while under Setting 2, the maximum improvement reaches 14.4% ($N = 1000$). This demonstrates that our method achieves faster convergence and yields higher recurrent-event durations.

### 4.4 Stability of the proposed strategy (RQ2)

Figure 3 illustrates the relationship between the AEV and $\log$ AEV and the number of epochs for Setting 1 and Setting 2, with results shown for various sample sizes $N$. It can be observed that our method leads to a faster decrease and stabilization of the variance.

Table 2 presents a comparison of the AAEV values for different models with varying sample sizes $N$. From Table 2, we can observe that as the sample size increases, the variance of AEV decreases. Moreover, our model consistently achieves a smaller variance compared to the baseline model, with a maximum reduction of 49% ($N = 100$) in Setting 1 and 99.89% ($N = 10000$) in Setting 2.

Table 2: Mean (Standard Deviation) of AAEV under different settings, models and sample sizes $N$

| Setting | $N$ | Aah | Cox | Baseline |
|---------|-----|-----|-----|----------|
| Setting 1 | 100 | **10.53 (0.001)** | 11.11 (0.016) | 20.70 (27.33) |
| | 1000 | **10.54 (<0.001)** | 11.95 (0.005) | 20.00 (19.47) |
| | 10000 | **10.54 (<0.001)** | 12.00 (<0.001) | 20.18 (20.85) |
| Setting 2 | 100 | 2533 ($1 \times 10^8$) | **224 ($2 \times 10^5$)** | 2251 ($3 \times 10^9$) |
| | 1000 | **202 ($4 \times 10^5$)** | 441 ($3 \times 10^6$) | 41189 ($2 \times 10^{10}$) |
| | 10000 | **49 ($2 \times 10^3$)** | 181 ($1 \times 10^5$) | 43657 ($3 \times 10^{10}$) |

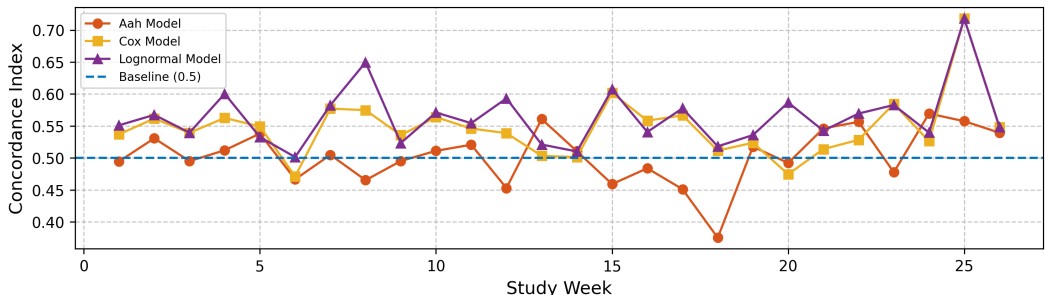

Figure 4: The values of the concordance index fitted for each stage under different models.

Figure 2, 3 and Table 1,2 together demonstrate that the proposed strategy not only maintains effectiveness but also exhibits high robustness. This indicates that the recurrent-event duration achieved by our method is not only higher than that of the baseline method but also shows significantly smaller variance fluctuations. Additional simulation results are provided in Appendix A.1 and A.4.

## 5 Real Data Analysis

In this section, we apply the proposed method to analyze decision-making strategies based on the 2018 Intern Health Study (IHS). IHS was a microrandomized trial (MRT) designed to evaluate the impact of app-based push notifications on physical and mental health outcomes. Over a 26-week period, participants were re-randomized weekly to receive activity suggestions.

**Data description**. The data contains weekly records for each participant, whether a message was sent, the duration of daily sleep, the count of steps and the mood scores. We define the daily average duration of sleep and the step count for the $i$-th participant in the $k$-th week as the state variable $\boldsymbol{X}_{i,k}$, the binary indicator of message sending as the decision action $A_{i,k}$, and the cumulative mood score as the reward $T_{i,k}$. Let stage duration $G_k = 70$, representing the maximum possible mood score per week. Since mood scores are not recorded daily within a week, instances with incomplete mood data are treated as censored, denoted by $\Delta_{i,k} = 0$. After data manipulation, the final dataset includes 1,176 participants followed for up to 26 weeks, with a censoring rate of 87.8%.

**Analysis**. Because the action space is discrete, we use an offline soft-update DQN method to determine the optimal policy $\pi$, with $\bar{K} = 26$. We analyze the fitting performance of different survival models, i.e., the Aah model, the Cox model and the Log-Normal accelerated failure time (AFT) model (as shown in Appendix A.5), to the dataset. Furthermore, given the optimal policy $\pi$, we calculate the average probability that the high mood score exceeds a threshold of the stage duration $\alpha_k G_k$ by

$$\bar{\mathsf{P}}^\pi = \frac{\sum_i \hat{\mathsf{P}}^\pi(\boldsymbol{x}_{i,1})}{1176}, \text{ where } \hat{\mathsf{P}}^\pi(\boldsymbol{x}_{i,t}) = \mathbb{E}^\pi \left[ \prod_{k=t}^{\bar{K}} \hat{S}_{T_{i,k}}(\alpha_k G_k | \boldsymbol{X}_{i,k}, A_{i,k}) \middle| \boldsymbol{X}_{i,t} = \boldsymbol{x}_{i,t} \right].$$

**Results**. Figure 4 presents the concordance index (C-index) for each model at each stage. C-index values are between 0 and 1; values greater than 0.5 indicate better than random discrimination. As shown in Figure 4, the Log-Normal AFT model demonstrates better discriminative performance than the Aah and Cox models.

We denote the average probability derived from the Aah model, the Cox model and the baseline model as $\bar{\mathsf{P}}^{\pi^A}$, $\bar{\mathsf{P}}^{\pi^C}$, and $\bar{\mathsf{P}}^{\pi^B}$, respectively. We then present the **log probability difference** $(\log \bar{\mathsf{P}}^{\pi^A} - \log \bar{\mathsf{P}}^{\pi^B})$ for the Aah model and $(\log \bar{\mathsf{P}}^{\pi^C} - \log \bar{\mathsf{P}}^{\pi^B})$ for the Cox model across varying $\alpha_k$ settings. As shown in Figure 5, our model outperforms the baseline model across all $\alpha_k$ values, achieving higher survival probabilities. The optimal performance occurs at $\alpha_k = 1$, where the probability improved by up to 35%. Additional analyses using different metrics are provided in Appendix A.5.

To conclude, our proposed model formulates policies by maximizing the probability that the high-mood duration in each stage exceeds a threshold, achieving higher probabilities of high mood duration

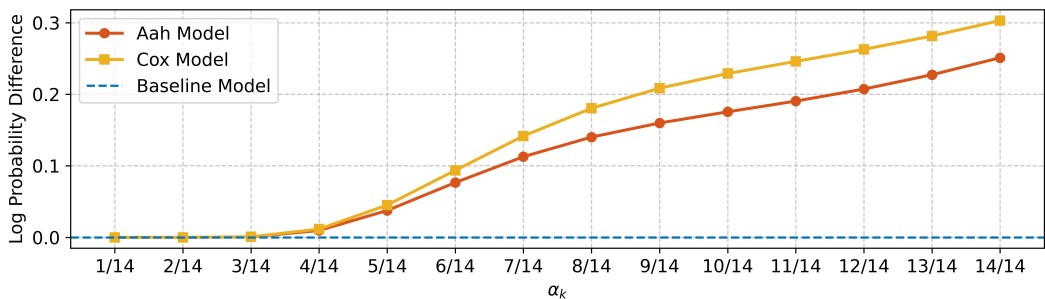

Figure 5: Comparison of log survival probability difference across policies with different $\alpha_k$.

compared to traditional RL strategies that maximize expected rewards. Additionally, the proposed method dynamically adapts the policies to individual preferences in different values of $\alpha_k$.

# 6 Conclusion

This paper presents a novel RL framework for survival analysis tailored to alternating recurrent event data, addressing the limitations of traditional methods designed for time-to-event data and expectation maximization. We propose a probability-based objective that maximizes the probability that recurrent-event durations exceed a given threshold, and reformulate the task as a standard RL problem by optimizing a lower bound of this objective. We provide theoretical guarantees for the equivalence of the optimal policy, as well as for the consistency and asymptotic normality of the estimated cumulative hazard functions. Simulations and real-data experiments demonstrate faster convergence, lower variance, and higher event-duration probabilities of our proposed method compared to existing traditional RL-based methods.

## Acknowledgments and Disclosure of Funding

This work was partly supported by the National Natural Science Foundation of China (12401366, 12301348, 12371292), the Natural Science Foundation of Shandong Province (ZR2023QA086), and the Shandong University Future Plan for Young Scholars. The authors are also grateful for the contributions of the researchers, administrators, and participants involved in the Intern Health Study (NCT03972293).

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

# A    Additional Data Analysis

This section provides supplementary data and analysis for the simulation study in Section 4. We conducted $M = 50$ epochs with a sample size of $N$, each progressing from stage 1 to stage $\bar{K} = 20$, with different methods updating their policies as the experiments progressed. The experiments were repeated for $S = 10$ different seeds. For each seed $s = 1, \ldots, S$ and epoch $m = 1, \ldots, M$, we recorded $T_{i,k}^{(s,m)}$ and $\theta_{i,k}^{(s,m)}$ at each stage $k$. We then predicted the values of $T_{i,k}$ and $\theta_{i,k}$ at epoch $m$ derived from simulations using the Cox, Aah, and Baseline methods, denoted as $\hat{T}_{i,k}^{(m)} = \sum_{s=1}^{S} T_{i,k}^{(s,m)}/S$ and $\hat{\theta}_{i,k}^{(m)} = \sum_{s=1}^{S} \theta_{i,k}^{(s,m)}/S$, respectively.

The following metrics are used to evaluate the effectiveness and robustness of the proposed method:

- Average Recurrent-Event Duration (**ARED**): $\text{ARED}^{(m)} = \sum_{i=1}^{N} \sum_{k=1}^{\bar{K}} \hat{T}_{i,k}^{(m)}/N$.

- Average Estimated Variance (**AEV**): $\text{AEV}^{(m)} = \sum_{i=1}^{N} \sum_{k=1}^{\bar{K}} \hat{\theta}_{i,k}^{(m)}/N$.

- Average ARED over Last 40 Epochs (**AARED**): $\text{AARED} = \sum_{m=11}^{50} \text{ARED}^{(m)}/40$.

- Average AEV over Last 40 Epochs (**AAEV**): $\text{AAEV} = \sum_{m=11}^{50} \text{AEV}^{(m)}/40$.

- Relative Difference of Estimated Variance (**RDEV**):

    - For the Cox method: $\text{RDEV}^{(m)} = (\sum_{i=1}^{N} \hat{\theta}_{i,k}^{C(m)} - \sum_{i=1}^{N} \hat{\theta}_{i,k}^{B(m)})/\sum_{i=1}^{N} \hat{\theta}_{i,k}^{B(m)}$.

    - For the Aah method: $\text{RDEV}^{(m)} = (\sum_{i=1}^{N} \hat{\theta}_{i,k}^{A(m)} - \sum_{i=1}^{N} \hat{\theta}_{i,k}^{B(m)})/\sum_{i=1}^{N} \hat{\theta}_{i,k}^{B(m)}$.

## A.1    Compare with RDEV and $\hat{\theta}_{i,k}$

Here, we present the RDEV values for each stage in Figure 6.

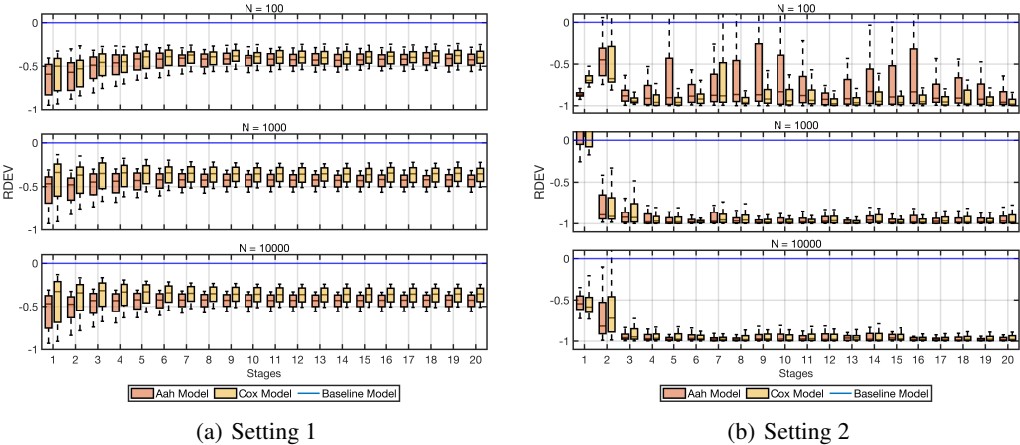

|                  |                  |
| :--------------: | :--------------: |
| (a) Setting 1    | (b) Setting 2    |

Figure 6: Relative difference of estimated variance at each stage under different sample sizes.

Furthermore, we examine the boxplots of $\hat{\theta}_{i,k}^{(m)}$ under varying sample sizes during the final 40 epochs in Figure 7. The results reveal two key observations:

- The variance of $\hat{\theta}_{i,k}^{(m)}$ decreases progressively with increasing sample size.
- This reduction in variance indicates accelerated convergence rates at larger sample sizes.

## A.2    Computation Cost

Here we compare the average time cost per epoch for one seed under Setting 1 in Table 3. The experiments were conducted on a GTX 1650 GPU.

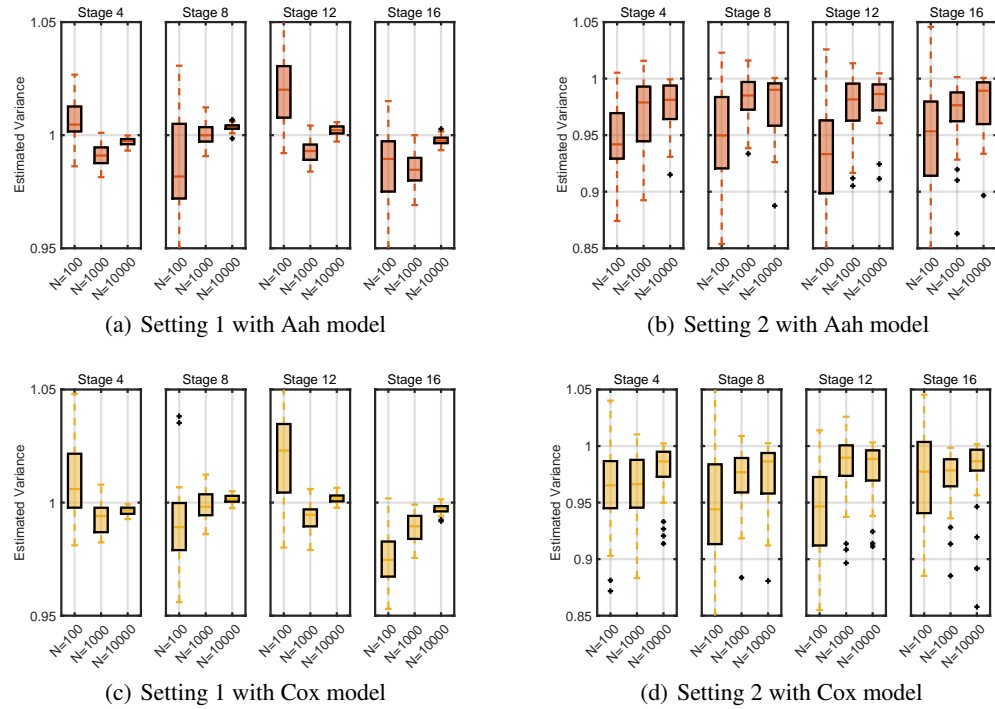

Figure 7: Boxplots of estimated variance $\hat{\theta}_{i,k}^{(m)}$ for each model under different settings across the last 40 epochs as sample size varies.

Table 3: Computation time (in seconds) of different models under varying sample sizes $N$

| Model | $N = 100$ | $N = 1000$ | $N = 10000$ |
|---|---|---|---|
| **Baseline** | 0.22s | 0.77s | 25.50s |
| **Cox** | 0.89s | 4.08s | 35.66s |
| **Aah** | 0.56s | 3.10s | 42.30s |

### A.3   Selection of $\alpha_k$

The selection of $\alpha_k$ is tailored to the specific research question being addressed. For example, in (16)'s study on HIV data, they chose $\alpha_k = 0.5$, focusing on the optimal strategy for maximizing the probability of survival exceeding the median. In our study, we selected $\alpha_k = 1/14$ to maximize the probability of an intern's positive mood exceeding one day per bi-weekly period. We generally advise using an $\alpha_k$ of 0.5 or less, mainly because $\alpha_k$ values near 1 rely heavily on tail probability estimation, which can be unreliable. These results are consistent with the numerical results we obtained from our experiment results. Here we present the AARED and AAEV under different $\alpha_k$ values in Setting 1 in Table 4 and 5 for reference.

### A.4   Compare with Epicare

In this section, we demonstrate the performance of our model on a dynamic treatment regimes RL benchmark, namely the Epicare environment (15). Since the Epicare environment does not include a censoring setting, but the original work defines a maximum number of inquiries after which the process becomes unobservable, we adopt this as our definition of censoring. We define a reinforcement learning environment $\mathcal{E} = \langle \mathcal{X}, \mathcal{A}, P, R \rangle$ to simulate the process of chronic disease treatment. Next, we describe the components of this environment in detail.

Table 4: Mean (Standard Deviation) of AARED under different $\alpha_k$ values and models

| $\alpha_k$ | Aah | Cox | Baseline |
|---|---|---|---|
| 1/14 | 19.9330 (0.0074) | **19.9421 (0.0091)** | |
| 1/7 | **19.9460 (0.0089)** | 19.9228 (0.0130) | |
| 3/14 | 19.1728 (0.0304) | **19.8896 (0.0157)** | |
| 2/7 | 18.9851 (0.0459) | 19.8285 (0.0177) | |
| 5/14 | 18.9972 (0.0438) | 19.8193 (0.0207) | |
| 3/7 | 18.9831 (0.0467) | 19.7919 (0.0192) | |
| 1/2 | 19.0010 (0.0435) | 19.7900 (0.0218) | **19.83 (7.9e-3)** |
| 4/7 | 19.0134 (0.0454) | 19.7571 (0.0295) | |
| 9/14 | 19.0015 (0.0387) | 19.7232 (0.0315) | |
| 5/7 | 19.0007 (0.0433) | 19.6999 (0.0291) | |
| 11/14 | 19.0027 (0.0453) | 19.6703 (0.0515) | |
| 6/7 | 19.0238 (0.0455) | 19.6379 (0.0497) | |
| 13/14 | 19.0107 (0.0434) | 19.6413 (0.0373) | |
| 1 | 19.0591 (0.0465) | 19.6885 (0.0334) | |

Table 5: Mean (Standard Deviation) of AAEV under different $\alpha_k$ values and models

| $\alpha_k$ | Aah | Cox | Baseline |
|---|---|---|---|
| 1/14 | **10.5408 (0.0027)** | 11.9509 (2.3916) | |
| 1/7 | **10.8309 (0.0818)** | 13.9811 (4.4648) | |
| 3/14 | 64.5515 (37.9364) | **17.7416 (12.5118)** | |
| 2/7 | 78.1629 (14.4570) | 20.6589 (16.5979) | |
| 5/14 | 78.5551 (13.8795) | 22.2620 (26.8484) | |
| 3/7 | 78.5710 (14.2459) | 23.5382 (27.2738) | |
| 1/2 | 78.6225 (13.8094) | 24.4125 (35.8999) | **20.00 (19.47)** |
| 4/7 | 78.6280 (14.3543) | 25.0995 (50.6035) | |
| 9/14 | 78.6196 (14.6698) | 27.7541 (67.8201) | |
| 5/7 | 78.4769 (17.6261) | 29.4384 (70.1726) | |
| 11/14 | 78.2809 (19.1925) | 30.3147 (91.0969) | |
| 6/7 | 78.0577 (19.7367) | 32.2957 (104.3780) | |
| 13/14 | 77.9280 (19.0170) | 32.6802 (79.8917) | |
| 1 | 71.9542 (83.7705) | 29.4135 (76.7759) | |

**Disease state:** Let the index set of disease states be $\mathcal{I} = \{1, \ldots, n_d\}$, and let $0$ denote the absorbing remission state. At each stage $t$, the underlying state is

$$d_t \in \{0\} \cup \mathcal{I}, \quad n_d = 16,$$

where $d_t = 0$ represents recovery and is absorbing. For each disease index $k \in \mathcal{I}$, symptoms follow a multivariate Gaussian $\mathcal{N}(\mu_k, \Sigma_k)$, with $\mu_k \in \mathbb{R}^{n_{\text{sym}}}$, $\Sigma_k \in \mathbb{R}^{n_{\text{sym}} \times n_{\text{sym}}}$, $n_{\text{sym}} = 8$. Parameters are randomly generated to induce diversity:

$$\mu_k \sim \mathcal{U}(0, 2)^{n_{\text{sym}}}, \quad \Sigma_k = A_k \operatorname{diag}(\sigma_{k1}^2, \ldots, \sigma_{k\, n_{\text{sym}}}^2) A_k^\top,$$

where $\sigma_{km} \sim \mathcal{U}(1, 2)$ and $A_k \in \mathbb{R}^{n_{\text{sym}} \times n_{\text{sym}}}$ is a random orthogonal matrix, ensuring $\Sigma_k$ is symmetric positive definite.

**Symptom state:** The observable state at stage $t$ is a normalized symptom vector

$$x_t \in [0, 1]^{n_{\text{sym}}}, \ \text{ for } d_t = k \in \mathcal{I}: \ \hat{x}_t \sim \mathcal{N}(\mu_k, \Sigma_k), \ x_t = \sigma(\hat{x}_t),$$

where $\sigma(\cdot)$ is the sigmoid, guaranteeing $x_t \in [0, 1]^{n_{\text{sym}}}$. If $d_t = 0$, the episode has terminated and no further observations are drawn.

**Action Space:** The action space is a finite set of treatments:

$$a_t \in \mathcal{A} = \{0, 1, \ldots, n_{\text{treat}} - 1\}, \quad n_{\text{treat}} = 16.$$

Each action corresponds to a treatment intervention selected by the agent.

**Transition matrix $P^*$:** Disease-to-disease evolution (excluding remission) is governed by a random row-stochastic matrix

$$P = [P_{kl}]_{k,l \in \mathcal{I}} \in \mathbb{R}^{n_d \times n_d}, \text{ where } P_{kl} = \mathbb{P}(d_{t+1} = l \mid d_t = k) \text{ and } \sum_{l \in \mathcal{I}} P_{kl} = 1,$$

constructed as

$$P_{kl} \sim \begin{cases} \mathcal{U}(0.01, 0.2), & \text{with probability } \frac{1}{n_d}, \ k \neq l, \\ 0, & \text{otherwise,} \end{cases} \quad P_{kk} = 1 - \sum_{l \neq k} P_{kl}.$$

This yields a sparse, approximately symmetric graph over $\mathcal{I}$, reflecting realistic limited comorbidity pathways. Remission is handled separately, for $k \in \mathcal{I}$ and action $a$,

$$\mathbb{P}(d_{t+1} = 0 \mid d_t = k, \ a_t = a) = p_k(a), \text{ where } p_k(a) \sim \mathcal{U}(0.8, 1.0).$$

Equivalently, define the augmented action-dependent kernel $P^*$ on $\{0\} \cup \mathcal{I}$:

$$P^*_{k0}(a) = p_k(a), \quad P^*_{kl}(a) = (1 - p_k(a)) P_{kl} \text{ for } l \in \mathcal{I}, \quad P^*_{00}(a) = 1.$$

**Stationary distribution:** The initial disease is sampled from the stationary distribution $\rho = (\rho_1, \cdots, \rho_{n_d})$ of $P$ over $\mathcal{I}$:

$$\rho^\top P = \rho^\top, \quad \sum_{k \in \mathcal{I}} \rho_k = 1, \quad d_0 \sim \rho.$$

(Note that remission $d_t = 0$ is excluded here since it is an absorbing terminal state.)

**Reward function:** The stage-wise reward balances cost, remission benefit, and adverse-event risk:

$$R_t = R_{\text{cost}}(a_t) + R_{\text{remission}}(d_{t+1}) + R_{\text{adverse}}(\boldsymbol{x}_{t+1}),$$

with

$$R_{\text{cost}}(a_t) = -c(a_t), \quad c(a_t) \sim \mathcal{U}(1, 5),$$

$$R_{\text{remission}} = \begin{cases} +r_{\text{rem}}, & d_{t+1} = 0, \\ 0, & \text{otherwise,} \end{cases} \quad r_{\text{rem}} = 64,$$

$$R_{\text{adverse}} = \begin{cases} -r_{\text{adv}}, & \max(\boldsymbol{x}_{t+1}) > 0.999, \\ 0, & \text{otherwise,} \end{cases} \quad r_{\text{adv}} = 64.$$

**Termination:** We assume that the process terminates once the patient enters the remission state, i.e., when $d_t = 0$. In addition, we impose a maximum follow-up horizon $K_{\max} = 8$. If termination has not occurred by $t = K_{\max}$, the trajectory is regarded as right-censored, meaning that the observation is incomplete. Formally, we define the event time as $\bar{K} = \min\{t \mid d_t = 0\}$, and introduce the event indicator $\Delta_t = 1_{\{t < \bar{K}\}} \times 1_{\{t \leq K_{\max}\}}$, which equals 1 if the event (remission) occurs before censoring, and 0 otherwise.

We defined $(\boldsymbol{x}_t, a_t, R_t, \Delta_t)$ for $t = 1, \ldots, K_{\max}$, where $R_t$ is analogous to $Y_t$ in the main text, both representing the reward. An offline dataset with sample size $N = 1000$ was generated. Using 10 different random seeds for network initialization, we trained multiple models and evaluated a variety of policies. For each policy, we report the cumulative reward as well as the final censoring rate. The following four approaches are compared:

- **Cox:** our proposed method utilizing the survival function estimated from the Cox proportional hazards model (10).

- **Aah:** our proposed method utilizing the survival function estimated from Aalen's additive hazards model (1).

- **Km:** the optimal treatment policy proposed by Liu et al. (23), which maximizes the expected value via Kaplan–Meier estimation. The corresponding objective function, derived from the method of Liu et al.'s (2023), is provided in Appendix B.8.

- **RL:** classical reinforcement learning methods that maximize reward without accounting for $\Delta$, consistent with the objective proposed in (15).

We set the number of training epochs to 50 and the number of evaluation episodes to 200. Since we require $Y_t > 0$, all $R_t$ values were normalized across individuals at each stage prior to estimation and training. We computed the mean and variance of the cumulative reward ($\sum_{t=1}^{K_{\max}} \sum_{i=1}^{N} R_{it}/N$) and censoring rate ($1 - \sum_{i=1}^{N} \Delta_{iK_{\max}}/N$) across all 10 random seeds, including results from our proposed methods under different values of $\alpha_k$. These results, reported in Table 6 and Table 7, indicate that our methods consistently achieve higher rewards and lower censoring rates, thereby demonstrating superior performance in the Epicare environment.

Table 6: Mean (Standard Deviation) of Cumulative Reward Under different model and $\alpha_k$

| $\alpha_k$ | Aah | Cox | Km | RL |
|---|---|---|---|---|
| 0.1 | **11.0323(4.9149)** | 8.9952(3.5328) | | |
| 0.2 | **11.0323(4.9149)** | 6.5327(4.3448) | | |
| 0.3 | **10.0921(3.4005)** | 7.2183(4.1138) | | |
| 0.4 | **10.0921(3.4005)** | 7.5531(4.6677) | | |
| 0.5 | **11.0028(4.3290)** | 8.5168(3.0017) | -15.8804(1.1859) | **6.2507(3.3373)** |
| 0.6 | **11.1656(3.8367)** | 7.3527(4.5401) | | |
| 0.7 | **11.1656(3.8367)** | 5.4090(2.7325) | | |
| 0.8 | **10.7549(4.7615)** | 7.4012(3.6093) | | |
| 0.9 | **10.7549(4.7615)** | 8.2964(4.4176) | | |
| 1.0 | -12.7179(3.0012) | 0.6448(6.0716) | | |

Table 7: Mean (Standard Deviation) of Censoring Rate Under different model and $\alpha_k$

| $\alpha_k$ | Aah | Cox | Km | RL |
|---|---|---|---|---|
| 0.1 | **0.5050(0.0575)** | 0.5405(0.0452) | | |
| 0.2 | **0.5050(0.0575)** | 0.5665(0.0525) | | |
| 0.3 | **0.5125(0.0453)** | 0.5605(0.0494) | | |
| 0.4 | **0.5125(0.0453)** | 0.5540(0.0602) | | |
| 0.5 | **0.5055(0.0535)** | 0.5450(0.0398) | 0.9610(0.0139) | 0.6410(0.0431) |
| 0.6 | **0.5060(0.0444)** | 0.5585(0.0539) | | |
| 0.7 | **0.5060(0.0444)** | 0.5785(0.0369) | | |
| 0.8 | **0.5120(0.0586)** | 0.5555(0.0440) | | |
| 0.9 | **0.5120(0.0586)** | 0.5455(0.0469) | | |
| 1.0 | 0.7975(0.0354) | **0.6365(0.0603)** | | |

## A.5 Real Data Analysis Model

We then presented the **relative probability** $(\bar{\mathsf{P}}^{\pi^A} - \bar{\mathsf{P}}^{\pi^B})/\bar{\mathsf{P}}^{\pi^B}$ for the Aah model and $(\bar{\mathsf{P}}^{\pi^C} - \bar{\mathsf{P}}^{\pi^B})/\bar{\mathsf{P}}^{\pi^B}$ for the Cox model across varying $\alpha_k$ settings in Figure 8. As shown in Figure 8, our model outperformed the baseline across all $\alpha_k$ values, achieving higher survival probabilities. At the same time, we observe that the Cox model also provides a satisfactory fit. Below, we present the results of estimating $\mathsf{P}^{\pi}$ in (1) under the Cox model, then we show the log probability differences in Figure 9 and the relative probabilities in Figure 10.

**Log Normal AFT model** assumes that the natural logarithm of the survival time $T_k$ follows a normal distribution: $\log(T_k) \sim \mathcal{N}(\mu(z_k), \sigma^2)$, where: The location parameter $\mu(z)$ is a linear combination of the covariates $z = [z_1, z_2, \ldots, z_n]$: $\mu(z) = a_0 + a_1 z_1 + a_2 z_2 + \cdots + a_n z_n$ Here, $a_0, a_1, \ldots, a_n$ are the regression coefficients to be estimated, reflecting the impact of the covariates $z$ on the mean of the logarithm of the survival time. The scale parameter $\sigma$ is a fixed constant (independent of covariates), describing the standard deviation of $\log(T)$ and controlling the degree of dispersion of the distribution. The model estimates parameters by maximizing the log-likelihood function.

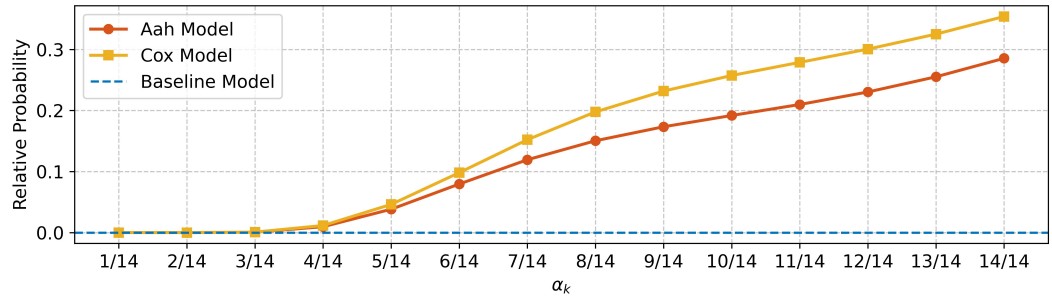

Figure 8: Comparison of relative survival probability improvements across policies with different $\alpha_k$ under Log Normal AFT model estimation.

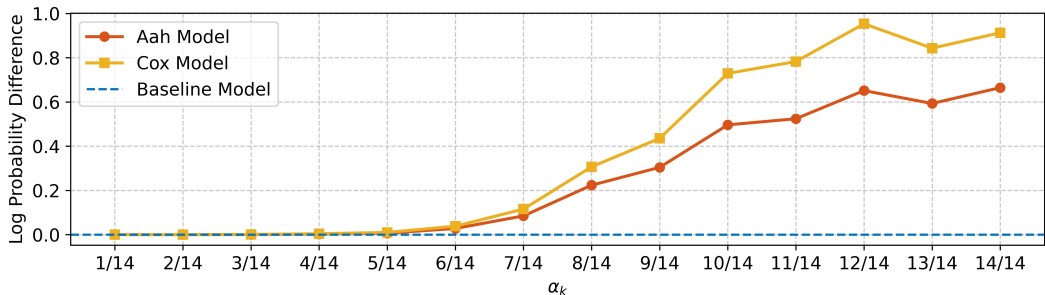

Figure 9: Comparison of log survival probability differences across policies with different $\alpha_k$ under Cox model estimation.

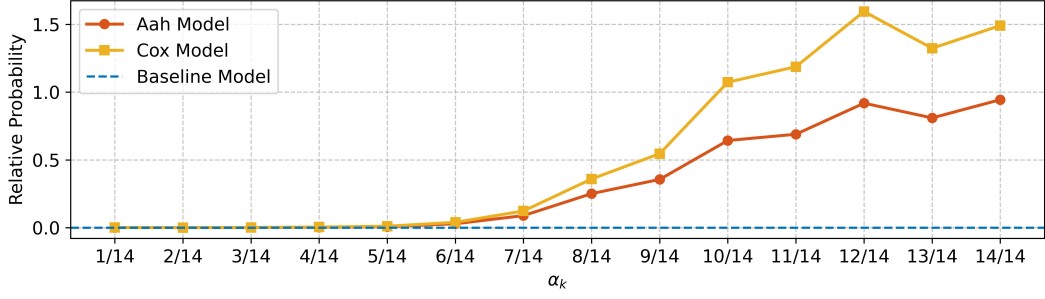

Figure 10: Comparison of relative survival probability across policies with different $\alpha_k$ under Cox model estimation.

## A.6 Algorithm for Recurrent Event Data

We use the Deep Deterministic Policy Gradient (DDPG) algorithm (20) to optimize the objective function defined in (8) with a continuous action space and use soft-update Deep Q-Network (DQN) to update the discrete action space. It is a model - free, off - policy algorithm that employs an initial network $Q$ and a target network $Q'$. Here we update the value of the Q-function using the optimal Bellman operator and compute the loss (temporal difference error) to adjust the weights of the network $Q$. Subsequently, we perform a soft update to the weights of the target network $Q'$.

**For discrete action space**. We define greedy optimal policy $\pi^*$ as any policy that selects the action that produces the highest $\hat{Q}$ value. Specifically, for any $\boldsymbol{x} \in \mathcal{X}$, $\pi^*(\cdot \mid \boldsymbol{x})$ satisfies $\pi^*(a \mid \boldsymbol{x}) = 0$ if $\hat{Q}^{\pi^*}(\boldsymbol{x}, a) \neq \max_{a' \in \mathcal{A}} \hat{Q}^{\pi^*}(\boldsymbol{x}, a')$. We denote the Bellman optimality operator as $\mathcal{T}$. To maximize the Q-function, following the strategy of Bellman's equation (3), an estimate of $\hat{Q}^\pi$ can be obtained via dynamic programming using the operator $\mathcal{T}$ as follows:

$$\mathcal{T}\hat{Q}(\boldsymbol{X}_k, A_k) := -\mathbb{E}\hat{\Lambda}_{T_k}(\alpha_k G_k \mid \boldsymbol{X}_k, A_k) + \gamma \mathbb{E} \max_{A_{k+1} \in \mathcal{A}} \hat{Q}(\boldsymbol{X}_{k+1}, A_{k+1}). \tag{A.1}$$

It is shown in Appendix B.4 that updating the Q-function using the Bellman operator $\mathcal{T}$ can converge to a unique Q-function.

**For continuous action space**. According to DDPG (20), we define the Policy Function: $\mu(\theta): \mathcal{S} \rightarrow \mathcal{A}$, a parameterized function (e.g., neural network) that maps states to actions. For a fixed policy $\mu(\theta)$, the Q-function follows the Bellman evaluation equation: $Q^{\mu(\theta)}(\boldsymbol{X}_t, A_t) = \mathbb{E}^{P_0}[-\hat{\Lambda}_{T_t}(\alpha_t G_t | \boldsymbol{X}_t, A_t) + \gamma Q^{\mu(\theta)}(\boldsymbol{X}_{t+1}, \mu(\theta)(\boldsymbol{X}_{t+1}))]$. To optimize $\mu(\theta)$, we maximize the expected return $J(\theta) = \mathbb{E}[Q^{\mu(\theta)}(\boldsymbol{X}_t, A_t)]$. Then, parameter updates are performed using the chain rule in DDPG.

One of the key features of DDPG and soft-update DQN is the use of soft updates for the target networks. Rather than directly copying the weights from the current network to the next network, they gradually updates the parameters. This approach enhances learning stability by smoothing out parameter updates, thus preventing large oscillations in parameter values. The proposed estimation procedure can be implemented as outlined in Algorithm 1. For the neural network in simulation, we employ a RL agent with the following hyperparameters: a batch size of 32, a replay buffer of 6400, a learning rate of 0.001, a soft update $\tau$ of 0.01, and a discount factor $\gamma$ of 0.99.

## A.7 Supplementary Experiment

Under identical assumptions for Setting 1 in Section 4, we vary $\bar{K}$ to compare the AAEV and AARED metrics across different models, using a sample size of $N = 1000$ for this analysis. The performance of AAEV and AARED is presented in Table 8 and Table 9, respectively. Because the censoring rates under different $\bar{K}$ are also different, we present the changes in censoring rates ( $1 - \sum_{i=1}^{N_1} \prod_{k=1}^{\bar{K}} \Delta_{i,k}/N_1$ ) under different Models in Table 10. The results demonstrate that our model consistently outperforms the baseline model across different values of $\bar{K}$ while maintaining comparable censoring rates.

Table 8: Mean (Standard Deviation) of AAEV under different $\bar{K}$ when $N = 1000$ in Setting 1

| Model | $\bar{K}$ | | | | |
|---|---|---|---|---|---|
| | 10 | 20 | 30 | 40 | 50 |
| **Aah** | **8.45(1.14)** | **13.14(0.68)** | **18.45(0.59)** | **23.99(0.59)** | **29.57(0.48)** |
| **Cox** | 9.94 (2.62) | 15.14 (2.69) | 21.13 (2.97) | 27.29 (3.52) | 33.65 (4.91) |
| **Baseline** | 16.46 (6.16) | 23.29 (6.53) | 29.97 (6.63) | 37.20 (7.35) | 43.81 (7.54) |

# B Definitions and Proofs of Theoretical Results

## B.1 Proof of Lemma 2.3

We first propose the following Lemma B.1 (the proof is provided in Section B.2).

---

**Algorithm 1** DDPG for Recurrent Event Data (Continuous and Discrete action space)

---

Randomly initialize action - value network $Q$ with weights $\theta^Q$.

Initialize target network $Q'$ with weights $\theta^{Q'} \leftarrow \theta^Q$.

**For continuous actions**:

– Initialize actor network $\mu$ with weights $\theta^\mu$.

– Initialize target actor network $\mu'$ with weights $\theta^{\mu'} \leftarrow \theta^\mu$.

– Initialize replay buffer $R$, set soft update weight $\tau$, exploration noise $\epsilon$, and $\alpha_k$.

**for** episode = 1 to $M$ **do**

  **For discrete actions**:

  – Initialize a random process $\epsilon$ as explore probability for discrete actions.

  **For continuous actions**:

  – Initialize a noise process $\mathcal{N}$ for continuous action exploration.

  Receive initial observation state $\{\mathbf{X}_{i,1}\}_{i=1}^{N_1}$.

  **for** $k = 1$ to $\bar{K}$ **do**

    **For continuous actions**:

    – With probability $\epsilon$, select a random action $A_{i,k}$ from the continuous action space perturbed by noise $\mathcal{N}$.

    – Otherwise, select $A_{i,k} = \mu(\mathbf{X}_{i,k}; \theta^\mu) + \mathcal{N}$.

    **For discrete actions**:

    – With probability $\epsilon$, select a random discrete action $A_{i,k}$.

    – Otherwise, select $A_{i,k} = \arg\max_A Q(\mathbf{X}_{i,k}, A; \theta^Q)$.

    Execute action $A_{i,k}$ in emulator and observe $Y_{i,k}$, $\Delta_{i,k}$ and image $\mathbf{X}_{i,k+1}$.

    Store transition $(\boldsymbol{X}_{i,t}, A_{i,t}, Y_{i,t}, \Delta_{i,t}, \alpha_t, \boldsymbol{X}_{i,t+1})$ in $R$.

    Sample a random minibatch of $(\boldsymbol{X}_{i,t}, A_{i,t}, Y_{i,t}, \Delta_{i,t}, \alpha_t, \boldsymbol{X}_{i,t+1})$ from $R$.

    Estimate $\hat{\Lambda}_{T_{i,t}}(\cdot|\boldsymbol{X}_{i,t}, A_{i,t})$ with $\{\boldsymbol{X}_{i,t}, A_{i,t}, Y_{i,t}, \Delta_{i,t}\}_{i=1}^{N_t}$ using Cox or Aah model.

    Calculate $R_{i,t} = -\hat{\Lambda}_{T_{i,t}}(\alpha_t G_t|\boldsymbol{X}_{i,t}, A_{i,t}) + \gamma \max_{A'} Q'(\boldsymbol{X}_{i,t+1}, A'; \theta^{Q'})$.

    **Update critic network**:

    Update $\theta^Q$ by minimizing the loss: $L = \frac{1}{N_t} \sum_{i=1}^{N_t} (R_{i,t} - Q(\boldsymbol{X}_{i,t}, A_{i,t}; \theta^Q))^2$.

    **Update actor network (only for continuous actions)**:

    – Update $\theta^\mu$ to maximize $Q(\boldsymbol{X}_{i,t}, \mu(\boldsymbol{X}_{i,t}; \theta^\mu); \theta^Q)$.

    **Update target networks**:

    $\theta^{Q'} \leftarrow \tau\theta^Q + (1-\tau)\theta^{Q'}$.

    **For continuous actions**:

    $-\theta^{\mu'} \leftarrow \tau\theta^\mu + (1-\tau)\theta^{\mu'}$.

  **end for**

**end for**

---

Table 9: Mean (Standard Deviation) of AARED under different $\bar{K}$ when $N = 1000$ in Setting 1

| Model | $\bar{K}$ | | | | |
|---|---|---|---|---|---|
| | 10 | 20 | 30 | 40 | 50 |
| **Aah** | **9.92(0.08)** | **19.89(0.10)** | **29.93(0.11)** | **39.91(0.12)** | **49.90(0.14)** |
| **Cox** | 9.91 (0.09) | 19.89 (0.11) | 29.90 (0.12) | 39.90 (0.15) | 49.86 (0.18) |
| **Baseline** | 9.81 (0.14) | 19.78 (0.16) | 29.78 (0.18) | 39.77 (0.22) | 49.74 (0.21) |

Table 10: Mean (Standard Deviation) of Censoring Rates under $\bar{K}$ when $N = 1000$ in Setting 1

| Model | $\bar{K}$ | | | | |
|---|---|---|---|---|---|
| | 10 | 20 | 30 | 40 | 50 |
| **Aah** | 15.1% (0.004) | 29.8% (0.005) | 42.4% (0.004) | 56.8% (0.005) | 71.0% (0.006) |
| **Cox** | 15.0% (0.004) | 29.9% (0.007) | 42.7% (0.006) | 56.8% (0.006) | 71.1% (0.006) |
| **Baseline** | 15.1% (0.005) | 29.6% (0.007) | 42.5% (0.005) | 56.6% (0.007) | 70.6% (0.008) |

**Lemma B.1.** *When Assumptions 2.1 and 2.2 hold, we have*

$$\mathbb{P}\left(T_1 > \alpha_1 G_1, \ldots, T_{\bar{K}} > \alpha_{\bar{K}} G_{\bar{K}} \mid \boldsymbol{X}_1, A_1, \ldots, \boldsymbol{X}_{\bar{K}}, A_{\bar{K}}\right) = \prod_{k=1}^{\bar{K}} S_{T_k}(\alpha_k G_k | \boldsymbol{X}_k, A_k),$$

*where $S_{T_k}(\cdot | \boldsymbol{X}_k, A_k)$ is the conditional survival function of the duration time $T_k$ given $\boldsymbol{X}_k$ and $A_k$.*

Next, we prove Lemma 2.3 as below.

*Proof.* We define $B_k = \{w | T_k(w) > \alpha_k G_k\} \subset \Omega_k$ for $k = 1, \ldots, \bar{K}$, where $\Omega_k$ is the measurable sample space of stage $k$. Then we have:

$$
\begin{aligned}
\mathbb{P}^{P_0, \pi}(B_t, \cdots, B_{\bar{K}} | \boldsymbol{X}_t) &= \mathbb{E}^{P_0, \pi}\left[\mathbb{1}_{(B_t, \cdots, B_{\bar{K}})} \mid \boldsymbol{X}_t\right] \\
&= \mathbb{E}^{P_0, \pi}\left[\mathbb{E}\left[\mathbb{1}_{(B_t, \cdots, B_{\bar{K}})} \mid \boldsymbol{X}_t, A_t, \cdots, \boldsymbol{X}_{\bar{K}}, A_{\bar{K}}\right] \mid \boldsymbol{X}_t\right] \\
&= \mathbb{E}^{P_0, \pi}\left[\mathbb{P}\left(B_t, \cdots, B_{\bar{K}} \mid \boldsymbol{X}_t, A_t, \cdots, \boldsymbol{X}_{\bar{K}}, A_{\bar{K}}\right) \mid \boldsymbol{X}_t\right] \\
&= \mathbb{E}^{P_0, \pi}\left[\prod_{k=t}^{\bar{K}} \mathbb{P}(B_k \mid \boldsymbol{X}_k, A_k) \,\middle|\, \boldsymbol{X}_t\right] \\
&= \mathbb{E}^{P_0, \pi}\left[\prod_{k=t}^{\bar{K}} S_{T_k}(\alpha_k G_k \mid \boldsymbol{X}_k, A_k) \,\middle|\, \boldsymbol{X}_t\right],
\end{aligned}
$$

where the second last equality follows from Lemma B.1. $\qquad\square$

## B.2 Proof of Lemma B.1

*Proof.* Define $B_k = \{w | T_k(w) > \alpha_k G_k\} \subset \Omega_k$, $k = 1, \ldots, \bar{K}$, where $\Omega_k$ is the measurable sample space of stage $k$. Let $O_k = (\boldsymbol{X}_k, A_k)$, then we have

$$
\begin{aligned}
&\mathbb{P}\left(B_1, \ldots, B_{\bar{K}} | O_1, \ldots, O_{\bar{K}}\right) \\
=&\mathbb{P}\left(B_{\bar{K}} | B_1, \ldots, B_{\bar{K}-1}, O_1, \ldots, O_{\bar{K}}\right) \mathbb{P}\left(B_1, \ldots, B_{\bar{K}-1} | O_1, \ldots, O_{\bar{K}}\right) \\
=&\mathbb{P}\left(B_{\bar{K}} | O_{\bar{K}}\right) \mathbb{P}\left(B_1, \ldots, B_{\bar{K}-1} | O_1, \ldots, O_{\bar{K}}\right) \\
=&\mathbb{P}(B_{\bar{K}} | O_{\bar{K}}) \frac{\mathbb{P}(B_1, \ldots, B_{\bar{K}-1} | O_1, \ldots, O_{\bar{K}-1}) \mathbb{P}(O_{\bar{K}} | B_1, \ldots, B_{\bar{K}-1}, O_1, \ldots, O_{\bar{K}-1})}{\mathbb{P}(O_{\bar{K}} | O_1, \ldots, O_{\bar{K}-1})} \\
=&\mathbb{P}(B_{\bar{K}} | O_{\bar{K}}) \mathbb{P}(B_1, \ldots, B_{\bar{K}-1} | O_1, \ldots, O_{\bar{K}-1}) \\
=&\mathbb{P}\left(B_{\bar{K}} | O_{\bar{K}}\right) \mathbb{P}\left(B_{\bar{K}-1} | O_{\bar{K}-1}\right) \mathbb{P}\left(B_1, \ldots, B_{\bar{K}-2} | O_1, \ldots, O_{\bar{K}-2}\right) \\
=&\prod_{k=1}^{\bar{K}} \mathbb{P}(B_k | O_k) \\
=&\prod_{k=1}^{\bar{K}} S_{T_k}(\alpha_k G_k | \boldsymbol{X}_k, A_k).
\end{aligned}
$$

$\qquad\square$

## B.3 Proof of Theorem 2.4

*Proof.* Let $\boldsymbol{x}_k^{P_0}$ be the state at stage $k$ under policy $P_0$. For any deterministic policy $P_0$, by definition, there exists a unique $\boldsymbol{x}_{k+1}^{P_0}$ such that

$$P_0(\boldsymbol{x}_{k+1}^{P_0} \mid \boldsymbol{x}_k^{P_0}, a_k) = 1, \text{ for } k \geq t,$$

where $\boldsymbol{x}_t^{P_0} = \boldsymbol{x}_t$ is the initial state. Define $\Gamma_t^{P_0}$ as the set of all possible trajectories, each consisting of a sequence of states, actions, and rewards from stage $t$ to $\bar{K}$. Specifically,

$$\Gamma_t^{P_0} = \left\{ \left(\boldsymbol{x}_t^{P_0}, a_t, T_t, \boldsymbol{x}_{t+1}^{P_0}, a_{t+1}, T_{t+1}, \ldots, \boldsymbol{x}_{\bar{K}}^{P_0}, a_{\bar{K}}, T_{\bar{K}}\right) \right\}_{a_k \in \mathcal{A}, \text{ for } t \leq k \leq \bar{K}},$$

where $T_k := T_k(\boldsymbol{x}_k^{P_0}, a_k)$ is the reward received at stage $k$ given the state-action pair $(\boldsymbol{x}_k^{P_0}, a_k)$. Then, for any $\gamma_t \in \Gamma_t^{P_0}$, we define the reward along the path $\gamma_t$ as

$$R(\gamma_t) = \prod_{k=t}^{\bar{K}} S_{T_k}(\alpha_k G_k \mid \boldsymbol{x}_k^{P_0}, a_k).$$

Then we can obtain

$$P^\pi(\boldsymbol{x}_t) = \sum_{\gamma_t \in \Gamma_t^{P_0}} \mathcal{P}^\pi(\gamma_t) \cdot R(\gamma_t), \quad V_0^\pi(\boldsymbol{x}_t) = \sum_{\gamma_t \in \Gamma_t^{P_0}} \mathcal{P}^\pi(\gamma_t) \cdot \log R(\gamma_t),$$

where

$$\mathcal{P}^\pi(\gamma_t) = \pi(a_t \mid \boldsymbol{x}_t^{P_0}) \prod_{k=t}^{\bar{K}-1} P_0(\boldsymbol{x}_{k+1}^{P_0} \mid \boldsymbol{x}_k^{P_0}, a_k) \pi(a_{k+1} \mid \boldsymbol{x}_{k+1}^{P_0}) = \prod_{k=t}^{\bar{K}} \pi(a_k \mid \boldsymbol{x}_k^{P_0}).$$

Note that $\sum_{\gamma_t \in \Gamma_t^{P_0}} \prod_{k=t}^{\bar{K}} \pi(a_k \mid \boldsymbol{x}_k^{P_0}) = 1$, therefore we have

$$P^\pi(\boldsymbol{x}_t) = \sum_{\gamma_t \in \Gamma_t^{P_0}} \prod_{k=t}^{\bar{K}} \pi(a_k \mid \boldsymbol{x}_k^{P_0}) \cdot R(\gamma_t) \leq \max_{\gamma_t \in \Gamma_t^{P_0}} R(\gamma_t), \text{ and}$$

$$V_0^\pi(\boldsymbol{x}_t) = \sum_{\gamma_t \in \Gamma_t^{P_0}} \prod_{k=t}^{\bar{K}} \pi(a_k \mid \boldsymbol{x}_k^{P_0}) \cdot \log R(\gamma_t) \leq \max_{\gamma_t \in \Gamma_t^{P_0}} \log R(\gamma_t).$$

Let us define

$$\Gamma_t^{p,*} = \left\{ \gamma_t^p \mid R(\gamma_t^p) = \max_{\gamma_t \in \Gamma_t^{P_0}} R(\gamma_t) \right\}, \text{ and } \Gamma_t^{v,*} = \left\{ \gamma_t^v \mid \log R(\gamma_t^v) = \max_{\gamma_t \in \Gamma_t^{P_0}} \log R(\gamma_t) \right\}.$$

The monotonicity of $\log(x)$ directly implies that $\Gamma_t^{v,*} = \Gamma_t^{p,*}$, which we then denote as $\Gamma_t^*$. Therefore we have

$$P^\pi(\boldsymbol{x}_t) \leq R(\gamma_t^*), \quad V_0^\pi(\boldsymbol{x}_t) \leq \log R(\gamma_t^*), \text{ for } \gamma_t^* \in \Gamma_t^*.$$

Define the policy set

$$\Pi^* = \left\{ \pi \mid \sum_{\gamma_t \in \Gamma_t^*} \prod_{k=t}^{\bar{K}} \pi(a_k \mid \boldsymbol{x}_k^{P_0}) = 1 \right\}.$$

For $P^\pi(\boldsymbol{x}_t)$, we denote $R^* = \max_{\gamma_t \in \Gamma_t^{P_0}} R(\gamma_t)$. Then, for $\gamma_t \notin \Gamma_t^*$, we have $R(\gamma_t) < R^*$. Thus,

$$P^\pi(\boldsymbol{x}_t) = \sum_{\gamma_t \notin \Gamma_t^*} \mathcal{P}^\pi(\gamma_t) \cdot R(\gamma_t) + \sum_{\gamma_t \in \Gamma_t^*} \mathcal{P}^\pi(\gamma_t) \cdot R^*.$$

It follows that $\pi \in \Pi^* \iff \sum_{\gamma_t \in \Gamma_t^*} \mathcal{P}^\pi(\gamma_t) = 1 \iff P^\pi(\boldsymbol{x}_t) = R^*$, i.e., $P^\pi(\boldsymbol{x}_t) = R(\gamma_t^*) \iff \pi \in \Pi^*$. Following a similar proof for $P^\pi(\boldsymbol{x}_t) = R(\gamma_t^*)$, we can demonstrate that $V_0^\pi(\boldsymbol{x}_t) = \log R(\gamma_t^*)$ holds if and only if $\pi \in \Pi^*$. Therefore, we conclude that $\Pi^v = \Pi^p = \Pi^*$.

For the continuous action space, the policy can be expressed as $\pi^\mu(a \mid \boldsymbol{x}) = \delta(a - \mu(\boldsymbol{x}))$, where $\delta(\cdot)$ is the Dirac delta function and $\mu : \mathcal{X} \to \mathcal{A}$ is a deterministic mapping. Theorem 2.4 also holds for the continuous action space. In the following, we provide an alternative and more direct proof, as shown in B.3.1. $\square$

### B.3.1   Proof of Theorem 2.4 for continuous action space

*Proof.* For a deterministic transition kernel $P_0$ and a deterministic policy $\mu$, starting from any initial state $\boldsymbol{x}_t$, the next state is uniquely determined by

$$P_0(\boldsymbol{x}_{k+1}^{P_0} \mid \boldsymbol{x}_k^{P_0}, \mu(\boldsymbol{x}_k^{P_0})) = 1, \text{ for } k \geq t,$$

so that there exists a unique trajectory

$$\gamma_t^\mu = \left(\boldsymbol{x}_t^{P_0}, \mu(\boldsymbol{x}_t^{P_0}), T_t, \boldsymbol{x}_{t+1}^{P_0}, \mu(\boldsymbol{x}_{t+1}^{P_0}), T_{t+1}, \ldots, \boldsymbol{x}_{\bar{K}}^{P_0}, \mu(\boldsymbol{x}_{\bar{K}}^{P_0}), T_{\bar{K}}\right).$$

Along this trajectory, define the reward as

$$R(\gamma_t^\mu) = \prod_{k=t}^{\bar{K}} S_{T_k}(\alpha_k G_k \mid \boldsymbol{x}_k^{P_0}, \mu(\boldsymbol{x}_k^{P_0})).$$

Then the corresponding objective functions simplify to

$$P^\mu(\boldsymbol{x}_t) = R(\gamma_t^\mu), \quad V_0^\mu(\boldsymbol{x}_t) = \log R(\gamma_t^\mu).$$

Define the set of optimal trajectories as

$$\Gamma_t^* = \left\{ \gamma_t^* \mid R(\gamma_t^*) = \max_{\gamma_t} R(\gamma_t) \right\},$$

where the maximization is taken over all possible deterministic policies $\mu$, equivalently over all possible trajectories. Since the logarithm is strictly increasing, maximizing $R(\gamma_t^\mu)$ is equivalent to maximizing $\log R(\gamma_t^\mu)$. Thus,

$$\pi^\mu \in \Pi^p \iff R(\gamma_t^\mu) = \max_{\gamma_t} R(\gamma_t) \iff \log R(\gamma_t^\mu) = \max_{\gamma_t} \log R(\gamma_t) \iff \pi^\mu \in \Pi^v.$$

Therefore,

$$\Pi^v = \Pi^p = \left\{ \pi^\mu(a_k \mid \boldsymbol{x}_k^{P_0}) = \delta(a_k - \mu(\boldsymbol{x}_k^{P_0})) \mid \gamma_t^\mu \in \Gamma_t^*, t \le k \le \bar{K} \right\},$$

which proves the equivalence of the optimal policy sets for the continuous action space. $\qquad\square$

### B.4 Proof of the coverage of $\mathcal{T}$

*Proof.* First we proof a statement: we have $\max_a f(a) - \max_a g(a) \le f(a^*) - g(a^*)$, and if $\max_a g(a) = g(a')$, we have $\max_a f(a) - \max_a g(a) \ge f(a') - g(a')$. Then get

$$\left| \max_a f(a) - \max_a g(a) \right| \le \max\left\{ |f(a') - g(a')|, |f(a^*) - g(a^*)| \right\} \le \max_a |f(a) - g(a)|.$$

Next, we proof the $\gamma$-contractive of $\mathcal{T}$:

$$\|\mathcal{T}Q - \mathcal{T}Q'\|_\infty = \max_{\boldsymbol{x},a} \left| \sum_{\boldsymbol{y}\in\mathcal{X}} P_0(\boldsymbol{y}|\boldsymbol{x},a) \left[ r(\boldsymbol{x},a) + \gamma \max_{b\in\mathcal{A}} Q(\boldsymbol{y},b) - r(\boldsymbol{x},a) + \gamma \max_{b\in\mathcal{A}} Q'(\boldsymbol{y},b) \right] \right|$$

$$= \max_{\boldsymbol{x},a} \gamma \left| \sum_{\boldsymbol{y}\in\mathcal{X}} P_0(\boldsymbol{y}|\boldsymbol{x},a) \left[ \max_{b\in\mathcal{A}} Q(\boldsymbol{y},b) - \max_{b\in\mathcal{A}} Q'(\boldsymbol{y},b) \right] \right|$$

$$\le \max_{\boldsymbol{x},a} \gamma \sum_{\boldsymbol{y}\in\mathcal{X}} P_0(\boldsymbol{y}|\boldsymbol{x},a) \left| \max_{b\in\mathcal{A}} Q(\boldsymbol{y},b) - \max_{b\in\mathcal{A}} Q'(\boldsymbol{y},b) \right|$$

$$\le \max_{\boldsymbol{x},a} \gamma \sum_{\boldsymbol{y}\in\mathcal{X}} P_0(\boldsymbol{y}|\boldsymbol{x},a) \max_b |Q(\boldsymbol{y},b) - Q'(\boldsymbol{y},b)|$$

$$\le \max_{\boldsymbol{x},a} \gamma \sum_{\boldsymbol{y}\in\mathcal{X}} P_0(\boldsymbol{y}|\boldsymbol{x},a) \|Q - Q'\|_\infty$$

$$= \gamma \|Q - Q'\|_\infty.$$

It is shown that the Bellman operator $\mathcal{T}$ is $\gamma$-contractive with respect to the supremum norm over $\mathcal{X} \times \mathcal{A}$. Specifically, for any two action-value functions $Q$ and $Q'$ defined on $\mathcal{X} \times \mathcal{A}$, it holds that $\|\mathcal{T}Q - \mathcal{T}Q'\|_\infty \le \gamma\|Q - Q'\|_\infty$. This contraction property forms the foundation of the well-known value iteration algorithm (41), which constructs a sequence of action-value functions $\{Q_k\}_{k\ge 0}$ by iteratively applying $\mathcal{T}$, where $Q_k = \mathcal{T}Q_{k-1}$ for all $k \ge 1$, starting from an arbitrary initial function $Q_0$. It follows that $\|Q_k - Q^{\pi^*}\|_\infty \le \gamma^k\|Q_0 - Q^{\pi^*}\|_\infty$, indicating that the sequence $\{Q_k\}_{k\ge 0}$ converges to the optimal value function $Q^{\pi^*}$ at a linear rate. $\qquad\square$

## B.5 Assumption for Cox model

We first define that

$$S_k^{(r)}(\boldsymbol{\eta}, s) := \frac{1}{N_k} \sum_{j=1}^{N_k} V_{jk}(s)\, e^{\boldsymbol{\eta}^\top \boldsymbol{Z}_{jk}}\, \boldsymbol{Z}_{jk}^{\otimes r}, \quad \bar{\boldsymbol{Z}}_k(\boldsymbol{\eta}, s) := \frac{S_k^{(1)}(\boldsymbol{\eta}, s)}{S_k^{(0)}(\boldsymbol{\eta}, s)},$$

$$e_k(\boldsymbol{\eta}, s) := \frac{S_k^{(1)}(\boldsymbol{\eta}, s)}{S_k^{(0)}(\boldsymbol{\eta}, s)}, \quad v_k(\boldsymbol{\eta}, s) := \frac{S_k^{(2)}(\boldsymbol{\eta}, s)}{S_k^{(0)}(\boldsymbol{\eta}, s)} - e_k(\boldsymbol{\eta}, s)^{\otimes 2}.$$

Let the cumulative baseline hazard function be $\Lambda_{0,k}(s)$ and let $\boldsymbol{\eta}_k$ be the true parameter at stage $k$.

**Assumption B.2.** (Finite interval).

$$|\Lambda_{0,k}(G_k) - \Lambda_{0,k}(0)| < \infty.$$

**Assumption B.3.** (Asymptotic stability). There exists a neighborhood $\mathcal{B}$ of $\boldsymbol{\eta}_k$ and deterministic bounded limit functions $s_k^{(0)}, s_k^{(1)}, s_k^{(2)} : \mathcal{B} \times [0, G_k] \to \mathbb{R}, \mathbb{R}^p, \mathbb{R}^{p \times p}$ such that for $j = 0, 1, 2$,

$$\sup_{s \in [0, G_k]} \sup_{\boldsymbol{\eta} \in \mathcal{B}} \left\| \mathcal{S}_k^{(j)}(\boldsymbol{\eta}, s) - s_k^{(j)}(\boldsymbol{\eta}, s) \right\| \xrightarrow{p} 0.$$

**Assumption B.4.** (Lindeberg). $\|\boldsymbol{Z}_{ik}\|$ are uniformly bounded, and there exists $\Delta > 0$ such that, with time–invariant covariates $\boldsymbol{Z}_{ik}$,

$$N_k^{-1/2} \sup_{1 \le i \le N_k} \sup_{s \in [0, G_k]} \|\boldsymbol{Z}_{ik}\| V_{ik}(s) \mathbf{1}\left\{ \boldsymbol{\eta}^\top \boldsymbol{Z}_{ik} > -\Delta \|\boldsymbol{Z}_{ik}\| \right\} \xrightarrow{p} 0 \quad \text{uniformly for } \boldsymbol{\eta} \in \mathcal{B}.$$

**Assumption B.5.** (Regularity). Let $e_k, v_k$ be as defined above. Require:

1. For every $s \in [0, G_k]$,

$$s_k^{(1)}(\boldsymbol{\eta}, s) = \frac{\partial}{\partial \boldsymbol{\eta}} s_k^{(0)}(\boldsymbol{\eta}, s), \qquad s_k^{(2)}(\boldsymbol{\eta}, s) = \frac{\partial^2}{\partial \boldsymbol{\eta}^2} s_k^{(0)}(\boldsymbol{\eta}, s),$$

and $s_k^{(j)}(\cdot, s)$ are continuous on $\mathcal{B}$ and bounded on $\mathcal{B} \times [0, G_k]$ for $j = 0, 1, 2$.

2. $s_k^{(0)}(\boldsymbol{\eta}, s)$ is bounded away from zero on $\mathcal{B} \times [0, G_k]$.

3. The matrix

$$\Sigma_k := \int_0^{G_k} v_k(\boldsymbol{\eta}_k, s)\, s_k^{(0)}(\boldsymbol{\eta}_k, s) d\Lambda_{0,k}(s)$$

is positive definite.

## B.6 Assumption for Aah model

**Assumption B.6** (Bounded covariates). The covariate vectors are uniformly bounded: $\|\boldsymbol{Z}_{ik}\| \le C_Z$.

**Assumption B.7** (Risk-set positivity). There exists a constant $c_R > 0$ such that

$$\sum_{j=1}^{N_k} V_{jk}(s) \ge c_R N_k \text{ for all } s \le G_k$$

in probability.

**Assumption B.8** (Baseline hazard and linear predictor bounds). The baseline hazard function satisfies $\lambda_{0,k} = \Lambda'_{0,k}$ locally integrable with

$$\Lambda_{0,k}(G_k) \le C_\Lambda < \infty, \text{ and } \sup_{s \le G_k} \{\lambda_{0,k}(s) + \|\boldsymbol{\beta}_k\| \|\boldsymbol{Z}_{ik}\|\} \le C_\lambda.$$

**Assumption B.9** (Asymptotic positive definiteness). We define

$$\tilde{\boldsymbol{Z}}_k(s) = \frac{\sum_{i=1}^{N_k} V_{ik}(s) \boldsymbol{Z}_{ik}}{\sum_{i=1}^{N_k} V_{ik}(s)}, \quad A_k = \frac{1}{N_k} \sum_{i=1}^{N_k} \int_0^{G_k} V_{ik}(s)\{\boldsymbol{Z}_{ik} - \tilde{\boldsymbol{Z}}_k(s)\}^{\otimes 2} ds.$$

The matrix $A_k$ converges in probability to a positive definite matrix $A_{k,\infty}$ whose eigenvalues are bounded away from 0 and $\infty$.

## B.7 Proof of Theorem 3.1

### B.7.1 Proof of Theorem 3.1 under Cox model

*Proof.* **(i) Consistency**. From Theorem 3.2 in Andersen & Gill (2), it holds that $\hat{\boldsymbol{\eta}}_k \xrightarrow{p} \boldsymbol{\eta}_k$ and $\sup_{t \le G_k} \left|\hat{\Lambda}_{0,k}(t) - \Lambda_{0,k}(t)\right| \xrightarrow{P} 0$. Under the Cox model, we have

$$\Lambda_{T_k}(t \mid \boldsymbol{Z}_k) = \int_0^t \exp\{\boldsymbol{\eta}_k^\top \boldsymbol{Z}_k\} d\Lambda_{0,k}(u), \quad \hat{\Lambda}_{T_k}(t \mid \boldsymbol{Z}_k) = \int_0^t \exp\{\hat{\boldsymbol{\eta}}_k^\top \boldsymbol{Z}_k\} d\hat{\Lambda}_{0,k}(u).$$

For any $t \le G_k$, write

$$\hat{\Lambda}_{T_k}(t \mid \boldsymbol{Z}_k) - \Lambda_{T_k}(t \mid \boldsymbol{Z}_k)$$
$$= \underbrace{\int_0^t \left[\exp\{\hat{\boldsymbol{\eta}}_k^\top \boldsymbol{Z}_k\} - \exp\{\boldsymbol{\eta}_k^\top \boldsymbol{Z}_k\}\right] d\Lambda_{0,k}(u)}_{=:R_{1k}(t)} + \underbrace{\int_0^t \exp\{\hat{\boldsymbol{\eta}}_k^\top \boldsymbol{Z}_k\} d(\hat{\Lambda}_{0,k} - \Lambda_{0,k})(u)}_{=:R_{2k}(t)}.$$

For $R_{1k}(t)$, by the mean value theorem and boundedness of $\boldsymbol{Z}_k$,

$$\left|\exp\{\hat{\boldsymbol{\eta}}_k^\top \boldsymbol{Z}_k\} - \exp\{\boldsymbol{\eta}_k^\top \boldsymbol{Z}_k\}\right| \le \exp\{C\} C_Z \|\hat{\boldsymbol{\eta}}_k - \boldsymbol{\eta}_k\| \xrightarrow{P} 0,$$

for some finite constant $C$. In addition, under assumption B.4, $\Lambda_{0,k}(t)$ is of bounded variation. That is, $\mathrm{TV}\{\Lambda_{0,k}; [0, G_k]\} = \Lambda_{0,k}(G_k) < \infty$. Hence

$$\sup_{t \le G_k} |R_{1k}(t)| \le \mathrm{TV}\{\Lambda_{0,k}; [0, G_k]\} \left|\exp(\hat{\boldsymbol{\eta}}_k^\top \boldsymbol{Z}_k) - \exp(\boldsymbol{\eta}_k^\top \boldsymbol{Z}_k)\right| \xrightarrow{P} 0.$$

For $R_{2k}(t)$, since $\hat{\boldsymbol{\eta}}_k \xrightarrow{p} \boldsymbol{\eta}_k$ and $\|\boldsymbol{Z}_k\| \le C_Z$, the continuous mapping theorem implies $\exp\{\|\hat{\boldsymbol{\eta}}_k\| C_Z\} \xrightarrow{p} \exp\{\|\boldsymbol{\eta}_k\| C_Z\}$. Hence for any $\varepsilon > 0$ there exists $M < \infty$ such that $\mathbb{P}\left(\exp\{\hat{\boldsymbol{\eta}}_k^\top \boldsymbol{Z}_k\} > M\right) < \varepsilon$ for sufficiently large $N_k$. Thus we have

$$\sup_{t \le G_k} |R_{2k}(t)| \le M \sup_{t \le G_k} \left|\hat{\Lambda}_{0,k}(t) - \Lambda_{0,k}(t)\right| \xrightarrow{P} 0.$$

It follows that

$$\sup_{t \le G_k} \left|\hat{\Lambda}_{T_k}(t \mid \boldsymbol{Z}_k) - \Lambda_{T_k}(t \mid \boldsymbol{Z}_k)\right| \le \sup_{t \le G_k} |R_{1k}(t)| + \sup_{t \le G_k} |R_{2k}(t)| \xrightarrow{P} 0.$$

**(ii) Convergence rate**. Applying the first-order Taylor expansion to $\hat{\Lambda}_{T_k}(t|\boldsymbol{Z}_k)$ with respect to $\hat{\boldsymbol{\eta}}_k$, we obtain

$$W_k(t, \boldsymbol{Z}_k) := \sqrt{N_k}\left\{\hat{\Lambda}_{T_k}(t \mid \boldsymbol{Z}_k) - \Lambda_{T_k}(t \mid \boldsymbol{Z}_k)\right\} = A_k(t, \boldsymbol{Z}_k) + H_k^\top(t, \boldsymbol{Z}_k)\mathcal{I}_k(\boldsymbol{\eta}_k)^{-1}B_k(t),$$

$$\text{(A.2)}$$

where

$$A_k(t, \boldsymbol{Z}_k) = N_k^{-\frac{1}{2}} \sum_{i=1}^{N_k} \int_0^t \frac{e^{\boldsymbol{\eta}_k^\top \boldsymbol{Z}_k}}{S_k^{(0)}(\boldsymbol{\eta}_k, s)} dM_{ik}(s),$$

$$B_k(t) = N_k^{-\frac{1}{2}} \sum_{i=1}^{N_k} \int_0^t \left\{\boldsymbol{Z}_{ik}(s) - \bar{\boldsymbol{Z}}_k(\boldsymbol{\eta}_k, s)\right\} dM_{ik}(s),$$

$$H_k(t, \boldsymbol{Z}_k) = \int_0^t e^{\boldsymbol{\eta}_k^\top \boldsymbol{Z}_k} \left\{\boldsymbol{Z}_k - \bar{\boldsymbol{Z}}_k(\boldsymbol{\eta}_k, s)\right\} d\Lambda_{0,k}(s),$$

$$\mathcal{I}_k(\boldsymbol{\eta}_k) = \int_0^{G_k} \left\{\frac{S_k^{(2)}(\boldsymbol{\eta}_k, s)}{S_k^{(0)}(\boldsymbol{\eta}_k, s)} - \left(\frac{S_k^{(1)}(\boldsymbol{\eta}_k, s)}{S_k^{(0)}(\boldsymbol{\eta}_k, s)}\right)^{\otimes 2}\right\} S_k^{(0)}(\boldsymbol{\eta}_k, s) d\Lambda_{0,k}(s),$$

and $M_{ik}(t) = U_{ik}(t) - \int_0^t V_{ik}(s) \exp\{\boldsymbol{\beta}^\top \boldsymbol{Z}_{ik}\} d\Lambda_{0,k}(s)$ is a local martingale. By Assumption B.3 and B.5(ii),

$$\sup_{s \le t} \left\|S_k^{(r)}(\boldsymbol{\eta}_k, s) - s_k^{(r)}(\boldsymbol{\eta}_k, s)\right\| \xrightarrow{P} 0, \quad s_k^{(0)}(\boldsymbol{\eta}_k, s) \ge c_0 > 0,$$

hence there exist constants $C_0$ and $c_0$, such that $S_k^{(0)}(\boldsymbol{\eta}_k, s) \geq c_0/2$ and $S_k^{(r)}(\boldsymbol{\eta}_k, s) \leq C_0$ for $r = 0, 1, 2$ and all $s \leq t$. Assumption B.4 implies that $\|\boldsymbol{Z}_{ik}\| \leq C_Z$, and thus $e^{\boldsymbol{\eta}_k^\top \boldsymbol{Z}_{ik}} \leq C_e$ for some $C_e < \infty$. Assumption B.2 implies that $\Lambda_{0,k}(t) \leq C_\Lambda < \infty$.

(a) Bounded variance of $A_k(t, \boldsymbol{Z}_k)$. Since $\{M_{ik}\}$ are orthogonal square integrable martingales with $d\langle M_{ik}\rangle(s) = V_{ik}(s)e^{\boldsymbol{\eta}_k^\top \boldsymbol{Z}_{ik}} d\Lambda_{0,k}(s)$, we have

$$Var(A_k(t, \boldsymbol{Z}_k)) = \frac{1}{N_k} \sum_{i=1}^{N_k} \int_0^t \Big( \frac{e^{\boldsymbol{\eta}_k^\top \boldsymbol{Z}_k}}{S_k^{(0)}(\boldsymbol{\eta}_k, s)} \Big)^2 V_{ik}(s) e^{\boldsymbol{\eta}_k^\top \boldsymbol{Z}_{ik}} d\Lambda_{0,k}(s).$$

Becasue $\frac{1}{N_k} \sum_{i=1}^{N_k} V_{ik}(s) e^{\boldsymbol{\eta}_k^\top \boldsymbol{Z}_{ik}} = S_k^{(0)}(\boldsymbol{\eta}_k, s)$, hence

$$Var(A_k(t, \boldsymbol{Z}_k)) = \int_0^t \frac{e^{2\boldsymbol{\eta}_k^\top \boldsymbol{Z}_k}}{S_k^{(0)}(\boldsymbol{\eta}_k, s)} d\Lambda_{0,k}(s) \leq \frac{2C_e^2 C_\Lambda}{c_0} < \infty. \tag{A.3}$$

(b) Bounded variance of $B_k(t)$ and scaling of $\mathcal{I}_k(\boldsymbol{\eta}_k)^{-1} B_k(t)$. By definition,

$$B_k(t) = N_k^{-1/2} \sum_{i=1}^{N_k} \int_0^{G_k} \big\{ \boldsymbol{Z}_{ik} - \bar{\boldsymbol{Z}}_k(\boldsymbol{\eta}_k, s) \big\} dM_{ik}(s).$$

Since $\{M_{ik}\}$ are orthogonal square integrable martingales with $d\langle M_{ik}\rangle(s) = V_{ik}(s)e^{\boldsymbol{\eta}_k^\top \boldsymbol{Z}_{ik}} d\Lambda_{0,k}(s)$, we have

$$Var(B_k(t)) = \int_0^{G_k} v_k(\boldsymbol{\eta}_k, s) S_k^{(0)}(\boldsymbol{\eta}_k, s) d\Lambda_{0,k}(s) = \mathcal{I}_k(\boldsymbol{\eta}_k). \tag{A.4}$$

Let $\tilde{\mathcal{I}}_k(\boldsymbol{\eta}_k) := \mathcal{I}_k(\boldsymbol{\eta}_k)/N_k$. By Assumption B.5(iii), $\tilde{\mathcal{I}}_k(\boldsymbol{\eta}_k) \xrightarrow{p} \Sigma_k \succ 0$. Therefore, for sufficiently large $N_k$, there exists a constant $C_I < \infty$ such that

$$\|\mathcal{I}_k(\boldsymbol{\eta}_k)^{-1}\| = \frac{1}{N_k} \|\tilde{\mathcal{I}}_k(\boldsymbol{\eta}_k)^{-1}\| \leq \frac{C_I}{N_k}.$$

Therefore,

$$Var\big(\mathcal{I}_k(\boldsymbol{\eta}_k)^{-1} B_k(t)\big) = (\mathcal{I}_k(\boldsymbol{\eta}_k)^{-1})^\top \leq \|\mathcal{I}_k(\boldsymbol{\eta}_k)^{-1}\| \leq \frac{C_I}{N_k},$$

which is uniformly bounded.

(c) Boundedness of $H_k(t, \boldsymbol{Z}_k)$ and variance of $H_k(t, \boldsymbol{Z}_k)^\top \mathcal{I}_k^{-1}(\boldsymbol{\eta}_k) B_k(t)$. Since $\|\boldsymbol{Z}_k\| \leq C_Z$ and $\|\bar{\boldsymbol{Z}}_k(\boldsymbol{\eta}_k, s)\| \leq 2C_0/c_0$ by Assumptions B.3–B.5, there exists a $C_H < \infty$, such that

$$\|H_k(t, \boldsymbol{Z}_k)\| = \int_0^t e^{\boldsymbol{\eta}_k^\top \boldsymbol{Z}_k} \big\{ \boldsymbol{Z}_k - \bar{\boldsymbol{Z}}_k(\boldsymbol{\eta}_k, s) \big\} d\Lambda_{0,k}(s) \leq C_e \Big( C_Z + \tfrac{2C_0}{c_0} \Big) C_\Lambda := C_H.$$

Then we conclude

$$Var(H_k^\top(t, \boldsymbol{Z}_k) \mathcal{I}_k(\boldsymbol{\eta}_k)^{-1} B_k(t)) \leq \|H_k(t, \boldsymbol{Z}_k)\|^2 Var(\mathcal{I}_k(\boldsymbol{\eta}_k)^{-1} B_k(t)) \leq \frac{C_H^2 C_I}{N_k},$$

which is also uniformly bounded.

(d) Boundedness of $Cov(A_k(t, \boldsymbol{Z}_k), B_k(t))$. We have

$$Cov(A_k(t, \boldsymbol{Z}_k), B_k(t)) = \frac{1}{N_k} \sum_{i=1}^{N_k} \int_0^t \frac{e^{\boldsymbol{\eta}_k^\top \boldsymbol{Z}_k}}{S_k^{(0)}(\boldsymbol{\eta}_k, u)} \big\{ \boldsymbol{Z}_{ik}(u) - \bar{\boldsymbol{Z}}_k(\boldsymbol{\eta}_k, u) \big\} V_{ik}(s) e^{\boldsymbol{\eta}_k^\top \boldsymbol{Z}_{ik}} d\Lambda_{0,k}(s) = 0.$$

Combining (a)–(d), there exists a $C_W < \infty$ such that,

$$\sup_{t \leq G_k} Var(W_k(t, \boldsymbol{Z}_k)) = \sup_{t \leq G_k} Var(A_k(t, \boldsymbol{Z}_k) + H_k^\top(t, \boldsymbol{Z}_k) \mathcal{I}_k(\boldsymbol{\eta}_k)^{-1} B_k(t)) = \frac{2C_e^2 C_\Lambda}{c_0} + \frac{C_H^2 C_I}{N_k} := C_W.$$

Thus, by Chebyshev inequality, $\mathbb{P}(\sup_t |W_k(t, \boldsymbol{Z}_k)| > M) \leq C_W/M^2$ for all $M > 0$, i.e., $\sup_t |W_k(t, \boldsymbol{Z}_k)| = O_p(1)$, then we have $\hat{\Lambda}_{T_k}(t \mid \boldsymbol{Z}_k) - \Lambda_{T_k}(t \mid \boldsymbol{Z}_k) = O_p(N_k^{-\frac{1}{2}})$.

**(iii) Asymptotic normality**. Following from the martingale central limit theorem, it can be obtained that the process $W_k(t, \boldsymbol{Z}_k)$ converges weakly to a zero-mean Gaussian process on $[0, G_k]$. Next we calculate the variance of the process. Since A.2, A.3, A.4, and $\mathcal{I}_k$ is a symmetric matrix, we have

$$Var(W_k(t, \boldsymbol{Z}_k)) = \text{Cov}(A_k(t, \boldsymbol{Z}_k), A_k(t, \boldsymbol{Z}_k)) + H_k^\top(t, \boldsymbol{Z}_k)\mathcal{I}_k(\boldsymbol{\eta}_k)^{-1} H_k(t, \boldsymbol{Z}_k)$$

$$= \int_0^t \frac{e^{2\boldsymbol{\eta}_k^\top \boldsymbol{Z}_k}}{S_k^{(0)}(\boldsymbol{\eta}_k, s)} d\Lambda_{0,k}(s) + H_k^\top(t, \boldsymbol{Z}_k)\mathcal{I}_k(\boldsymbol{\eta}_k)^{-1} H_k(t, \boldsymbol{Z}_k). \quad (A.5)$$

Thus $W_k(t, \boldsymbol{Z}_k)$ converges weakly to a zero-mean Gaussian process with covariance given by (A.5). Replace all population quantities by their sample counterparts, a consistent estimator of (A.5) is

$$\widehat{Var}(W_k(t, \boldsymbol{Z}_k)) = \int_0^t \frac{e^{2\widehat{\boldsymbol{\eta}}_k^\top \boldsymbol{Z}_k}}{\widehat{S}_k^{(0)}(\widehat{\boldsymbol{\eta}}_k, s)} d\widehat{\Lambda}_{0,k}(s) + \widehat{H}_k^\top(t, \boldsymbol{Z}_k)\widehat{\mathcal{I}}_k^{-1} \widehat{H}_k(t, \boldsymbol{Z}_k), \quad (A.6)$$

where

$$\widehat{S}_k^{(0)}(\widehat{\boldsymbol{\eta}}_k, s) = \frac{1}{N_k}\sum_{i=1}^{N_k} V_{ik}(s)e^{\widehat{\boldsymbol{\eta}}_k^\top \boldsymbol{Z}_{ik}}, \quad \widehat{H}_k(t, \boldsymbol{Z}_k) = \int_0^t e^{\widehat{\boldsymbol{\eta}}_k^\top \boldsymbol{Z}_k}\big\{ \boldsymbol{Z}_k - \widehat{\boldsymbol{Z}}_k(s)\big\} d\widehat{\Lambda}_{0,k}(s),$$

$$\widehat{\boldsymbol{Z}}_k(s) = \frac{S_k^{(1)}(\widehat{\boldsymbol{\eta}}_k, s)}{S_k^{(0)}(\widehat{\boldsymbol{\eta}}_k, s)}, \quad \widehat{\mathcal{I}}_k = \int_0^{G_k} \left\{ \frac{S_k^{(2)}(\widehat{\boldsymbol{\eta}}_k, s)}{S_k^{(0)}(\widehat{\boldsymbol{\eta}}_k, s)} - \left(\frac{S_k^{(1)}(\widehat{\boldsymbol{\eta}}_k, s)}{S_k^{(0)}(\widehat{\boldsymbol{\eta}}_k, s)}\right)^{\otimes 2} \right\} S_k^{(0)}(\widehat{\boldsymbol{\eta}}_k, s) d\widehat{\Lambda}_{0,k}(s).$$

$\square$

### B.7.2 Proof of the Aalen Additive Hazards Model Estimator

*Proof.* **(i) Consistency**. According to Lin & Ying (21), we have

$$\widehat{\boldsymbol{\beta}}_k \xrightarrow{p} \boldsymbol{\beta}_k, \quad \sup_{t \le G_k} \big|\widehat{\Lambda}_{0,k}(t) - \Lambda_{0,k}(t)\big| \xrightarrow{p} 0.$$

Under assumption B.6,

$$\sup_{t \le G_k} \big|\widehat{\Lambda}_{T_k}(t \mid \boldsymbol{Z}_k) - \Lambda_{T_k}(t \mid \boldsymbol{Z}_k)\big| \le \sup_{t \le G_k} \big|\widehat{\Lambda}_{0,k}(t) - \Lambda_{0,k}(t)\big| + G_k C_Z \|\widehat{\boldsymbol{\beta}}_k - \boldsymbol{\beta}_k\| \xrightarrow{p} 0.$$

**(ii) Convergence rate**. Let us define $d\Lambda_{0,k}(s) = \lambda_{0,k}(s)ds$. According to assumptions B.6-B.9, we have (i) $R_k(s) \ge c_R N_k$, (ii) $\|\boldsymbol{Z}_{ik}\| \le C_Z$, (iii) $\lambda_{0,k}(s) + \boldsymbol{\beta}_k^\top \boldsymbol{Z}_{ik} \le C_\lambda$, (iv) $c_A I_d \preceq A_k$, with finite strictly positive constants $c_R, C_Z, C_\lambda, c_A$. According to Lin & Ying (21), we have

$$\sqrt{N_k}(\widehat{\boldsymbol{\beta}}_k - \boldsymbol{\beta}_k) = A_k^{-1} \frac{1}{\sqrt{N_k}}\sum_{i=1}^{N_k} \int_0^{G_k} \{\boldsymbol{Z}_{ik} - \tilde{\boldsymbol{Z}}_k(s)\} dM_{ik}(s),$$

$$\widehat{\Lambda}_{0,k}(t) - \Lambda_{0,k}(t) = \frac{1}{N_k}\sum_{i=1}^{N_k} \int_0^t \frac{1}{R_k(s)} dM_{ik}(s) - C_k(t)^\top (\widehat{\boldsymbol{\beta}}_k - \boldsymbol{\beta}_k).$$

where $M_{ik}(s) = U_{ik}(s) - \int_0^s V_{ik}(u)\{\lambda_{0,k}(u) + \boldsymbol{\beta}_k^\top \boldsymbol{Z}_{ik}\}du$ is a local martingale, and

$$\tilde{\boldsymbol{Z}}_k(s) = \frac{\sum_{i=1}^{N_k} V_{ik}(s)\boldsymbol{Z}_{ik}}{\sum_{i=1}^{N_k} V_{ik}(s)}, \quad A_k = \frac{1}{N_k}\sum_{i=1}^{N_k} \int_0^{G_k} V_{ik}(s)\{\boldsymbol{Z}_{ik} - \tilde{\boldsymbol{Z}}_k(s)\}^{\otimes 2} ds,$$

$$C_k(t) = \int_0^t \tilde{\boldsymbol{Z}}_k(s)ds, \quad R_k(s) = \sum_{j=1}^{N_k} V_{jk}(s).$$

Hence, for the cumulative hazard at covariate $\boldsymbol{Z}_k$,

$$W_k(t, \boldsymbol{Z}_k) := \sqrt{N_k}\{\widehat{\Lambda}_{T_k}(t \mid \boldsymbol{Z}_k) - \Lambda_{T_k}(t \mid \boldsymbol{Z}_k)\} = A_k^{(1)}(t) + A_k^{(2)}(t, \boldsymbol{Z}_k),$$

with

$$A_k^{(1)}(t) = \frac{1}{\sqrt{N_k}} \sum_{i=1}^{N_k} \int_0^t \frac{1}{R_k(s)} dM_{ik}(s), \ A_k^{(2)}(t, \boldsymbol{Z}_k) = \frac{1}{\sqrt{N_k}} \sum_{i=1}^{N_k} \int_0^{G_k} \phi_{ik}(s; t, \boldsymbol{Z}_k) dM_{ik}(s),$$

where

$$\phi_{ik}(s; t, \boldsymbol{Z}_k) := \{t\, \boldsymbol{Z}_k - C_k(t)\}^\top A_k^{-1} \{\boldsymbol{Z}_{ik} - \tilde{\boldsymbol{Z}}_k(s)\}.$$

Since $d\langle M_{ik}\rangle(s) = V_{ik}(s)\{\lambda_{0,k}(s) + \boldsymbol{\beta}_k^\top \boldsymbol{Z}_{ik}\}\, ds \leq C_\lambda\, ds$,

$$Var\big(A_k^{(1)}(t)\big) = \frac{1}{N_k} \sum_{i=1}^{N_k} \int_0^t \frac{1}{R_k(s)^2}\, d\langle M_{ik}\rangle(s) \leq \frac{C_\lambda t}{c_R^2 N_k^2} \leq \frac{C_\lambda G_k}{c_R^2}.$$

According to assumptions B.6 and B.9, we have $\|A_k^{-1}\| \leq \frac{1}{c_A}$ and $\|\tilde{\boldsymbol{Z}}_k(s)\| \leq C_Z$. Then

$$|\phi_{ik}(s; t, \boldsymbol{Z}_k)| \leq \big(t\|\boldsymbol{Z}_k\| + \|C_k(t)\|\big)\|A_k^{-1}\|\|\boldsymbol{Z}_{ik} - \tilde{\boldsymbol{Z}}_k(s)\| \leq 2\frac{G_k C_Z^2 + G_k C_Z^2}{c_A} =: C_\phi,$$

hence

$$Var\big(A_k^{(2)}(t, \boldsymbol{Z}_k)\big) = \frac{1}{N_k} \sum_{i=1}^{N_k} \int_0^{G_k} \phi_{ik}(s; t, \boldsymbol{Z}_k)^2\, d\langle M_{ik}\rangle(s) \leq C_\phi^2 C_\lambda G_k.$$

By the covariance formula for martingale integrals and Cauchy–Schwarz inequality,

$$\big|\operatorname{Cov}(A_k^{(1)}(t), A_k^{(2)}(t, \boldsymbol{Z}_k))\big| \leq \sqrt{Var(A_k^{(1)}(t))} \sqrt{Var(A_k^{(2)}(t, \boldsymbol{Z}_k))} \leq \sqrt{\frac{C_\lambda G_k}{c_R^2}} \sqrt{C_\phi^2 C_\lambda G_k}.$$

There exists a $C_W < \infty$ (independent of $N_k$) such that

$$\sup_{t \leq G_k} Var\big(W_k(t, \boldsymbol{Z}_k)\big) \ \leq \ \sup_{t \leq G_k} \Big\{ Var(A_k^{(1)}(t)) + Var(A_k^{(2)}(t, \boldsymbol{Z}_k)) + 2|\operatorname{Cov}(A_k^{(1)}(t), A_k^{(2)}(t, \boldsymbol{Z}_k))|\Big\}$$
$$\leq \ C_W.$$

By Chebyshev's inequality,

$$\sup_{t \leq G_k} \mathbb{P}\big(|W_k(t, \boldsymbol{Z}_k)| > M\big) \leq \frac{C_W}{M^2} \Rightarrow \sup_{t \leq G_k} \big|\hat{\Lambda}_{T_k}(t \mid \boldsymbol{Z}_k) - \Lambda_{T_k}(t \mid \boldsymbol{Z}_k)\big| = O_p(N_k^{-1/2}).$$

**(iii) Asymptotic normality.** Following from the martingale central limit theorem, it can be obtained that the process $W_k(t, \boldsymbol{Z}_k)$ converges weakly to a zero-mean Gaussian process on $[0, G_k]$. Next we calculate the variance of the process.

(a) Variance of $A_k^{(2)}(t, \boldsymbol{Z}_k)$. Let $O_k(t, \boldsymbol{Z}_k) = t\boldsymbol{Z}_k - C_k(t)$, we have

$$Var(A_k^{(2)}(t, \boldsymbol{Z}_k)) = \frac{1}{N_k} \sum_{i=1}^{N_k} \int_0^{G_k} \phi_{ik}(s; t, \boldsymbol{Z}_k)^2\, d\langle M_{ik}\rangle(s)$$

$$= \frac{1}{N_k} \sum_{i=1}^{N_k} \int_0^{G_k} \left(O_k(t, \boldsymbol{Z}_k)^\top A_k^{-1}\{\boldsymbol{Z}_{ik} - \tilde{\boldsymbol{Z}}_k(s)\}\right)^2 d\langle M_{ik}\rangle(s)$$

$$= O_k(t, \boldsymbol{Z}_k)^\top A_k^{-1} \left\{ \frac{1}{N_k} \sum_{i=1}^{N_k} \int_0^{G_k} \{\boldsymbol{Z}_{ik} - \tilde{\boldsymbol{Z}}_k(s)\}^{\otimes 2}\, d\langle M_{ik}\rangle(s) \right\} A_k^{-1} O_k(t, \boldsymbol{Z}_k).$$

Define the matrix-valued process

$$B_k(v) := \frac{1}{N_k} \sum_{i=1}^{N_k} \int_0^v \{\boldsymbol{Z}_{ik} - \tilde{\boldsymbol{Z}}_k(s)\}^{\otimes 2} dU_{ik}(s).$$

Then according to Lemma B.10, we have

$$Var(A_k^{(2)}(t, \boldsymbol{Z}_k)) = O_k(t, \boldsymbol{Z}_k)^\top A_k^{-1} B_k(G_k) A_k^{-1} O_k(t, \boldsymbol{Z}_k).$$

(b) Covariance of $A_k^{(1)}(t)$ and $A_k^{(2)}(t)$. Again by orthogonality across $i$ and the martingale isometry,

$$
\begin{aligned}
Cov(A_k^{(1)}(t), A_k^{(2)}(t, \mathbf{Z}_k)) &= \frac{1}{N_k} \sum_{i=1}^{N_k} \int_0^t \frac{1}{R_k(s)} \phi_{ik}(s; t, \mathbf{Z}_k) \, d\langle M_{ik}\rangle(s) \\
&= \frac{1}{N_k} \sum_{i=1}^{N_k} \int_0^t \frac{1}{R_k(s)} O_k(t, \mathbf{Z}_k)^\top A_k^{-1}\{\mathbf{Z}_{ik} - \tilde{\mathbf{Z}}_k(s)\} \, d\langle M_{ik}\rangle(s) \\
&= O_k(t, \mathbf{Z}_k)^\top A_k^{-1} \frac{1}{N_k} \int_0^t \frac{\sum_{i=1}^{N_k}\{\mathbf{Z}_{ik} - \tilde{\mathbf{Z}}_k(s)\} \, d\langle M_{ik}\rangle(s)}{R_k(s)}.
\end{aligned}
$$

Let

$$
D_k(v) := \int_0^v \frac{\sum_{i=1}^{N_k}\{\mathbf{Z}_{ik} - \tilde{\mathbf{Z}}_k(s)\} dU_{ik}(s)}{R_k(s)}.
$$

By Lemma B.10, we have

$$
Cov(A_k^{(1)}(t), A_k^{(2)}(t, \mathbf{Z}_k)) = O_k(t, \mathbf{Z}_k)^\top A_k^{-1} D_k(t).
$$

Thus,

$$
\begin{aligned}
&Var(W_k(t, \mathbf{Z}_k)) \\
&= Var(A^{(1)}(t)) + Var(A^{(2)}(t, \mathbf{Z}_k)) + 2Cov(A^{(1)}(t), A^{(2)}(t, \mathbf{Z}_k)) \\
&= \int_0^t \frac{\sum_{i=1}^{N_k} dU_{ik}(s)}{N_k R_k(s)^2} + O_k(t, \mathbf{Z}_k)^\top A_k^{-1} B_k(t) A_k^{-1} O_k(t, \mathbf{Z}_k) + 2O_k(t, \mathbf{Z}_k)^\top A_k^{-1} D_k(t).
\end{aligned}
$$

A consistent estimator of $Var(W_k(t, \mathbf{Z}_k))$ is obtained by

$$
\widehat{\mathrm{Var}}\{W_k(t, \mathbf{Z}_k)\} = \int_0^t \frac{\sum_{i=1}^{N_k} dU_{ik}(s)}{N_k \widehat{R}_k(s)^2} + \widehat{O}_k(t, \mathbf{Z}_k)^\top \widehat{A}_k^{-1} \widehat{B}_k(t) \widehat{A}_k^{-1} \widehat{O}_k(t, \mathbf{Z}_k) + 2\widehat{O}_k(t, \mathbf{Z}_k)^\top \widehat{A}_k^{-1} \widehat{D}_k(t),
$$

$$
\text{(A.7)}
$$

where

$$
\widehat{R}_k(s) = \sum_{j=1}^{N_k} V_{jk}(s), \ \widehat{\mathbf{Z}}_k(s) = \frac{\sum_{j=1}^{N_k} V_{jk}(s) \mathbf{Z}_{jk}}{\sum_{j=1}^{N_k} V_{jk}(s)}, \ \widehat{C}_k(t) = \int_0^t \widehat{\mathbf{Z}}_k(s) ds, \ \widehat{O}_k(t, \mathbf{Z}_k) = t\mathbf{Z}_k - \widehat{C}_k(t),
$$

$$
\widehat{A}_k = \frac{1}{N_k} \sum_{i=1}^{N_k} \int_0^{G_k} V_{ik}(s)\{\mathbf{Z}_{ik} - \widehat{\mathbf{Z}}_k(s)\}^{\otimes 2} ds, \quad \widehat{B}_k = \frac{1}{N_k} \sum_{i=1}^{N_k} \int_0^{G_k} \{\mathbf{Z}_{ik} - \widehat{\mathbf{Z}}_k(s)\}^{\otimes 2} dU_{ik}(s),
$$

$$
\widehat{D}_k(t) = \int_0^t \frac{\sum_{i=1}^{N_k}\{\mathbf{Z}_{ik} - \widehat{\mathbf{Z}}_k(s)\} dU_{ik}(s)}{\widehat{R}_k(s)}.
$$

$\square$

## B.8 The objective function for Liu et al.'s (2023) method

Next we will show that the objective function in RL from Liu et al.'s (2023) is

$$
Q^\pi(\mathbf{x}_t, a_t) = \mathbb{E}^{P_0, \pi}\left[\sum_{k=t}^{\bar{K}} \gamma^{k-t} T_k \,\middle|\, \mathbf{x}_t, a_t\right] = \mathbb{E}^{P_0, \pi}\left[\sum_{k=t}^{\infty} \gamma^{k-t} \frac{Y_k \Delta_k}{\hat{S}_C(\sum_{i=1}^k Y_i)} \,\middle|\, \mathbf{X}_t, A_t\right]. \quad \text{(A.8)}
$$

where $\hat{S}_C$ is the Kaplan-Meier estimator.

*Proof.* Note that

$$
\mathbb{E}\left[\Delta_k \middle| \sum_{i=1}^k T_i\right] = P\left(C \geq \sum_{i=1}^k T_i\right) = S_C\left(\sum_{i=1}^k T_i\right),
$$

and thus

$$\mathbb{E}\left[\left.\frac{\Delta_k}{S_C\left(\sum_{i=1}^k T_i\right)}\right| \boldsymbol{x}_k, a_k, T_k\right] = 1.$$

Then we have

$$\mathbb{E}^{P_0,\pi}\left[\left.\sum_{k=t}^{\bar{K}} \gamma^{k-t} T_k \right| \boldsymbol{x}_t, a_t\right] = \mathbb{E}^{P_0,\pi}\left\{\left.\sum_{k=t}^{\bar{K}} \gamma^{k-t} T_k \mathbb{E}\left[\left.\frac{\Delta_k}{S_C\left(\sum_{i=1}^k T_i\right)}\right| \boldsymbol{x}_k, a_k, T_k\right]\right| \boldsymbol{x}_k, a_k\right\}$$

$$= \mathbb{E}^{P_0,\pi}\left\{\left.\sum_{k=t}^{\bar{K}} \gamma^{k-t} \mathbb{E}\left[\left.\frac{T_k \Delta_k}{S_C\left(\sum_{i=1}^k T_i\right)}\right| \boldsymbol{x}_k, a_k, T_k\right]\right| \boldsymbol{x}_t, a_t\right\}$$

$$= \mathbb{E}^{P_0,\pi}\left[\left.\sum_{k=t}^{\bar{K}} \gamma^{k-t} \frac{Y_k \Delta_k}{S_C\left(\sum_{i=1}^k Y_i\right)}\right| \boldsymbol{x}_t, a_t\right].$$

$\square$

## B.9 Large-sample replacement of $d\langle M\rangle$ by empirical $dU$

**Lemma B.10.** *Fix a stage $k$ and subjects $i = 1, \ldots, N_k$. Let $U_{ik}(t)$ be counting processes with predictable intensities $\lambda_{ik}(t)$ and at–risk indicators $V_{ik}(t)$. Assume:*

(A1) *(Doob–Meyer) $U_{ik}(t) = \Lambda_{ik}(t) + M_{ik}(t)$ with $\Lambda_{ik}(t) = \int_0^t V_{ik}(s)\lambda_{ik}(s)\,ds$; hence $d\langle M_{ik}\rangle(s) = d\Lambda_{ik}(s) = V_{ik}(s)\lambda_{ik}(s)ds$.*

(A2) *(Orthogonality/independence across $i$) The martingales $\{M_{ik}\}_{i \leq N_k}$ are pairwise orthogonal (e.g. subjects independent).*

(A3) *(Bounded predictable weights) For each $i$, $H_{ik}(s)$ is predictable, $\sup_{s \leq G_k} |H_{ik}(s)| \leq C_H < \infty$.*

(A4) *(Integrability) $\sup_{i, s \leq G_k} \lambda_{ik}(s) \leq C_\lambda < \infty$.*

*Define, for $t \in [0, G_k]$,*

$$R_{N_k}(t) := \frac{1}{N_k} \sum_{i=1}^{N_k} \int_0^t H_{ik}(s)\,dU_{ik}(s) - \frac{1}{N_k} \sum_{i=1}^{N_k} \int_0^t H_{ik}(s)\,d\langle M_{ik}\rangle(s) = \frac{1}{N_k} \sum_{i=1}^{N_k} \int_0^t H_{ik}(s)\,dM_{ik}(s).$$

*Then:*

$$R_{N_k}(t) = o_p(1) \quad \text{for each fixed } t, \quad \sup_{t \leq G_k} |R_{N_k}(t)| = o_p(1).$$

*In particular, for any bounded predictable $H_{ik}$,*

$$\frac{1}{N_k} \sum_{i=1}^{N_k} \int_0^t H_{ik}(s)\,dU_{ik}(s) = \frac{1}{N_k} \sum_{i=1}^{N_k} \int_0^t H_{ik}(s)\,d\langle M_{ik}\rangle(s) + o_p(1).$$

*Proof.* By (A1), $U_{ik} = \Lambda_{ik} + M_{ik}$ and $d\langle M_{ik}\rangle = d\Lambda_{ik}$, hence

$$R_{N_k}(t) = \frac{1}{N_k} \sum_{i=1}^{N_k} \int_0^t H_{ik}(s)\,dM_{ik}(s).$$

Using orthogonality in (A2) and the isometry for martingale integrals,

$$\mathbb{E}\left[R_{N_k}(t)^2\right] = \frac{1}{N_k^2} \sum_{i=1}^{N_k} \mathbb{E}\left[\int_0^t H_{ik}(s)^2\,d\langle M_{ik}\rangle(s)\right] \leq \frac{C_H^2}{N_k^2} \sum_{i=1}^{N_k} \mathbb{E}\left[\int_0^{G_k} V_{ik}(s)\lambda_{ik}(s)\,ds\right] \leq \frac{C_H^2 C_\lambda G_k}{N_k}.$$

Thus $R_{N_k}(t) \xrightarrow{p} 0$ for each fixed $t$. For the uniform version, Doob's $L^2$ inequality yields

$$\mathbb{E}\left[\sup_{t \leq G_k} |R_{N_k}(t)|^2\right] \leq 4\mathbb{E}\left[R_{N_k}(G_k)^2\right] \leq \frac{4C_H^2 C_\lambda G_k}{N_k} \longrightarrow 0,$$

hence $\sup_{t \leq G_k} |R_{N_k}(t)| = o_p(1)$. $\qquad\qquad\square$

*Remark* B.11. If we define $d\Lambda_{0,k}(s) = \lambda_{0,k}(s)ds$, then in the Cox PH model, $\lambda_{ik}(s) = \lambda_{0,k}(s)e^{\boldsymbol{\eta}_k^\top \boldsymbol{Z}_{ik}}$ so $d\langle M_{ik}\rangle(s) = V_{ik}(s)e^{\boldsymbol{\eta}_k^\top \boldsymbol{Z}_{ik}}d\Lambda_{0,k}(s)$; In the Aalen additive model, $\lambda_{ik}(s) = \lambda_{0,k}(s) + \boldsymbol{\beta}_k^\top \boldsymbol{Z}_{ik}$ so $d\langle M_{ik}\rangle(s) = V_{ik}(s)\{\lambda_{0,k}(s) + \boldsymbol{\beta}_k^\top \boldsymbol{Z}_{ik}\}ds$. Lemma B.10 justifies replacing these predictable quadratic variations by the corresponding empirical averages of $dU_{ik}$ in large samples when evaluating variances/covariances of martingale integral terms.

