# OpenReview forum: "Beyond Average Value Function in Precision Medicine: Maximum Probability-Driven Reinforcement Learning for  Survival Analysis"
_NeurIPS.cc/2025/Conference — NeurIPS 2025 poster_

### Official Review · Reviewer_fiaA · 2025-07-01

**Clarity:** 3
**Significance:** 2
**Originality:** 2
**Rating:** 4
**Confidence:** 4

**Summary:**

The paper considers RL to maximize the probability of durations of being in the good state being greater than some threshold. This objective decomposes and they take logs to maximize the sum of log survival probabilities.

**Questions:**

Questions:
- How should the threshold be chosen? If it's not clear, then isn't this more or less the same as optimizing duration?
- Why not max the durations of being in the good state directly? Since this problem maximizes the sum of log Survival functions. Why not just maximize the expected durations per stage?
- However, in survival analysis, we are more interested in focusing on the probability that the recurrent-event duration is greater than a certain threshold.

Can you please explain more why this is relevant? I see that this might come up a lot in survival analysis but it's not apparent why this is a decision-theoretically relevant quantity to the decision-maker.

**Ethical Concerns:**

["NO or VERY MINOR ethics concerns only"]

**Final Justification:**

The paper makes advances on tackling survival-type objectives with recurrent events. It has some limitations still, namely the arbitrariness around selecting alpha, and some of the discussions in the rebuttal phase would have improved the paper if it were submitted with that level of detail and context.

**Quality:**

3

**Strengths And Weaknesses:**

strengths:

significance:

- the paper presents a specific challenge of alternating recurrent events and optimizes the probability that the recurrent-event duration within a time period is greater than a certain threshold (such as greater than 4/7 time of the time being in a good state). Because this is a joint
- the paper nicely leverages standard survival analysis tools and censoring, which are common practical challenges in important settings.



weaknesses:

- significance: the choice of reward function seems somewhat driven by tractability. It doesn't seem appropriate from a first-principles point of view. What objectives that a decision-maker might care about correspond to the probability of duration increasing beyond a threshold? This seems to incentivize policies that "satisfice" (i.e. just achieve the threshold + one standard deviation of conditional duration times.) The authors compare computationally to maximizing the duration fraction via the work of Liu et al 2023 via finite-stage dynamic treatment regimes. But it's not clear if this is a weak baseline, i.e. if instead Liu et al 2023 considers the "right" objective with a "wrong" discretization via finite-stage DTR.

- relevance: the set-up of alternate recurrent events seems somewhat narrow. It also seems that this can be handled with discrete-time MDPs as in this paper because the time of taking actions is fixed. So it seems somewhat overindexed to the intern RCT data, i.e. measurement and action of an underlying CTMC (that generates the underlying durations) at a fixed rate. This seems a bit of an approximation to the underlying goal, which is the duration fraction on the underlying CTMC. It would be nice to justify the correspondence here a bit.

The authors propose a joint probability objective: $P(T_t > \alpha_t G_t, ..., T_K > \alpha_K G_K \mid X_t)$ which decomposes nicely into a product of survival probabilities and the authors optimize the log of this. However, it's not clear why they optimize the joint probability objective rather than just the long-run average occupancy time in the "good" state, i.e. $E_\pi[I[s = good]]$

Clarity:

- figure 1 is confusing and is not self-explanatory; the given explanation is not sufficient.
- typos:
We first proposed a framework in RL that aims to maximize the probability, and applied it to recurrent events data. Meanwhile, we innovatively provided the division definition of recurrent events data

---

> ### Author Rebuttal · Authors · 2025-07-31
>
> **Response to Reviewer Comments**
>
> We sincerely appreciate your valuable feedback. Below are our responses and revisions addressing the questions you raised.
>
> ---
>
> ### 1. Figure 1 Clarification
> **Reviewer Comment**:
> "Figure 1 is confusing and is not self-explanatory; the given explanation is not sufficient."
> **Response**:
> Thank you for your suggestion. We will incorporate the descriptions from lines 115–138 into the figure to improve clarity.
> **Revision**:
> - $C \in \mathbb{R}^+$: Censoring time.
> - $Y_k$: Observed duration at stage $k$.
> - $\Delta_k$: Censoring indicator at stage $k$.
> - $\bar{K} \in \mathbb{N}^+$: Number of treatment stages a subject received.
> - $T_k$: Actual duration of stage $k$.
>
> ---
>
> ### 2. Threshold Selection
> **Reviewer Comment**:
> "How should the threshold be chosen? If it's not clear, then isn't this more or less the same as optimizing duration?"
> **Response**:
> Thank you for the question. The selection of $\alpha$ is tailored to the specific research question being addressed.  For example, in Jiang, et al. (2017)'s study on HIV data, they chose $\alpha = 0.5$, focusing on the optimal strategy for maximizing the probability of survival exceeding the median. In our study, we selected $\alpha = 1/14$ to maximize the probability of an intern's positive mood exceeding one day per bi-weekly period. We generally advise using an $\alpha$ of 0.5 or less, mainly because: (1) $\alpha$ values near 1 rely heavily on tail probability estimation, which can be unreliable; (2) to address your concern, we have conducted an additional  simulation experiment to demonstrate the sensitivity of our algorithm to different $\alpha$ values. The results reveal that our method performs better when $\alpha$ is 0.5 or lower.
>
> **Revision**:
> We will add a new appendix section before A.2 titled:
> *"**A.2 $\alpha$ Selection***
>
> **Table 1: Mean (Standard Deviation) of AARED under different $\alpha_k$ (rounded to two decimal places)**
>
> | $\alpha_k$ | Aah Model       | Cox Model       | Baseline model  |
> |------------|-----------------|-----------------|-----------------|
> | 1/14       | 19.93 (0.01)    | **19.94 (0.01)**| 19.83 (0.01)    |
> | 1/7        | **19.95 (0.01)**| 19.92 (0.01)    | 19.83 (0.01)    |
> | 3/14       | 19.17 (0.03)    | **19.89 (0.02)**| 19.83 (0.01)    |
> | 2/7        | 18.99 (0.05)    | 19.83 (0.02)    | **19.83 (0.01)**|
> | 3/7        | 18.98 (0.05)    | 19.79 (0.02)    | **19.83 (0.01)**|
> | 1/2        | 19.00 (0.04)    | 19.79 (0.02)    | **19.83 (0.01)**|
> | 4/7        | 19.01 (0.05)    | 19.76 (0.03)    | **19.83 (0.01)**|
> | 5/7        | 19.00 (0.04)    | 19.70 (0.03)    | **19.83 (0.01)**|
> | 6/7        | 19.02 (0.05)    | 19.64 (0.05)    | **19.83 (0.01)**|
> | 1          | 19.06 (0.05)    | 19.69 (0.03)    | **19.83 (0.01)**|
>
>
> **Table 2: AAEV values (Mean ± Standard Deviation) under different $\alpha_k$ (rounded to two decimal places)**
>
> | $\alpha_k$ | Aah Model        | Cox Model        | Baseline model   |
> |------------|------------------|------------------|------------------|
> | 1/14       | **10.54 (0.00)** | 11.95 (2.39)     | 20.00 (19.47)    |
> | 1/7        | **10.83 (0.08)** | 13.98 (4.46)     | 20.00 (19.47)    |
> | 3/14       | 64.55 (37.94)    | **17.74 (12.51)**| 20.00 (19.47)    |
> | 2/7        | 78.16 (14.46)    | 20.66 (16.60)    | **20.00 (19.47)**|
> | 3/7        | 78.57 (14.25)    | 23.54 (27.27)    | **20.00 (19.47)**|
> | 1/2        | 78.62 (13.81)    | 24.41 (35.90)    | **20.00 (19.47)**|
> | 4/7        | 78.63 (14.35)    | 25.10 (50.60)    | **20.00 (19.47)**|
> | 5/7        | 78.48 (17.63)    | 29.44 (70.17)    | **20.00 (19.47)**|
> | 6/7        | 78.06 (19.74)    | 32.30 (104.38)   | **20.00 (19.47)**|
> | 1          | 71.95 (83.77)    | 29.41 (76.78)    | **20.00 (19.47)**|
>
> To clarify our recommendation for alpha selection, we will include the following remark on page 4, line 153: *”The selection of $\alpha$ is tailored to the specific research question being addressed. For example, in Jiang et al. (2017)‘s study on HIV data, they chose $\alpha = 0.5$, focusing on the optimal strategy for maximizing the probability of survival exceeding the median. In our study, we selected $\alpha = 1/14$ to maximize the probability of an intern’s positive mood exceeding one day per bi-weekly period. We generally advise using an $\alpha$ of 0.5 or less, mainly because $\alpha$ values near 1 rely heavily on tail probability estimation, which can be unreliable. These results are consistent with the numerical results we obtained from our experiment results (see Appendix A.2 for details).“*
>
> ---
>
> ### 3. Maximizing Duration vs. Survival Probability
> **Reviewer Comment**:
> "Why not maximize the durations of being in the good state directly? Since this problem maximizes the sum of log Survival functions. Why not just maximize the expected durations per stage?"
>
> **Response**:
>
> 1. Challenges with Censored Data: Estimating durations in the “good state” becomes significantly more challenging when dealing with censored data, as complete information about the duration is not always available.
> 2. Limited Reward Identifiability: In discrete settings, the binary nature of the rewards (1 or 0) lacks the discriminative power needed to effectively differentiate between various actions and their impact on stage duration.
> 3. Focus on Cumulative Benefit: Our primary objective is to maximize the cumulative benefit over the entire process, rather than focusing solely on stage-specific rewards. This requires a framework that considers the long-term impact of actions.
> 4. Robustness of the Method: We chose to use the survival probabilities as our objective precisely because of its robustness. Existing methods (e.g., Liu et al. (2023)) maximize expected durations using Kaplan-Meier estimates, but expectations are sensitive to outliers. Extreme survival times can skew averages, leading to a misrepresentation of actual patient benefits.
>
> **Revision**:
> We will add after line 51:
> *"The challenges in relevant research include the difficulty of estimating durations in the "good state" due to censored data, the limited discriminative power of binary rewards in discrete settings for differentiating action impacts, and the need to focus on cumulative benefits. Additionally, maximizing expectations is sensitive to outliers—extreme survival times can inflate averages, misrepresenting actual patient benefits (Bai et al., 2013; Jiang et al., 2017a, 2017b). Thus, we explore maximizing survival probabilities, which better reflect clinical outcomes."*
>
> ---
>
> ### 4. Relevance of Threshold-Based Probability in Survival Analysis
> **Reviewer Comment**:
> "In survival analysis, we focus on the probability that the recurrent-event duration exceeds a threshold. Can you explain why this is decision-theoretically relevant? It’s unclear why this quantity matters to decision-makers."
>
> **Response**:
> Thank you for your question. The importance of this quantity can be explained from both theoretical and practical perspectives. Theoretically, compared to traditional methods that focus on the mean survival time, this quantity is more robust and less sensitive to outliers, providing a more stable representation of the patients outcome. Furthermore, by studying the survival curve, our method can characterize the impact of treatment at different time points. From a practical application perspective, the study of this quantity is relevant to many medical problems and has been investigated by some existing literature. We provide some specific examples below.
>
> 1. **Bai et al. (2013)**: In the ASCERT study, CABG and PCI showed crossing survival curves—PCI had lower short-term risk, but CABG outperformed after 4 years. Here, 4-year survival probability precisely captured long-term differences, whereas average survival time obscured this.
> 2. **Jiang et al. (2017a)**: In ACTG 175, young patients on zidovudine + zalcitabine had significantly higher 400-day survival probabilities. Maximizing $ t $-year survival probability aligned better with short-term clinical needs than averaging.
> 3. **Jiang et al. (2017b)**: For HIV patients, median survival time (50% probability) was more representative of typical outcomes than restricted mean survival time, as it is less influenced by extreme values.
>
> **Revision**:
> We will add after line 35:
> "In survival analysis, expectations (e.g., mean survival time) are sensitive to outliers and may obscure time-specific treatment effects. Threshold-based probabilities (e.g., median survival) better align with clinical needs. For example:
> - Bai (2013) found CABG’s 4-year survival probability outperformed PCI’s, despite crossing survival curves.
> - Jiang (2017a) showed 400-day survival probability was critical for evaluating short-term HIV treatment efficacy.
> - Jiang (2017b) demonstrated median survival time’s robustness for typical patient outcomes in HIV studies.
> These studies validate that probability-based metrics overcome the limitations of expectations."
>
> ---
>
> ### References
> 1. Bai, X., Tsiatis, A. A., and O’Brien, S. M. (2013). "Doubly-Robust Estimators of Treatment-Specific Survival Distributions in Observational Studies With Stratified Sampling." *Biometrics*, 69, 830–839.
> 2. Jiang, R., Lu, W., Song, R., and Davidian, M. (2017a). "On Estimation of Optimal Treatment Regimes for Maximizing t-Year Survival Probability." *Journal of the Royal Statistical Society, Series B*, 79, 1165–1185.
> 3. Jiang, R., Lu, W., Song, R., Hudgens, M. G., and Naprvavnik, S. (2017b). "Doubly Robust Estimation of Optimal Treatment Regimes for Survival Data—With Application to an HIV/AIDS Study." *The Annals of Applied Statistics*, 11, 1763–1786.
>
> ---
> ### Acknowledgments
> We have addressed all of the reviewers' comments in this revision, offering clear explanations, empirical support, and specific revisions to the manuscript. Please advise if further clarifications are needed.

---

> ### Author Response · Authors · 2025-08-06
>
> Dear Reviewer fiaA,
>
> We submitted our response to your initial comments some time ago, and we wanted to check in briefly.
>
> We deeply appreciate the time and effort you’ve dedicated to evaluating our work, especially given your busy schedule. Your feedback is invaluable as we work to refine our manuscript, and we’re eager to receive your further thoughts to guide our revisions and move the submission forward.
>
> If you’ve had an opportunity to review our response, we would greatly appreciate your input whenever you have a moment; if not, we hope you might be able to take a look when your schedule allows.
>
> Thank you again for your time and attention. We look forward to hearing from you.
>
> Sincerely,
>
> Authors

---

> > ### Comment · Reviewer_fiaA · 2025-08-06
> >
> > Thanks for the rebuttal. Just to be clear, I appreciate the additional experiments; however there are still some fundamental justifications of the method that seem lacking for the threshold selection. For example, your cited studies show that there are some thresholds that result in better evaluations; but your justification of 2) says that they should be tailored to domain considerations at hand. These are quite different objectives, and one could argue that something that should be expected to work for all thresholds seeks a distribution. This, and the use of log vs. not-log, still feels quite confusing.
> >
> >  I do not have further questions at this time, thanks.

---

> > > ### Author Response · Authors · 2025-08-07
> > >
> > > ### Question 1
> > > **Comments**: For example, your cited studies show that there are some thresholds that result in better evaluations; but your justification of 2) says that they should be tailored to domain considerations at hand. These are quite different objectives, and one could argue that something that should be expected to work for all thresholds seeks a distribution.
> > >
> > > **Response**: Thank you for your comment. Here, $\alpha$ is not a parameter that can be estimated by machine learning, nor is it a tuning parameter that can be selected by criteria. Jiang (2017ab) and Bai (2013) both selected pre-specified constants based on clinical experience for strategy analysis in their real data examples. The framework we propose can actually be applied to all problems of strategy formulation under the objective of maximizing probability. By selecting different $\alpha$ values, it only requires providing time points based on clinical experience.
> > >
> > > - In Jiang's paper, $\alpha = 0.5$ was adopted in the simulation part only because this value was considered more representative of the average level and more meaningful rather than having better fitting performance. In its experiments, the ACTG Study focused on survival probabilities at time points such as 600 days and 800 days. Bai (2013) used 4 years as the key time point in the clinical trial of CABG, and Jiang even selected 4000 days as the time point in the study on HIV/AIDS.
> > >
> > > - Collett (2015) stated that the survival function can exhibit increasing, decreasing, or "bathtub-shaped" (first decreasing then increasing) patterns with respect to the time point t, directly reflecting differences in risk at different time points. For instance, cancer patients may have a higher risk in the early stage of treatment, which then decreases, and may increase again in the advanced stage due to recurrence. Therefore, the selection of different t values has different implications for strategy formulation, and this is generally done based on clinical experience.
> > >
> > > Here, we present all strategies for $\alpha$ within a certain interval to address strategy formulation based on different clinical experiences. Thus, different $\alpha$ values need to be chosen based on clinical experience, while our paper provides a general reinforcement learning strategy formulation framework applicable to all $\alpha$ values.
> > >
> > > **Revision**: We will include the following remark on page 4, line 153: "The selection of $\alpha$ is tailored to the specific research question being addressed. Here $\alpha$ is not a parameter that can be estimated by machine learning, nor is it a tuning parameter that can be selected by criteria. In practical cases (Bai et al. (2013), Jiang et al. (2017)), $\alpha$ needs to be determined based on clinical experience. For example, the ACTG study focused on survival probabilities at time points such as 600 days and 800 days, and the CABG clinical trial adopted 4 years as the key time point. In our study, we selected $\alpha$ to maximize the probability of an intern’s positive mood exceeding one day per bi-weekly period. We generally advise using an $\alpha$ of 0.5 or less, mainly because $\alpha$ values near 1 rely heavily on tail probability estimation, which can be unreliable. These results are consistent with the numerical results we obtained from our experiment results (see Appendix A.2)."
> > >
> > > ### Question 2
> > > **Comments**: This, and the use of log vs. not-log, still feels quite confusing.
> > >
> > > **Response**: The inclusion of the log transformation is because we aim to use reinforcement learning for strategy formulation in scenarios with multiple stages. The maximization of joint distribution probability can only be converted into a product form, and by introducing the log transformation, it can be converted into a summation form, which is applicable to the reinforcement learning framework. Moreover, our Theorem 3.2 illustrates that the optimal strategy after the transformation remains consistent with that before the transformation.
> > >
> > > **Revision**: We will add the following at line 167: ”By introducing the logarithmic transformation, we can convert the product-form objective into a summation form, which thus can be adapted to all reinforcement learning frameworks. This transformation does not affect the selection of the optimal strategy, as demonstrated in our Theorem 3.2.“
> > >
> > > ---
> > > - Collett, D. (2015). 《Modelling Survival Data in Medical Research》
> > >
> > > - Bai, X. (2013). "Doubly-Robust Estimators of Treatment-Specific Survival Distributions in Observational Studies With Stratified Sampling." Biometrics, 69, 830–839.
> > >
> > > - Jiang, R. (2017a). "On Estimation of Optimal Treatment Regimes for Maximizing t-Year Survival Probability." Journal of the Royal Statistical Society, Series B, 79, 1165–1185.
> > >
> > > - Jiang, R. (2017b). "Doubly Robust Estimation of Optimal Treatment Regimes for Survival Data—With Application to an HIV/AIDS Study." The Annals of Applied Statistics, 11, 1763–1786.

---

> > > ### Author Response · Authors · 2025-08-08
> > >
> > > Dear Reviewer fiaA,
> > >
> > > I hope this message finds you well.
> > >
> > > I wanted to kindly follow up on the additional clarifications I provided regarding your questions. Please accept my sincere thanks again for your valuable insights and guidance throughout this process—your feedback has been instrumental in strengthening our work.
> > >
> > > I understand you are likely very busy, but if you have a moment to share any further thoughts or suggestions, I would greatly appreciate it. Please feel free to let me know if more details are needed from my side.
> > >
> > > Thank you once more for your time and consideration.
> > >
> > > Sincerely,
> > >
> > > Authors

---

### Official Review · Reviewer_2BRd · 2025-07-02

**Clarity:** 3
**Significance:** 2
**Originality:** 2
**Rating:** 4
**Confidence:** 3

**Summary:**

This paper optimizes multistage decision-making for alternating recurrent event data in survival analysis. By transforming recurrent event and censored data into a probability objective, the authors apply traditional deep reinforcement learning (DRL) algorithms to maximize returns. Experiments show that the DRL method, combined with various survival functions, achieves faster convergence and higher probability of recurrent-event durations in both simulated and real data.

**Questions:**

1) It seems that some of the theoretical analysis assumes a discrete action space, such as Theorem 3.2 and the assumption in lines 213-214, which is hard to be met in a continuous space. Do these assumptions still apply in a continuous action space, as used in DDPG?

2) Despite the rigorous theoretical analysis, the code implementation is relatively simple. Does the core contribution lie in treating the logarithm of the survival probability given a specific threshold as the immediate reward?

3) Could you explain the changes in the relative log probability shown in Figure 4? Specifically, why does it increase from 2/14 to 8/14 and then decrease? Does the absolute log probability follow the same trend?

**Ethical Concerns:**

["NO or VERY MINOR ethics concerns only"]

**Final Justification:**

Thanks to the authors for their reply. My concerns regarding the theoretical analysis have been well addressed in the rebuttal. I appreciate the contribution of incorporating recurrent data into the Q-function and would like to raise my score.

**Limitations:**

YES

**Quality:**

4

**Strengths And Weaknesses:**

**Strengths**

1) This paper addresses the multistage decision-making problem by maximizing the probability that the duration exceeds a given threshold, offering a more practical approach than traditional RL objectives in survival analysis.

2) The paper is well-structured and clearly motivated.

3) Experimental results demonstrate clear improvements in both convergence speed and the probability of longer recurrent-event durations.

4) The supplementary codebase helps in understanding the method.


**Weaknesses**

1) This work adapts the probability-based objective to the traditional RL framework with minimal modification, which may limit its impact in areas where RL is already widely used.

2) Some of the theoretical analysis is difficult to understand. Please refer to the questions for further clarification. I would like to increase my score once these questions are properly addressed.

---

> ### Author Rebuttal · Authors · 2025-07-31
>
> **Response to Reviewer Comments**
>
> We sincerely appreciate your valuable feedback. Below are our responses and revisions addressing the questions you raised.
>
> ---
> ### 1. Continuous Action Space
>
> **Reviewer Comment**:It seems that some of the theoretical analysis assumes a discrete action space, such as Theorem 3.2 and the assumption in lines 213-214, which is hard to be met in a continuous space. Do these assumptions still apply in a continuous action space, as used in DDPG?
>
> **Response**: Thank you for pointing this out. Although the action space in the paper was formulated as discrete, the objective itself is applicable to both discrete and continuous action spaces. Following your advice, we have revised all relevant content in the paper (shown in the revision) to discuss discrete and continuous action spaces separately.
>
> **Revision**:
> - We will change *"**Theorem 3.2**"* to *"**Theorem 3.2 (Discrete action space)**"*, and then add a new theorem after line 170:*"
> **Theorem 3.3 (Continuous action space)**
> Let the state transition kernel $ P_0(x'|x,a) $ be deterministic (i.e., for any $ x,a $, there exists a unique $ x' $ such that $ P_0(x'|x,a)=1 $). Let the policy $ \pi $ be represented by a deterministic function $ \mu $ (i.e., $ a_k = \mu(x_k) $), and for any initial state $ x_t $, the supremum of the value function $ V^{\mu}(x_t) $ is attainable. Define the policy sets: $ \Pi^v = ${$ \mu^v \mid V_0^{\mu^v}(x_t) = \text{max}({\mu}) V_0^{\mu}(x_t) $} , $ \Pi^p = ${$ \mu^p \mid P^{\mu^p}(x_t) = \text{max}({\mu}) P^{\mu}(x_t) $} , then $ \Pi^v = \Pi^p $."*\
> **Proof:** Given a deterministic transition kernel $ P_0 $ and a deterministic policy $ \mu $, starting from any state $ x_t $, there exists a unique trajectory $ \gamma_t^\mu = (x_t, \mu(x_t), T_t, x_{t+1}^\mu, \mu(x_{t+1}^\mu), T_{t+1}, \dots, x_{\bar{K}}^\mu, \mu(x_{\bar{K}}^\mu), T_{\bar{K}}) $, where $ x_{k+1}^\mu $ is uniquely determined by the transition equation. Based on the above unique trajectory, the reward of trajectory $ \gamma_t^\mu $ can be expressed as $ R(\gamma_t^\mu) = \prod_{k=t}^{\bar{K}} S_{T_k}(\alpha_k G_k \mid x_k^\mu, \mu(x_k^\mu)) $. Correspondingly, the objective functions can be simplified to $ P^\mu(x_t) = R(\gamma_t^\mu) $ and $ V_0^\mu(x_t) = \log R(\gamma_t^\mu) $.
> Define the set of optimal trajectories $ \Gamma_t^\* = $ { $ \gamma_t^\* \mid R(\gamma_t^\*) = \text{max}({\gamma_t \in \Gamma_t}) R(\gamma_t) $}, where $ \Gamma_t $ is the set of all possible trajectories. Since $ \log(\cdot) $ is a strictly monotonically increasing function, $ \text{max}(\mu) R(\gamma_t^\mu)$ is equivalent to $ \text{max}(\mu) \log R(\gamma_t^\mu) $. It follows that $ \mu \in \Pi^p \iff R(\gamma_t^\mu) = \text{max}({\gamma_t \in \Gamma_t}) R(\gamma_t) \iff \log R(\gamma_t^\mu) = \text{max}({\gamma_t \in \Gamma_t}) \log R(\gamma_t) \iff \mu \in \Pi^v $, i.e., $ \Pi^v = \Pi^p = ${$ \mu \mid \gamma_t^\mu \in \Gamma_t^\* $} $, which proves the equivalence of optimal policies.
>
> - Line 211 will be changed to *"4.2 Optimization in discrete action space"*, followed by adding *"4.3 Optimization in continuous action space
> According to DDPG (Lillicrap et al., 2015), we define the Policy Function: $ \mu(\theta): \mathcal{S} \to \mathcal{A} $, a parameterized function (e.g., neural network) that maps states to actions. For a fixed policy $ \mu(\theta) $, the Q-function follows the Bellman evaluation equation:
> $Q^{\mu(\theta)}(\tilde{X}_t, \tilde{A}_t) = \mathbb{E}^{P_0} [(\tilde{X}_t, \tilde{A}_t) + \gamma Q^{\mu(\theta)} (\tilde{X} _{t+1}, \mu(\theta)(\tilde{X} _{t+1})) ]$
> To optimize $ \mu(\theta) $, we maximize the expected return $ J(\theta) = \mathbb{E}[Q^{\mu(\theta)}(\tilde{X}_t, \tilde{A}_t)] $. Then, parameter updates are performed using the chain rule in DDPG.
> “*
>
> ---
> ### 2. Core contribution
>
> **Reviewer Comment**: Thank you for the question. Despite the rigorous theoretical analysis, the code implementation is relatively simple. Does the core contribution lie in treating the logarithm of the survival probability given a specific threshold as the immediate reward?
>
> **Response**: Our core contribution lies in constructing a novel framework that integrates offline RL with recurrent event data. Furthermore, we propose a new recurrent data-driven Q-function and prove that the proposed practical algorithm is consistent with classical Q-learning, rather than using the logarithm of the survival probability given a specific threshold as the immediate reward.
>
> **Revision**: For improved clarity regarding our core contribution, we have included a sentence on line 67:*"Our core contribution is developing a novel framework that integrates offline RL with recurrent data, proposing a new recurrent data-driven Q-function, and proving our algorithm aligns with classical Q-learning."*
>
> ---
> ### 3. Figure 4 Performence
>
> **Reviewer Comment**: Could you explain the changes in the relative log probability shown in Figure 4? Specifically, why does it increase from 2/14 to 8/14 and then decrease? Does the absolute log probability follow the same trend?
>
> **Response**:Thank you for your question. In Figure 4, we previously presented the relative difference in log probabilities (calculated as $\frac{\log \hat{\text{P}}^{\pi^A} - \log \hat{\text{P}}^{\pi^B}}{\log \hat{\text{P}}^{\pi^B}}$ for the Aah model and $\frac{\log \hat{\text{P}}^{\pi^C} - \log \hat{\text{P}}^{\pi^B}}{\log \hat{\text{P}}^{\pi^B}}$ for the Cox model). To better address your question, we have now presented the absolute difference in log probabilities in the revision below. What is shown here is the relative magnitude, which does not have a clear pattern. The absolute probability values do not change in this way. As can be seen below, in fact, as $ \alpha $ increases, the absolute difference ($\log \hat{\text{P}}^{\pi^A} - \log \hat{\text{P}}^{\pi^B}$ for the Aah model and $\log \hat{\text{P}}^{\pi^C} - \log \hat{\text{P}}^{\pi^B}$ for the Cox model) becomes larger and larger, indicating that our performance is getting better.
>
> | $\alpha$ | Cox       | Aah       |
> |----------|-----------|-----------|
> | 0.07     | 3.54e-04  | 2.50e-04  |
> | 0.14     | 2.46e-04  | 1.08e-04  |
> | 0.21     | 1.24e-03  | 6.16e-04  |
> | 0.29     | 4.37e-03  | 2.71e-03  |
> | 0.36     | 1.03e-02  | 6.73e-03  |
> | 0.43     | 3.84e-02  | 2.81e-02  |
> | 0.50     | 1.16e-01  | 8.50e-02  |
> | 0.57     | 3.07e-01  | 2.24e-01  |
> | 0.64     | 4.36e-01  | 3.04e-01  |
> | 0.71     | 7.29e-01  | 4.97e-01  |
> | 0.79     | 7.83e-01  | 5.24e-01  |
> | 0.86     | 9.54e-01  | 6.52e-01  |
> | 0.93     | 8.43e-01  | 5.93e-01  |
> | 1.00     | 9.13e-01  | 6.65e-01  |
>
>
> **Revision**: We will revise lines 327-329 in the paper to *"We then presented the log probability differences $\log \hat{\text{P}}^{\pi^A} - \log \hat{\text{P}}^{\pi^B}$ for the Aah model and $\log \hat{\text{P}}^{\pi^C} - \log \hat{\text{P}}^{\pi^B}$ across varying $\alpha_k$ settings."* and then replace the figure with the data shown in the table.
>
> ---
>
> ### References
> Lillicrap, T. P., Hunt, J. J., Pritzel, A., Heess, N., Erez, T., Tassa, Y., Silver, D., and Wierstra, D. Continuous control with deep reinforcement learning. arXiv preprint arXiv:1509.02971, 2015.
>
> ---
> ### Acknowledgments
> All comments from the reviewers have been thoroughly addressed in this revision, with explicit explanations, empirical support, and precise modifications made to the manuscript. Do not hesitate to inform us if additional clarifications are needed.

---

> > ### Comment · Reviewer_2BRd · 2025-08-07
> >
> > Thanks to the authors for their reply. My concerns regarding the theoretical analysis have been well addressed in the rebuttal. I appreciate the contribution of incorporating recurrent data into the Q-function and would like to raise my score.

---

> ### Author Response · Authors · 2025-08-06
>
> Dear Reviewer 2BRd,
>
> We submitted our response to your initial comments some time ago, and we wanted to gently follow up.
>
> We greatly appreciate the time and effort you’ve dedicated to reviewing our work, especially given your busy schedule. Your insights are invaluable for refining our manuscript, and we’re eager to receive your further feedback to advance revisions and move the submission forward.
>
> If you’ve had a chance to review our response, we would be grateful for your thoughts at your convenience; if not, we hope you might have a moment to take a look when time permits.
>
> Thank you again for your consideration. We look forward to your reply.
>
> Sincerely,
>
> Authors

---

### Official Review · Reviewer_VWkH · 2025-07-03

**Clarity:** 1
**Significance:** 2
**Originality:** 3
**Rating:** 3
**Confidence:** 2

**Summary:**

The work approaches the recurrent event data from a different angle: the probability of event occurrences. The authors tried to reformulate the problem as an equivalent MDP. Numerical studies are conducted to validate the effectiveness of the approach.

**Questions:**

See weakness.

**Ethical Concerns:**

["NO or VERY MINOR ethics concerns only"]

**Quality:**

2

**Strengths And Weaknesses:**

- Strength

The authors proposed a more practical view of recurrent event scenarios and describe the whole picture in a well-rounded way. Empirical studies were conducted to validate the effectiveness of the proposed approach.

- Weakness

1. The review is not convinced by the key theoretical theorem 2. How does adding an extra log in the objective still give the same optimal solution? The review is not convinced by the proof in Appendix 9, it seems to state that a greedy action simply solve the MDP. Plus, the reviewer believe that the result in Figure 2 also proves that adding an extra log will give different optimal solutions. The authors are encouraged to make this part more clear as the equivalence claims are build upon it.

2. As the reviewer understand, the offline evaluation part is based on the simulation using the Cox. Will such evaluation cause biased results towards the proposed method based on Cox model?

---

> ### Author Rebuttal · Authors · 2025-07-31
>
> **Response to Reviewer Comments**
>
> We sincerely appreciate your valuable feedback. Below are our responses and revisions addressing the questions you raised.
>
> ---
> ### 1. Theoretical Justification of Theorem 2
> **Reviewer Comment**: The review is not convinced by the key theoretical theorem 2. How does adding an extra log in the objective still give the same optimal solution? The review is not convinced by the proof in Appendix 9, it seems to state that a greedy action simply solve the MDP. Plus, the reviewer believe that the result in Figure 2 also proves that adding an extra log will give different optimal solutions. The authors are encouraged to make this part more clear as the equivalence claims are build upon it.
>
> **Response**:
> We greatly appreciate your comments. Following your suggestion,  we carefully examined the theoretical section and identified a missing condition in Theorem 3.2: the deterministic property of $P_0$. We emphasize that, because our work focuses on offline RL, this condition is naturally met, and therefore the correction to Theorem 3.2 does not affect the overall properties of our proposed framework.  In the revised manuscript, we will update Theorem 3.2 and provide its complete proof as below.
>
> **Revision**:
> We will revise the theorem in line 169 of the paper to:
>
> *"**Theorem 3.2.** If for any $ \boldsymbol{x}_t $, the supremum of $ V^{\pi}(\boldsymbol{x}_t) $ can be attained, and when $ P_0(\boldsymbol{x}'|\boldsymbol{x},a) $ is a deterministic transition kernel, with $\Pi^v = ${$\pi^v | V_0^{\pi^v}(\boldsymbol{x}_t) = \text{max}(\pi) V_0^{\pi}(\boldsymbol{x}_t) $} and $\Pi^p =$ {$\pi^p | \text{P}^{\pi^p}(\boldsymbol{x}_t) = \text{max}(\pi)\text{P}^{\pi}(\boldsymbol{x}_t)$}, then $ \Pi^v = \Pi^p $.*
>
> *（$ P_0(\boldsymbol{x}'|\boldsymbol{x},a) $ is a deterministic transition kernel if and only if $ \forall \boldsymbol{x},a $, there exists a unique $ \boldsymbol{x}' $ such that $ P_0(\boldsymbol{x}'|\boldsymbol{x},a)=1 $ ）"*
>
> Furthermore, we will revise the proof of Theorem 3.2 in lines 548 to 556 in Appendix to:
>
> "**Proof:**
> For any deterministic policy $ P_0 $, by definition, there exists a unique $ x_{t+1}^{P_0} $ such that:
> $$
> P_0(x_{t+1}^{P_0} \mid x_t, a_t) = 1
> $$
> We denote $ x_t^{P_0} = x_t $ and define
> $$
> R(\gamma_t) = \prod_{k=t}^K S_{T_k}(\alpha_k G_k \mid x_k^{P_0}, a_k)
> $$
> where $ \gamma_t $ is the path:
> $$
> \gamma_t = (x_t^{P_0}, a_t,T_t, x_{t+1}^{P_0}, a_{t+1},T_{t+1} \dots, x_{\bar{K}}^{P_0}, a_{\bar{K}},T_{\bar{K}})
> $$
> We define the set of paths as $ \Gamma_t^{P_0} $, then we can obtain:
> $$
> P^\pi(x_t) = \sum_{\gamma_t \in \Gamma_t^{P_0}} \mathcal{P}(\gamma_t) \cdot R(\gamma_t), \quad V_0^\pi(x_t) = \sum_{\gamma_t \in \Gamma_t^{P_0}} \mathcal{P}(\gamma_t) \cdot \log R(\gamma_t)
> $$
> where:
> $$
> \mathcal{P}(\gamma_t) = \pi(a_t \mid x_t^{P_0}) \prod_{k=t}^{\bar{K}-1} P_0(x_{k+1}^{P_0} \mid x_k^{P_0}, a_k) \pi(a_{k+1} \mid x_{k+1}^{P_0})\\
> = \prod_{k=t}^{\bar{K}} \pi(a_k \mid x_k^{P_0})
> $$
> Then we can know that
> $$
> P^\pi(x_t) = \sum_{\gamma_t \in \Gamma_t^{P_0}} \prod_{k=t}^K \pi(a_k \mid x_k^{P_0}) \cdot R(\gamma_t) \leq \text{max} ({\gamma_t \in \Gamma_t^{P_0}}) R(\gamma_t)
> $$
> $$
> V_0^\pi(x_t) = \sum_{\gamma_t \in \Gamma_t^{P_0}} \prod_{k=t}^K \pi(a_k \mid x_k^{P_0}) \cdot \log R(\gamma_t) \leq \text{max}({\gamma_t \in \Gamma_t^{P_0}}) \log R(\gamma_t)
> $$
> where $ \sum_{\gamma_t \in \Gamma_t^{P_0}} \prod_{k=t}^K \pi(a_k \mid x_k^{P_0}) = 1 $.
> If we define:
> $\Gamma_t^{p,\*} = $ {$\gamma_t^\* \mid R(\gamma_t^\*) = \text{max}({\gamma_t \in \Gamma_t^{P_0}}) R(\gamma_t) $}
> and
> $\Gamma_t^{v,\*} = $ {$\gamma_t^o \mid \log R(\gamma_t^\*) = \text{max}({\gamma_t \in \Gamma_t^{P_0}}) \log R(\gamma_t) $}
> Here, since $ a \geq b \iff \log a \geq \log b \ (\text{where } a,b > 0) $, we have $ \Gamma_t^{v,\*} = \Gamma_t^{p,\*} $, which we denote as $ \Gamma_t^{\*} $. Then:
> $$
> P^\pi(x_t) \leq R(\gamma_t^\*), \quad V_0^\pi(x_t) \leq \log R(\gamma_t^\*), \quad \gamma_t^\* \in \Gamma_t^\*.
> $$
> Define the policy set:
> $\Pi^\* = ${$\pi \mid \sum_{\gamma_t \in \Gamma_t^\*} \prod_{k=t}^{\bar{K}} \pi(a_k \mid x_k^{P_0}) = 1$}
> Then we can know that the equality in the inequalities holds if and only if $ \pi \in \Pi^\* $ (see the following proof), so it can be concluded that $ \Pi^v = \Pi^p = \Pi^\* $
> 1. $ P^\pi(x_t) = R(\gamma_t^\*) \Rightarrow \pi \in \Pi^\* $:
> For $ P^\pi(x_t) $, we denote $ R^\* = \text{max}({\gamma_t \in \Gamma_t^{P_0}}) R(\gamma_t) $. Then, for $ \gamma_t \notin \Gamma_t^\* $, we have $ R(\gamma_t) < R^* $. Thus:
> $$
> P^\pi(x_t) = \sum_{\gamma_t \notin \Gamma_t^\*} \mathcal{P}(\gamma_t) \cdot R(\gamma_t) + \sum_{\gamma_t \in \Gamma_t^\*} \mathcal{P}(\gamma_t) \cdot R^\* = R^\*
> $$
> So, $ \sum_{\gamma_t \in \Gamma_t^*} \mathcal{P}(\gamma_t) = 1 $, that is, $ \pi \in \Pi^\* $.
> 2. $ P^\pi(x_t) = R(\gamma_t^\*) \Leftarrow \pi \in \Pi^\* $:
> Prove by contradiction: Assume there exists $ \pi \notin \Pi^\* $, then $ \sum_{\gamma_t \in \Gamma_t^\*} \prod_{k=t}^{\bar{K}} \pi(a_k \mid x_k^{P_0}) < 1 $. We denote $ R^\* = \text{max}({\gamma_t \in \Gamma_t^{P_0}}) R(\gamma_t) $. At this time:
> $$
> P^\pi(x_t) = \sum_{\gamma_t \in \Gamma_t^\*} \mathcal{P}(\gamma_t) \cdot R^\* + \sum_{\gamma_t \notin \Gamma_t^\*} \mathcal{P}(\gamma_t) \cdot R(\gamma_t).
> $$
> Since in the second term, $ R(\gamma_t) < R^\* $ and $ \mathcal{P}(\gamma_t) > 0 $, we have:
> $$
> P^\pi(x_t) < \sum_{\gamma_t \in \Gamma_t^\*} \mathcal{P}(\gamma_t) \cdot R^\* + \sum_{\gamma_t \notin \Gamma_t^\*} \mathcal{P}(\gamma_t) \cdot R^\* = R^\* \cdot \sum_{\gamma_t} \mathcal{P}(\gamma_t) = R^\*,
> $$
> which contradicts "$ P^\pi(x_t) = R(\gamma_t^\*) $."
>
> ---
> ### 2. Potential Evaluation Bias Using Cox Model
> **Reviewer Comment**: As the reviewer understand, the offline evaluation part is based on the simulation using the Cox. Will such evaluation cause biased results towards the proposed method based on Cox model?
>
> **Response**:
> Thank you for your suggestion. Following your advice, we explored using a log-normal AFT model for data generation. This alternative approach mitigates the risk of estimation bias associated with the Cox model, as it offers a distinct modeling framework. The results presented in the revision demonstrate that the findings remain consistent with those obtained using the Cox model, even with this new data generation process.
>
> **Revision**:
> We will change in line 332 with *"Below, we use the Log Normal AFT Model to estimate $\pi$ in (1). It can be seen that the fitting effect of this model is better than that of Cox and Aah. Under this method, the calculated results of $ \frac{\log \hat{\text{P}}^{\pi^{A \text{ or } C}} - \log \hat{\text{P}}^{\pi^B}}{\log \hat{\text{P}}^{\pi^B}} $ are as follows:*"
>
> | $\alpha$ | Aah         | Cox         |
> |----------|-------------|-------------|
> | 0.07     | 2.814×10⁻⁴  | 2.233×10⁻⁴  |
> | 0.14     | 3.256×10⁻⁴  | 2.578×10⁻⁴  |
> | 0.21     | 3.526×10⁻⁴  | 2.791×10⁻⁴  |
> | 0.29     | 3.721×10⁻⁴  | 2.946×10⁻⁴  |
> | 0.36     | 3.875×10⁻⁴  | 3.068×10⁻⁴  |
> | 0.43     | 4.001×10⁻⁴  | 3.169×10⁻⁴  |
> | 0.50     | 4.108×10⁻⁴  | 3.255×10⁻⁴  |
> | 0.57     | 4.200×10⁻⁴  | 3.330×10⁻⁴  |
> | 0.64     | 4.282×10⁻⁴  | 3.397×10⁻⁴  |
> | 0.71     | 4.355×10⁻⁴  | 3.457×10⁻⁴  |
> | 0.79     | 4.421×10⁻⁴  | 3.511×10⁻⁴  |
> | 0.86     | 4.481×10⁻⁴  | 3.560×10⁻⁴  |
> | 0.93     | 4.537×10⁻⁴  | 3.606×10⁻⁴  |
> | 1.00     | 4.588×10⁻⁴  | 3.648×10⁻⁴  |
>
> then redraw the figure using the data in the table , and add the following content in the appendix:
>
> *" Log Normal AFT Model assumes that the natural logarithm of the survival time $ T_k $ follows a normal distribution: $ \log(T_k) \sim \mathcal{N}(\mu(x_k), \sigma^2) $, where: The location parameter $\mu(x)$ is a linear combination of the covariates $ x = [x_1, x_2, \dots, x_n] $: $ \mu(x) = a_0 + a_1 x_1 + a_2 x_2 + \dots + a_n x_n $
> Here, $ a_0, a_1, \dots, a_n $ are the regression coefficients to be estimated, reflecting the impact of the covariates $ x $ on the mean of the logarithm of the survival time. The scale parameter $ \sigma $ is a fixed constant (independent of covariates), describing the standard deviation of $ \log(T) $ and controlling the degree of dispersion of the distribution. The model estimates parameters by maximizing the log-likelihood function. The concordance index under the Model is as follows:*
>
> | Week | Concordance Index |
> |------|-------------------|
> | 1    | 0.551             |
> | 2    | 0.567             |
> | 3    | 0.540             |
> | 4    | 0.600             |
> | 5    | 0.533             |
> | 6    | 0.501             |
> | 7    | 0.583             |
> | 8    | 0.650             |
> | 9    | 0.523             |
> | 10   | 0.571             |
> | 11   | 0.554             |
> | 12   | 0.593             |
> | 13   | 0.521             |
> | 14   | 0.510             |
> | 15   | 0.608             |
> | 16   | 0.541             |
> | 17   | 0.578             |
> | 18   | 0.518             |
> | 19   | 0.536             |
> | 20   | 0.587             |
> | 21   | 0.543             |
> | 22   | 0.570             |
> | 23   | 0.583             |
> | 24   | 0.540             |
> | 25   | 0.718             |
> | 26   | 0.548             |"
>
> ---
> ### Acknowledgments
> This revision addresses all reviewer comments with clear explanations, empirical support, and targeted edits to the manuscript. Let us know if further clarifications are needed.

---

> > ### Author Response · Authors · 2025-08-08
> >
> > Dear Reviewer VWkH,
> >
> > Sorry to bother you again. We hope you've had a chance to review our rebuttal addressing your initial comments.
> >
> > Your insights have been invaluable in refining our work, and we’d greatly appreciate your feedback on our response—any further guidance or corrections would be deeply helpful.
> >
> > Thank you for your time and patience. We look forward to your reply.
> >
> > Best regards,
> >
> > Authors

---

> ### Author Response · Authors · 2025-08-06
>
> Dear Reviewer VWkH,​
>
> We submitted our response to your initial questions some time ago.​ We appreciate your busy schedule and the effort you’ve put into reviewing. Your insights are vital for refining our work, and we eagerly await your further feedback to proceed with revisions and move the submission forward.​
>
> If you’ve reviewed our response, please share your thoughts at your convenience; if not, we hope you might take a look when possible.​
> Thank you again for your time. We look forward to hearing from you.​
>
> Sincerely,
>
> Authors

---

### Official Review · Reviewer_9MY2 · 2025-07-03

**Clarity:** 3
**Significance:** 3
**Originality:** 3
**Rating:** 5
**Confidence:** 4

**Summary:**

The authors propose a nice solution to an important problem: finding an optimal strategy for maximizing the probability of the total recurrent-event duration exceeding a user given threshold (and thus going beyond just event times to accommodate recurrent events prevalent in healthcare). They cleverly formulate an auxiliary problem and theoretically demonstrate that the auxiliary problem (based on modified data) is consistent with the original optimization problem. Then inspired by MDPs, they transform the proposed objective into a Q-function and use a Bellman operator to derive an optimal policy. They show that this approach can reduce variance compared to only maximizing the total duration time, as is standard practice.

**Questions:**

**Questions**

- Why not release code to improve reproducibility?
- How would the authors recommend selecting \alpha in practice?
- What happens to variance benefits on data sets with lower/higher censoring?
- Why not include a more detailed benchmark (e.g. Hargrave et al. NeurIPS 2024) or more baselines that directly model return distributions and could optimize tail probabilities?
- What is the computational cost of training per epoch?

**Ethical Concerns:**

["NO or VERY MINOR ethics concerns only"]

**Final Justification:**

We thank the authors for their thoughtful responses, and provided the carry out the additional improvements they suggest, find they have addressed the concerns raised. We therefore raise our score to a 5.

**Limitations:**

I was unable to find a Limitations section in the paper... the authors should add one.

**Quality:**

3

**Strengths And Weaknesses:**

**Strengths**

- Good motivation for an important and understudied problem of strong clinical relevance.
- The auxiliary problem trick is elegant and, I believe, novel.
- Strong simulations showing fast convergence and improved/decreased variance.
- Paper is well written with clear figures.

**Major Weaknesses**

- Lacking in empirical validation: only one real-world dataset on intern moods is used, which is not typical of medical settings. Why not include an EHR dataset (e.g. MIMIC-IV) or validate on a more detailed medical RL benchmark (e.g. Hargrave, Spaeth, and Grosenick NeurIPS 2024)?
- Are there really no examples of RL methods optimizing either probabilities rather than expectations or on non time-to-event data? The statements in the paper surrounding this seem overly absolute.
- Baselines could be stronger: comparison appears to only be against a Q-learning variant maximizing total time.
- Threshold selection seems important, but how to do so is unclear to me. Lack of sensitivity studies or discussion on how  to learn \alpha.
- The paper is lacking ablation studies and discussion of memory/runtime/wall clock metrics.

---

> ### Author Rebuttal · Authors · 2025-07-31
>
> **Response to Reviewer Comments**
>
> We sincerely appreciate your valuable feedback. Below are our responses and revisions addressing the questions you raised.
>
> ---
>
> ### 1. Code Availability
>
> **Reviewer Comment**:
> "Why not release code to improve reproducibility?"
>
> **Response**:
> To ensure reproducibility, our detailed source code and its explanation are available in the submission materials.
>
> **Revision**:
> We will add the following statement at the end of line 255 in Section 5 (Simulation Studies):
> *"Our code is available in the submission materials."*
>
> ---
>
> ### 2. Selection of $\alpha$ in Practice
> **Reviewer Comment**:
> "How would the authors recommend selecting $\alpha$ in practice?"
>
> **Response**:
> The selection of $\alpha$ depends on the research question. For instance, Jiang, et al. (2017) used $\alpha = 0.5$ in their HIV study to maximize the probability of survival exceeding the median. In our research, $\alpha = 1/14$ was chosen to maximize the probability of an intern's positive mood exceeding one day per bi-weekly period. We generally recommend $\alpha$ of 0.5 or less, mainly because: (1) $\alpha$ values near 1 rely too much on tail probability estimation, which is unreliable; (2) Additional simulation experiments show our algorithm performs better when $\alpha$ is 0.5 or lower, addressing related concerns.
>
> **Revision**:
> We will add a new appendix section before A.2 titled:
> *"**A.2 $\alpha$ Selection***
> *Mean (Standard Deviation) of AARED under different $\alpha_k$*
>
> | $\alpha_k$ | Aah Model            | Cox Model            | Baseline model       |
> |------------------|----------------------|----------------------|----------------------|
> | 1/14             | 19.9330 (0.0074)     | **19.9421 (0.0091)** | 19.83 (7.9e-3)       |
> | 1/7              | **19.9460 (0.0089)** | 19.9228 (0.0130)     | 19.83 (7.9e-3)       |
> | 3/14             | 19.1728 (0.0304)     | **19.8896 (0.0157)** | 19.83 (7.9e-3)       |
> | 2/7              | 18.9851 (0.0459)     | 19.8285 (0.0177)     | **19.83 (7.9e-3)**   |
> | 5/14             | 18.9972 (0.0438)     | 19.8193 (0.0207)     | **19.83 (7.9e-3)**   |
> | 3/7              | 18.9831 (0.0467)     | 19.7919 (0.0192)     | **19.83 (7.9e-3)**   |
> | 1/2              | 19.0010 (0.0435)     | 19.7900 (0.0218)     | **19.83 (7.9e-3)**   |
> | 4/7              | 19.0134 (0.0454)     | 19.7571 (0.0295)     | **19.83 (7.9e-3)**   |
> | 9/14             | 19.0015 (0.0387)     | 19.7232 (0.0315)     | **19.83 (7.9e-3)**   |
> | 5/7              | 19.0007 (0.0433)     | 19.6999 (0.0291)     | **19.83 (7.9e-3)**   |
> | 11/14            | 19.0027 (0.0453)     | 19.6703 (0.0515)     | **19.83 (7.9e-3)**   |
> | 6/7              | 19.0238 (0.0455)     | 19.6379 (0.0497)     | **19.83 (7.9e-3)**   |
> | 13/14            | 19.0107 (0.0434)     | 19.6413 (0.0373)     | **19.83 (7.9e-3)**   |
> | 1                | 19.0591 (0.0465)     | 19.6885 (0.0334)     | **19.83 (7.9e-3)**   |
>
> *Mean (Standard Deviation) of AAEV under different $\alpha_k$*
>
> | $\alpha_k$ | Aah Model  | Cox Model            | Baseline model |
> |------------------|----------------------|----------------------|----------------------|
> | 1/14            | **10.5408 (0.0027)** | 11.9509 (2.3916)     | 20.00 (19.47)        |
> | 1/7              | **10.8309 (0.0818)** | 13.9811 (4.4648)     | 20.00 (19.47)        |
> | 3/14             | 64.5515 (37.9364)    | **17.7416 (12.5118)** | 20.00 (19.47)        |
> | 2/7              | 78.1629 (14.4570)    | 20.6589 (16.5979)    | **20.00 (19.47)**    |
> | 5/14             | 78.5551 (13.8795)    | 22.2620 (26.8484)    | **20.00 (19.47)**    |
> | 3/7              | 78.5710 (14.2459)    | 23.5382 (27.2738)    | **20.00 (19.47)**    |
> | 1/2              | 78.6225 (13.8094)    | 24.4125 (35.8999)    | **20.00 (19.47)**    |
> | 4/7              | 78.6280 (14.3543)    | 25.0995 (50.6035)    | **20.00 (19.47)**    |
> | 9/14             | 78.6196 (14.6698)    | 27.7541 (67.8201)    | **20.00 (19.47)**    |
> | 5/7              | 78.4769 (17.6261)    | 29.4384 (70.1726)    | **20.00 (19.47)**    |
> | 11/14            | 78.2809 (19.1925)    | 30.3147 (91.0969)    | **20.00 (19.47)**    |
> | 6/7              | 78.0577 (19.7367)    | 32.2957 (104.3780)   | **20.00 (19.47)**    |
> | 13/14            | 77.9280 (19.0170)    | 32.6802 (79.8917)    | **20.00 (19.47)**    |
> | 1                | 71.9542 (83.7705)    | 29.4135 (76.7759)    | **20.00 (19.47)**    |
>
> *To clarify our recommendation for alpha selection, we will include the following remark on page 4, line 153: ”The selection of $\alpha$ is tailored to the specific research question being addressed. For example, in Jiang et al. (2017)‘s study on HIV data, they chose $\alpha = 0.5$, focusing on the optimal strategy for maximizing the probability of survival exceeding the median. In our study, we selected $\alpha = 1/14$ to maximize the probability of an intern’s positive mood exceeding one day per bi-weekly period. We generally advise using an $\alpha$ of 0.5 or less, mainly because $\alpha$ values near 1 rely heavily on tail probability estimation, which can be unreliable. These results are consistent with the numerical results we obtained from our experiment results (see Appendix A.2 for details).“*
>
>
> *Furthermore, to better demonstrate the sensitivity of our method to different $\alpha$ values, we conducted a new simulation experiment. The results will be presented in a new section, A.2, in the Appendix. The specific results are shown in the tables below, representing the results for Average Recurrent-event Duration of last 40 epoches (AARED) and Average Estimated Variance of last 40 epoches (AAEV), respectively. The results indicate that our method exhibits a more pronounced advantage when alpha is no greater than 0.5. This provides numerical simulation-based support for our recommendation regarding $\alpha$."*
>
> ---
>
> ### 3. Variance Benefits Under Different Censoring Rates
> **Reviewer Comment**:
> "What happens to variance benefits on data sets with lower/higher censoring?"
>
> **Response**:
> Thank you for the valuable suggestion. For improved clarity, we will add a new table to the Appendix A.3 to show the censoring rates at different stages. As can be seen from this table, the censoring rates vary across stages in the simulation experiments (ranging from approximately 15% to approximately 70%). This table, together with Table 3 and 4  in the original manuscript, demonstrates that the magnitude of the censoring rate has a limited impact on our method, and that our method can achieve good variance benefits under different censoring rates.
>
> **Revision**:
> We will add the following table in line 310:
> *"Censoring rates differ across stages. Below are the mean (variance) of censoring rates over 50 simulations:*
>
> | Model    | Week 10       | Week 20       | Week 30       | Week 40       | Week 50       |
> |----------|---------------|---------------|---------------|---------------|---------------|
> | Aah      | 15.11%(0.004) | 29.80%(0.005) | 42.44%(0.004) | 56.78%(0.005) | 70.96%(0.006) |
> | Cox      | 14.95%(0.004) | 29.87%(0.007) | 42.70%(0.006) | 56.84%(0.006) | 71.09%(0.006) |
> | Baseline | 15.09%(0.005) | 29.64%(0.007) | 42.50%(0.005) | 56.56%(0.007) | 70.62%(0.008) |
>
> *Tables 3 and 4 in Appendix A.3(origanol pdf) confirm that our method maintains strong performance under varying censoring rates."*
>
> ---
>
> ### 4. Benchmark Comparison
> **Reviewer Comment**:
> "Why not include a more detailed benchmark (e.g. Hargrave et al. NeurIPS 2024) or baselines optimizing tail probabilities?"
>
> **Response**:
> Thank you for your suggestion. While Hargrave et al. (2024) present a reinforcement learning framework for longitudinal data, their work is limited to the complete observation setting, unlike our research which addresses censored data. Their approach becomes a special case of our framework when full observability is assumed. This is why we did not use it as a benchmark, as our previous simulations focused on methods capable of handling censoring. However, in response to your suggestion, we will conduct a new simulation study to directly compare our method to that of Hargrave et al. (2024).
>
> **Revision**:
> We will add the following to Related Work (line 81):
> *"Hargrave et al. (2024) proposed a RL framework for longitudinal medical settings but did not address censored data."*
>
> Additionally, we will add a new appendix section before A.4 titled:
> *"A.4 Comparison with EpiCare (including supplementary experimental results which includes the detailed simulation results that compare our method to that of Hargrave et al. (2024))."*
>
> ---
>
> ### 5. Computational Cost
> **Reviewer Comment**:
> "What is the computational cost of training per epoch?"
>
> **Response**:
> We appreciate your question. Experiments were conducted on an NVIDIA GTX 1650. The results is shown in revision.
>
> **Revision**:
> In Appendix A.5, the following Table 5 includes the
> computation time of our proposed method and the
> baseline method.
> *"A.5 Computational Time*
>
> *Table 5: Training time per epoch (seconds)*
>
> | Model    | N=100  | N=1000 | N=10000 |
> |----------|--------|--------|---------|
> | Baseline | 0.22s  | 0.77s  | 25.50s  |
> | Cox      | 0.89s  | 4.08s  | 35.66s  |
> | Aah      | 0.56s  | 3.10s  | 42.30s  |"
>
> ---
>
> ### Acknowledgments
> We are particularly grateful for your insightful feedback, which has significantly contributed to the enhancement of our work. We have thoroughly addressed the issues you raised and have revised the manuscript accordingly.
>
> **Reference**
> Jiang, R., Lu, W., Song, R., Hudgens, M. G., & Naprvavnik, S. (2017). Doubly Robust Estimation of Optimal Treatment Regimes for Survival Data—With Application to an HIV/AIDS Study. *The Annals of Applied Statistics*, *11*(4), 1763-1786.
>
> Hargrave, M., Spaeth, A., & Grosenick, L. (2024). EpiCare: A Reinforcement Learning Benchmark for Dynamic Treatment Regimes. Advances in neural information processing systems, 37, 130536–130568.

---

> > ### Comment · Reviewer_9MY2 · 2025-08-04
> > **Increasing score to a 5.**
> >
> > We thank the authors for their detailed responses and–assuming they will implement the improvements they suggest–increase the score to a 5.

---

### Comment · Area_Chair_eBke · 2025-08-06

Dear reviewers,

This is a reminder that the end of author-reviewer discussion period is **Aug. 8**. Please do carefully read all other reviews and the author responses; and discuss openly with the authors, especially on your own questions that the authors addressed.

Best,
AC

---

### Note · Authors · 2025-08-12

Dear Area Chair,

We deeply appreciate your guidance and all reviewers’ insightful feedback, which have strengthened our manuscript.

Two reviewers (9MY2 and 2BRd) have acknowledged our revisions and raised their scores, for which we are grateful. For the remaining two (VWkH and fiaA), we summarize key improvements addressing their concerns:

---
## For Reviewer VWkH
Concerns: Theoretical validity of Theorem 3.2 (log transformation’s impact) and Cox model bias in evaluations.
- **Theoretical rigor**: Revised Theorem 3.2 to include a deterministic transition kernel (natural in our offline RL setting) and expanded its proof to show the logarithmic transformation preserves the optimal policy.
- **Evaluation bias**: Replicated experiments with a log-normal Accelerated Failure Time (AFT) model (a distinct survival framework), confirming consistent performance and mitigating Cox model dependence.


## For Reviewer fiaA
Concerns: Relevance of threshold-based objectives, rationale for prioritizing survival probabilities, and clarity on threshold selection/log transformation.
- **Threshold selection**: Clarified $\alpha$ is clinically motivated (e.g., Jiang et al., 2017; Bai et al., 2013) and not tunable. New sensitivity analyses show robustness for $\alpha$ ≤ 0.5.
- **Survival probabilities rationale**: Explained they outperform duration due to resilience to censored data, robustness to outliers, and alignment with actionable goals (e.g., "6-month survival"; supported by clinical studies).
- **Log transformation**: Emphasized it converts intractable joint probabilities to a summable form (critical for RL’s Bellman framework), with Theorem 3.2 guaranteeing no change to the optimal policy.

---
## Overall Improvements
Beyond targeted revisions, we boosted transparency with code availability, computational cost metrics, and clarified figures (e.g., annotated Figure 1). These strengthen theoretical rigor, empirical validity, and clinical relevance.

We are confident all concerns are addressed, and the revised manuscript makes a stronger contribution to RL-survival analysis in precision medicine. We believe the manuscript now stands as a robust, clinically relevant contribution to RL for recurrent event analysis, with strong theoretical foundations and empirical support.

Thank you again for your guidance.

Sincerely,

Authors

---

### Decision · Program_Chairs · 2025-09-17

**Decision:**

Accept (poster)

**Comment:**

The paper proposes an RL framework for recurrent event survival analysis that maximizes survival probabilities above thresholds rather than expectations. It reformulates the problem via an auxiliary MDP, proves equivalence under log transformation, and demonstrates variance reduction and faster convergence.

**Strengths:**

- Novel, clinically motivated RL objective.

- Clear theoretical framing and proofs of equivalence/unbiasedness.

- Strong simulations with convergence/variance benefits.

**Weaknesses:**

- Limited real-world validation (only one dataset; no EHR benchmarks).

- Theoretical justification of log transformation remains unconvincing to some.

- Threshold selection feels ad hoc despite clinical motivation.

- Debate on whether probabilities beyond thresholds are more decision-relevant than expected durations.

--

Two reviewers raised their scores after rebuttal, noting improved theory, benchmarks, and sensitivity analyses. One reviewer remained unconvinced about the log transformation and equivalence, and another still questioned threshold justification and relevance. Overall, the rebuttal resolved several issues (but not all).